Dated 05 Sep 2024 – submitted to Ocean Science

**TEOS-10 and the Climatic Relevance of Ocean-Atmosphere Interaction**

Rainer Feistel

Leibniz Institute for Baltic Sea Research (IOW), 18119 Warnemünde, Germany

**Correspondence**: Rainer Feistel (rainer.feistel@io-warnemuende.de)

**Abstract**: Unpredicted observations in the climate system, such as recently an excessive ocean warming, are often lacking immediate causal explanations and are challenging the numerical models. As a highly advanced mathematical tool, the Thermodynamic Equation of Seawater – 2010 (TEOS-10) had been established by international bodies as an interdisciplinary standard and is recommended for use in geophysics, such as especially in climate research. From its very beginning, the development of TEOS-10 was supported by *Ocean Science* through publishing successive stages and results. Here, the history and properties of TEOS-10 are briefly reviewed. With focus on the air-sea interface, selected current problems of climate research are discussed and tutorial examples for the possible use of TEOS-10 in the associated context are presented, such as related to ocean heat content, latent heat and rate of marine evaporation, properties of sea spray aerosol, or climatic effects of low-level clouds. Appended to this article, a list of publications and their metrics is provided for illustrating the uptake of TEOS-10 by the scientific community, along with some continued activities, addressing still pending, connected issues such as uniform standard definitions of uncertainties, of relative humidity, seawater salinity or pH.

This article is dedicated to the Jubilee celebrating 20 years of Ocean Science.

This article is also dedicated to the memory of Wolfgang Wagner who sadly and unexpectedly has passed away on 12 August 2024. His contributions to TEOS-10 are truly indispensable constituents; Wolfgang was an essential co-author of various related documents and articles. He will deeply be missed.

*All the rivers run into the sea; yet the sea is not full;*
*unto the place from whence the rivers come, thither they return again.*
The King James Bible: Ecclesiastes, 450 – 150 BCE

*He wraps up the waters in his clouds,*
*yet the clouds do not burst under their weight.*
Holy Bible: New International Version, Job 26:8

*Of the air, the part receiving heat is rising higher.*
*So, evaporated water is lifted above the lower air.*
Leonardo da Vinci: Primo libro delle acque, Arundel Codex, ca. 1508

*Two-thirds of the Sun's energy falling on the Earth's surface is needed*
*to vaporize ... water ... as a heat source for a gigantic steam engine.*
Heinrich Hertz: Energiehaushalt der Erde, 1885

*The sea surface interaction is obviously*
*a highly significant quantity in simulating climate.*
Andrew Gilchrist, Klaus Hasselmann: Climate Modelling, 1986

*The climate of the Earth is ultimately determined*
*by the temperatures of the oceans.*
Donald Rapp: Assessing Climate Change, 2014

## 1 Introduction

Quite recently in 2024, climate research has published alarming news: "The world's oceans absorbed more heat in 2023 than in any other year since records began in the 1950s. … Data show that the heat stored in the upper 2,000 metres of oceans increased by 15 zettajoules (1 zettajoule is $10^{21}$ joules) in 2023 compared with that stored in 2022. This is an enormous amount of energy — for comparison, the world's total energy consumption in 2022 was roughly 0.6 zettajoules" (You 2024: p. 434). Dividing this value by the global ocean surface area and by the duration of a year, the reported ocean's average warming rate amounts to 1.3 W m$^{-2}$, and is apparently even increasing. "Earth's net global energy imbalance (12 months up to September 2023) amounts to +1.9 W m$^{-2}$, … ensuring further heating of the ocean" (Kuhlbrodt et al. 2024: p. E474). „Climate models struggle to explain why planetary temperatures spiked suddenly. … No year has confounded climate scientists' predictive capabilities more than 2023. … This sudden heat spike greatly exceeds predictions made by statistical climate models" (Schmidt 2024: p. 467).

The currently observed *ocean heat content* (OHC) represents a merely transient maximum after a decade-long systematic warming process in the past, see Fig. 18 in **Section 6**, which may proceed to even higher values in the future. In **Section 3**, thermodynamic aspects of related OHC definitions will be considered. Regarding the long-term period since 1971, "the drivers of a larger Earth energy imbalance in the 2000s than [before] are still unclear. … Future studies are needed to further explain the drivers of this change" (von Schuckmann et al. 2023: p. 1694). Laterally, the observed heat excess is unevenly distributed over the world ocean (Fig. 1), in contrast to what naively may be expected from rising atmospheric $CO_2$ concentrations. Rather, warming seems to be most pronounced in the cloudy austral and boreal west-wind belts. Selected thermodynamic relations between OHC and cloudiness are briefly discussed in **Section 6**.

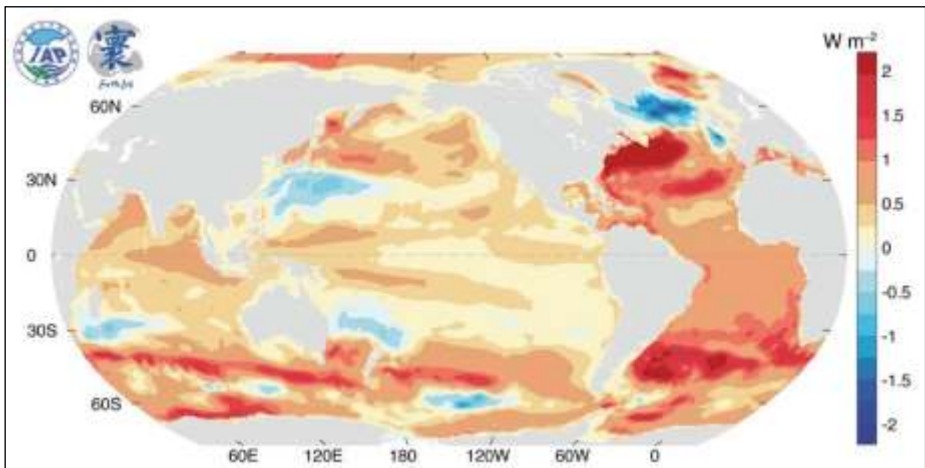

Fig. 1: Observed trend 1958 through 2022 of the upper 2000 m ocean heat content (WMO 2024). Image reproduction permitted by WMO Copyright.

Sunlight is the only available heat source of sufficient power to cause the observed warming, while the globally averaged geothermal heat flux is estimated to be just 0.087 W m$^{-2}$ (Pollack et al. 1993),

and is not expected to suddenly rise recently due to human impact. Irradiation is hampered by
clouds, dust and absorbing gases, and water surface reflection such as by whitecaps, waves or
plankton layers (Cahill et al. 2023). Heat absorbed in the water column may effectively exit the ocean
again only across the air-sea interface via sensible, radiative and latent heat flux. All these effects
may vary in the climate system in a complicated, mutually interacting manner. Typically, present
numerical climate models suffer from an "ocean heat budget closure problem" (Josey et al. 1999)
and describe the ocean-atmosphere heat flux only to within uncertainties between 10 W m$^{-2}$ and 30
W m$^{-2}$ (Josey et al. 2013). According to recent model comparison studies, many of those "models fail
to provide as much heat into the ocean as observed" (Weller et al. 2022: p. E1968). Dynamical
models, rather than observed correlations, are the most reliable tools for the detection and
verification of causal relations (Feistel 2023), however, such as in this case of air-sea interaction,
large uncertainties may prevent any significant conclusions to be drawn regarding the causes of the
observed ocean warming rate of 1.3 W m$^{-2}$.
Of the increasing amount of water vapour contained in the global troposphere, 85 % results from
ocean evaporation (Gimeno et al. 2013). Corresponding to 1200 mm annual evaporation (Budyko
1963, 1984, Baumgartner and Reichel 1975, Peters-Lidard et al. 2019), the associated latent heat flux
of about 95 W m$^{-2}$ per ocean surface area represents the strongest energy supply for the
atmospheric dynamics (Albrecht 1940) and at the same time the strongest cooling process of the sea.
This flux depends sensitively on the relative humidity (RH) at the water surface; an RH increase by 1
%rh can be estimated to reduce evaporation by 5 W m$^{-2}$ (Feistel 2015, 2024, Feistel and Hellmuth
2021, 2023), so that minor additional 0.2 %rh may already suffice to warm up the ocean by the
observed 1.3 W m$^{-2}$. Unfortunately, marine RH is observed only with uncertainties between 1 and 5
%rh (Lovell-Smith et al. 2016), or, accordingly, between 5 and 25 W m$^{-2}$ of latent heat flux, which is
roughly corresponding to unknown variations ranging up to 50 … 250 mm evaporation per year. It
remains unclear to what extent minor, yet unnoticed changes in marine RH may be responsible for
the recent ocean warming.

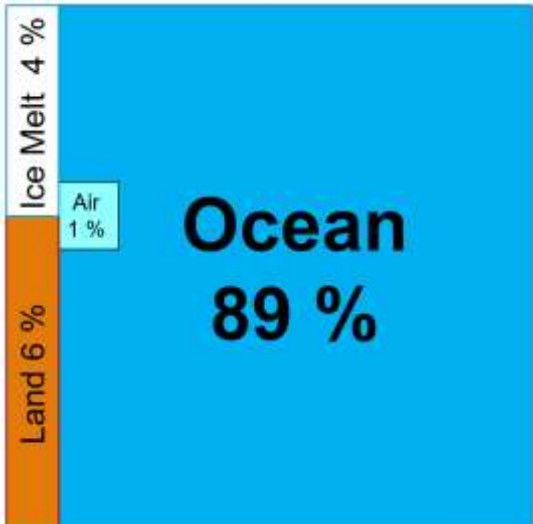


Fig. 2: Heat fractions stored additionally in the different parts of the Earth system 1960–2020 (values
from von Schuckmann et al. 2023), represented graphically by partial areas. Obviously, the oceans
dominate global warming.

According to Fig. 2, a paramount share of 94 % of global warming occurs in different phases and
geophysical mixtures of water, in particular in seawater. Considering this situation, the *Scientific*
*Committee on Oceanic Research* (SCOR) in cooperation with the *International Association for the*
*Physical Sciences of the Oceans* (IAPSO) decided at its 2005 Cairns meeting the establishment of the
*SCOR/IAPSO Working Group 127 on Thermodynamics of Seawater* (WG127) (Millero 2010, Pawlowicz
et al. 2012, Smythe-Wright et al. 2019), which held its inaugural meeting in 2006 at Warnemünde
(Fig. 3). It had been recognised that "modelling of the global heat engine needs accurate expressions
for the entropy, enthalpy, and internal energy of seawater so that heat fluxes can be more accurately
determined in the ocean" (Millero 2010: p. 28) while such properties were not available from the
thermodynamic seawater standard at that time, the 1980 Equation of State of Seawater (EOS-80)
(Fofonoff and Millard Jr. 1983).

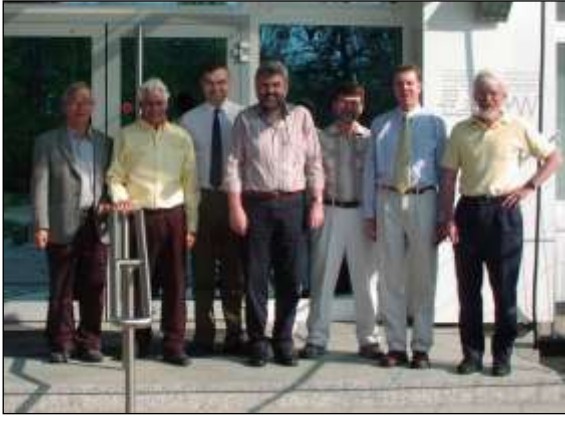


Fig. 3: Participants of the 2006 kick-off meeting of SCOR/IAPSO WG127 at the Leibniz Institute for
Baltic Sea Research (IOW) in Warnemünde, Germany. From left to right: Chen-Tung Arthur Chen
(Taiwan), Frank Millero (USA), Brian King (UK), Rainer Feistel (WG vice chair, Germany), Daniel Wright
(Canada, deceased 2010), Trevor McDougall (WG chair, Australia) and Giles Marion (USA).

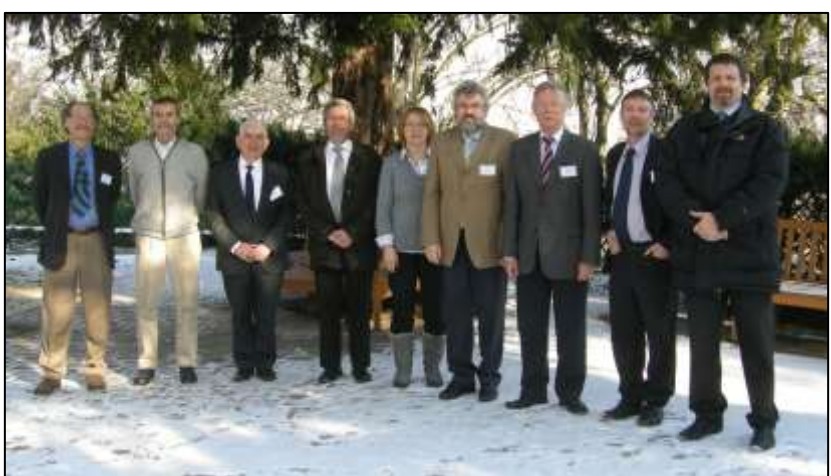


Fig. 4: Participants of the BIPM-IAPWS meeting in February 2012 at the Pavillon de Breteuil, Sèvres.
From left to right: Dan Friend (IAPWS), Karol Daučik (IAPWS president), Jeff Cooper (IAPWS), Alain
Picard (BIPM, deceased 2015), Petra Spitzer (WG127), Rainer Feistel (WG127), Michael Kühne
(director BIPM), Andy Henson (BIPM) and Robert Wielgosz (BIPM).

The foundation of WG127 happened almost coincidently with the establishment of the *Ocean*
*Science* journal of the *European Geosciences Union* (EGU) in 2004/05. The development of the new
standard by WG127, the Thermodynamic Equation of Seawater – 2010 (TEOS-10) was very
successfully supported by Ocean Science, publishing the Special Issue #14 on "Thermophysical
Properties of Seawater" with 16 articles between 2008 and 2012 (Feistel et al. 2008a). **Appendix A**
reports the current metrics of this Special Issue. Also in 2008, at its conference in Berlin, Germany,
the *International Association for the Properties of Water and Steam* (IAPWS) established a new
*Subcommittee on Seawater* (SCSW) that cooperated closely with WG127. In the form of carefully
verified mathematical formulations for properties of water, ice, seawater and humid air, IAPWS
adopted nine fundamental documents related to TEOS-10 (IAPWS AN6-16 2016), see **Appendix A**.
With respect to problems yet pending after the official adoption of TEOS-10, especially for the
preparation of future novel international definitions of seawater salinity, seawater pH and
atmospheric relative humidity (Feistel et al. 2016, Pawlowicz et al. 2016, Dickson et al. 2016, Lovell-
Smith et al. 2016), the standing *IAPSO/SCOR/IAPWS Joint Committee on the Properties of Seawater*
(JCS) was established in 2012. In 2011, IAPWS also extended its cooperation with the *International
Bureau for Weights and Measures* (BIPM), see Fig. 4. Further details on TEOS-10 (IOC et al. 2010,
McDougall et al. 2013, Feistel 2018, Wikipedia 2024) are available from the TEOS-10 homepage,
www.teos-10.org, and are briefly reviewed in **Section 2** and **Appendix B**.
In the context of the predecessor EOS-80, the *ocean heat content* (OHC) was defined in terms of
*potential temperature* (Abraham et al. 2013). Improving this method, TEOS-10 entropy and enthalpy
of seawater provided a proper quantitative basis for a novel, thermodynamically rigorous definition
of the OHC in the form of seawater *potential enthalpy* (McDougall 2003, McDougall et al. 2013,
Graham and McDougall 2013, McDougall et al. 2021), equivalently defined as *Conservative
Temperature* and briefly discussed in **Section 3**.

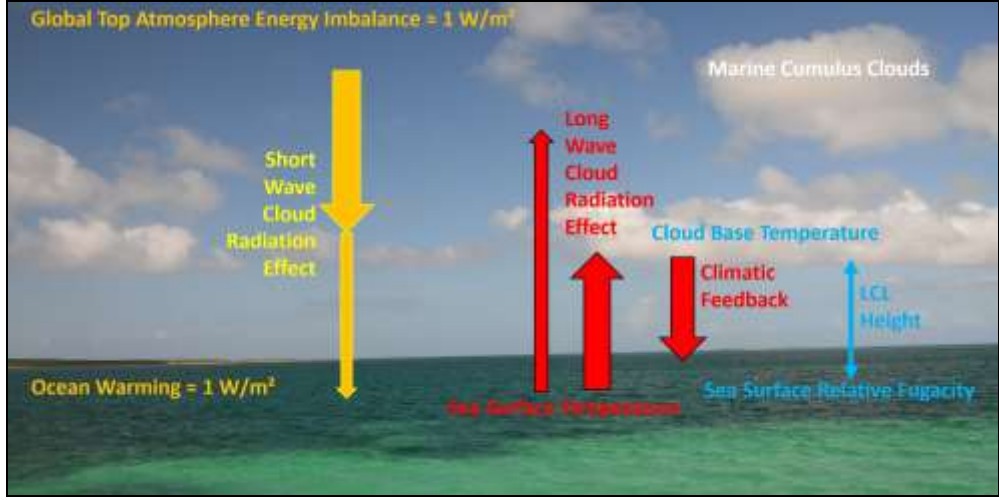


Fig. 5: Schematic of *cloud radiation effects* (CRE). The *short-wave effect* (SW CRE) controls the
downward flux of solar irradiation while the *long-wave effect* governs the infrared radiation balance
between water surface and cloud base. By thermal convection, cumulus clouds emerge at the
isentropic *lifted condensation level* (LCL). Figure from Feistel and Hellmuth (2024b)

Currently implemented parameterisations of marine evaporation rates in the form of historical
*Dalton equations* (Stewart 2008, Josey et al. 1999, 2013) may be replaced by TEOS-10 chemical
potentials which provide the proper quantitative basis for a thermodynamically rigorous formulation
of non-equilibrium Onsager forces and fluxes in terms of *relative fugacity* (RF) of humid air (Kraus
and Businger 1994, Feistel and Lovell-Smith 2017, Feistel and Hellmuth 2023, 2024a), as described in
**Section 4.** *Relative humidity* (RH) is defined relative to the saturation state of moist air, which in turn
is controlled by the chemical potentials of water in the gas and liquid phase. It is only natural,
therefore, to define RH in terms of chemical potentials, which in fact is performed by RF. The
uncertainty of latent heat flux with respect to the uncertainty of surface RH observation is shown to
be significantly larger than the observed warming of $1.3 \ \mathrm{W \ m^{-2}}$, so that this unpredicted warming
may or may not be caused by so-far ignored minor RH increase.
The conceptual model of sea air as a two-phase composite thermodynamic system is outlined in
**Section 5**. The roles of enthalpy, chemical potential and entropy are explained by means of explicit
theoretical descriptions of three simplified tutorial examples, (i) for the latent heat of evaporation,
(ii) for the heat capacity of humid air containing sea spray, and (iii) for the entropy production of
irreversible evaporation.
Clouds do not only release the latent heat which water vapour has carried away from the ocean, they
also interfere substantially in the global radiation balance, cooling the surface by reflecting short-
wave solar irradiation, and warming the surface by sending back down long-wave thermal radiation,
see Fig. 5. In the course of global warming, cloudiness has been found to exhibit a systematic trend of
reduction, see **Section 6**, which affects the ocean heat content in a non-trivial, non-uniform manner.
Marine cumulus clouds arise by isentropic uplift of thermal convection. Their height controls their
temperature and their thermal downward radiation, affecting the ocean's energy balance. Updating
previous results (Romps 2017) for the *lifted condensation level* (LCL) of marine cumulus clouds to
thermodynamically rigorous TEOS-10 standard equations (Feistel and Hellmuth 2024b), the radiative
effect of those clouds can be estimated from *sea-surface temperature* (SST) and surface relative
humidity. This effect turns out to be weakly cooling and cannot provide a reasonable explanation for
the so-far unclear strong ocean warming. The effect of increasing SST in the past decades turns out
to be minor in comparison to that caused by RH uncertainty.
**Section 7** provides a summery of this paper, **Appendix A** reports collections of publications with
respect to TEOS-10 as well as their metrics, and **Appendix B** gives a short introduction into the
concept of thermodynamic potentials.

**2   Thermodynamic Equation of Seawater – 2010 (TEOS-10)**
In the climate system, the omnipresent and dominant substance is water in various phases and
mixtures. For example, "water vapor is by far the most important greenhouse gas, in the sense that it
absorbs more irradiance from the Earth than all other greenhouse gases combined" (Rapp 2014: p.
381). Textbooks and other publications offer numerous collections of various different property
equations for water, ice, seawater or moist air, but uncertainties and mutual consistencies of those
equations are often unclear. To improve this situation, a novel *Thermodynamic Equation of Seawater*
*– 2010* (TEOS-10) was developed by the members of the SCOR/IAPSO Working Group 127 (WG 127)
in close cooperation with the International Association for the Properties of Water and Steam
(IAPWS). TEOS-10 is described in a detailed Manual (IOC et al. 2010) and has been adopted and
recommended by IOC-UNESCO (2009) in Paris and by the IUGG (2011) in Melbourne, see also Feistel
(2008b, 2012, 2018), Valladares et al. (2011) and Pawlowicz et al. (2012). Starting in 2008 with a
Special Issue of *Ocean Science* (Feistel et al. 2008a), a large number of scientific publications has
appeared in the meantime, supporting, extending or exploiting TEOS-10. A collection of selected
papers related to TEOS-10 is summarised in Appendix A together with metrics that illustrate the
growing uptake of TEOS-10 by the scientific community.
The development of the first numerical thermodynamic Gibbs potentials (see Appendix B) for
seawater (Feistel 1991, 1993, Feistel and Hagen 1995) was based on the works of Millero and Leung
(1976) and Millero (1982, 1983), together with high-pressure background data of the previous EOS-
80 standard (Unesco 1981). Independently of that, a Helmholtz potential for pure fluid water had
been adopted by IAPWS in 1996 at Fredericia (Harvey 1998, Wagner and Pruß 2002). These were the
key activities which eventually culminated in the formulation of TEOS-10 about two decades later. By
combining those equations for pure and seawater, some known pending problems of EOS-80
(Fofonoff and Millard Jr. 1983) could incidentally be resolved (Feistel 2003). In the end, TEOS-10 has
been assembled from four basic thermodynamic potentials derived from mutually consistent, most
comprehensive and accurate datasets of measured properties available at that time. Those
potentials are (IAPWS R6-95 2016, IAPWS R10-06 2009, IAPWS R13-08 2008, IAPWS G8-10 2010,
respectively):

(i)     A Helmholtz function of fluid water, $f^{\mathrm{F}}(T, \rho) \equiv f^{\mathrm{W}}(T, \rho) \equiv f^{\mathrm{V}}(T, \rho)$, known as the
IAPWS-95 formulation (Wagner and Pruß 2002), which is identical for liquid water,
$f^{\mathrm{W}}(T, \rho)$ and for water vapour, $f^{\mathrm{V}}(T, \rho)$. It describes de-aerated water of a fixed
isotopic composition, termed *Standard Mean Ocean Water* (SMOW), with density $\rho$ and
temperature $T$.

(ii)    A Gibbs function of ambient hexagonal ice I, $g^{\mathrm{Ih}}(T, p)$, or IAPWS-06 formulation (Feistel
and Wagner 2006), see Tables A2 and A3 of Appendix A, depending on pressure $p$.

(iii)   A Gibbs function of *IAPSO Standard Seawater*, $g^{\mathrm{SW}}(S, T, p)$, or IAPWS-08 formulation
(Feistel 2008a), see Tables A2 and A3 of Appendix A. The variable $S$, at which a subscript
A is omitted here for simplicity, is the specific or *Absolute Salinity*, the mass fraction of
dissolved salt in seawater, which differs from *Practical Salinity*, $S_{\mathrm{P}}$, measured by present-
231               day oceanographic instruments, as well as from various other obsolete salinity scales
(Millero et al. 2008). Throughout this paper, the term "salinity" is exclusively short hand
for TEOS-10 Absolute Salinity. Sea salt is assumed to have stoichiometric *Reference*
*Composition*. The pure-water limit, $g^{\mathrm{SW}}(0, T, p) = g^{\mathrm{W}}(T, p)$, is the Gibbs function of
liquid water computed from the IAPWS-95 Helmholtz function $f^{\mathrm{W}}(T, \rho)$. For brackish
seawater, $g^{\mathrm{SW}}$ has implemented Debye's root law of dilute electrolyte solutions (Landau
and Lifschitz 1966, Falkenhagen et al. 1971).

(iv)   A Helmholtz function of humid air, $f^{\mathrm{AV}}(A, T, \rho)$, or IAPWS-10 formulation (Feistel et al.
2010a), see Tables A1 and A2 of Appendix A. The variable $A$ is the mass fraction of dry air
admixed with water vapour, so that $q = 1 - A$ is the *specific humidity*. The dry-air limit
$f^{\mathrm{AV}}(1, T, \rho) = f^{\mathrm{A}}(T, \rho)$ equals, up to modified reference-state conditions, the equation
of state of Lemmon et al. (2000). The air-free limit $f^{\mathrm{AV}}(0, T, \rho) = f^{\mathrm{V}}(T, \rho)$ equals the
IAPWS-95 Helmholtz function of water vapour. In $f^{\mathrm{AV}}$, the interaction of water vapour
with dry air is described by 2nd and 3rd virial coefficients.

Thermodynamic potentials include certain adjustable constants expressing the absolute energies and
entropies of the particular substances, which are not available from measurement (Planck 1906,
Feistel 2019b) and have, in turn, no effect on measurable properties derived from those potentials.
In fact, among the comprehensive experimental data sets from which the TEOS-10 equations were
derived, none of those are suitable for fitting the empirical coefficients that represent absolute
energies and entropies of those equations. For this reason, the International Conference on the
Properties of Steam at London defined in 1967 the common triple point of water as the reference
state at which those absolute values were arbitrarily set. Since then, no evidence has appeared for
putative conflicts caused by such settings with any technical or scientific applications of the
equations. Despite this, Feistel and Wagner (2006) and Feistel et al. (2008b) discuss the
implementation of alternative residual entropies of water, if that should be of interest in exceptional
applications of TEOS-10. For recent discussions of Pauling's absolute "residual" entropy at zero kelvin
and Nernst's Third Law of thermodynamics, see Kozliak and Lambert (2008), Gutzow and Schmelzer
(2011), Takada et al. (2015), Schmelzer and Tropin (2018), Feistel (2019b), or Shirai (2023).

The TEOS-10 reference states (Feistel et al. 2008b, 2010a) are the triple point of water, $T_{\mathrm{TP}} =$
273.16 K, $p_{\mathrm{TP}} = 611.654\ 771$ Pa, where the conditions

$\qquad \eta_{\mathrm{TP}}^{\mathrm{W}} = 0, \qquad e_{\mathrm{TP}}^{\mathrm{W}} = 0,$ (1)

are imposed, and the standard ocean state at Absolute Salinity, $S_{\mathrm{SO}} = 35.165\ 04$ g kg$^{-1}$, absolute
temperature, $T_{\mathrm{SO}} = 273.15$ K, and absolute pressure, $p_{\mathrm{SO}} = 101\ 325$ Pa, with the conditions for sea
salt,

$\qquad \eta_{\mathrm{SO}}^{\mathrm{SW}} = 0, \qquad h_{\mathrm{SO}}^{\mathrm{SW}} = 0,$ (2)

and for dry air,

$\qquad \eta_{\mathrm{SO}}^{\mathrm{A}} = 0, \qquad h_{\mathrm{SO}}^{\mathrm{A}} = 0.$ (3)

Here, $\eta$, $e$ and $h$, respectively, are specific entropy, internal energy and enthalpy of water
(superscript W), seawater (superscript SW) and dry air (superscript A). The TEOS-10 potential
functions and properties derived thereby are numerically implemented in two different libraries, the
Sea-Ice-Air (SIA) and the Gibbs-Seawater (GSW) libraries, see Table A4 in Appendix A.

The SIA library includes empirical coefficients only in the four fundamental potentials (Feistel 2010d,
Wright et al. 2010). All other potential functions and properties are derived strictly by mathematical
operations to ensure consistent results, even at the cost of low computation speeds as a result of
stacked iteration procedures. All quantities are exclusively expressed in basic SI units such as kg, m, J
or Pa. A more recent extension of SIA code is reported in Feistel et al. (2022) for the computation of
relative fugacity.

The GSW library is tailored for oceanographic models, optimised in computation speed (Roquet et al
2015). For fast numerical evaluation, GSW procedures contain new empirical coefficients determined
from the SIA library functions by regression. Units and variables are adjusted to common
oceanographic practice such as pressure in decibars relative to surface pressure, or temperatures in
°C. *Conservative Temperature* (CT) is used as a new preferred thermal variable. An additional
thermodynamic potential has been constructed (McDougall et al. 2023) that supports the use of CT
universally as an independent variable.

**3   Potential Enthalpy and Ocean Heat Content (OHC)**
Thermodynamically, the term "Ocean Heat Content" (OHC) is a sloppy wording. "Content" means a
state quantity of a body or volume while, by contrast, "heat" is an exchange quantity rather than a
state quantity. "We have … a right to speak of heat as a *measurable quantity*, … however, … we have
no right to treat heat as a *substance*" (Maxwell 1888: p. 7). "The obsolete hypothesis of heat being a
substance is excluded" (Sommerfeld 1988: p. 6). "Heat is not a substance! More formally: Heat is not
a thermodynamic function of state" (Romer 2001: p. 107). This distinction is qualitatively
fundamental (Feistel 2023). Physical conservation quantities such as energy or mass have the key

306 property that the change of that quantity in a volume equals the flux of that quantity across the
307 boundary (Landau and Lifschitz 1966, Glansdorff and Prigogine 1971), but this does not apply to
308 "heat". For example, a heat engine receives a permanent net heat flux without getting permanently
309 hotter. While asking how much "heat" is contained in the ocean may find ambiguous answers, it is
310 well defined to say how much heat has entered or left the ocean across its boundary by a specified
311 process that transfers the ocean from a certain state of reference to the current state of interest. In
312 this section, based upon TEOS-10, related states and processes are described which may properly
313 specify what is commonly termed OHC. This consideration intrinsically connects OHC with ocean-
314 atmosphere exchange processes relevant for climate change.

315 Since a long time, measuring and calculating the ocean's "heat" has been a question of central
316 interest to oceanography. Recently, this issue has become even more important and urgent in the
317 context of climate change. "The total energy imbalance at the top of atmosphere is best assessed by
318 taking an inventory of changes in energy storage. The main storage is in the ocean" (Abraham et al.
319 2013: p. 450). The conventional approach is a formally defined mathematical procedure based on
320 potential temperatures. "Changes to ocean heat content (OHC) can be calculated from
321 measurements of the temperature evolution of the ocean. The OHC is attained from the difference
322 of the measured potential temperature profile and the potential temperature climatology. This
323 difference is integrated over a particular reference depth (for instance, 700 m) and is multiplied by a
324 constant ocean density reference and heat capacity" (Abraham et al. 2013: p. 468). However, in
325 representing a kind of "heat substance", this OHC definition has no rigorous thermodynamic
326 justification, and the relation to processes of ocean-atmosphere heat fluxes is not entirely clear. If a
327 sea-air heat flux of 1 W m$^{-2}$ warms up the atmosphere, by what rate exactly will that OHC decrease?

328 Making the seawater properties entropy and enthalpy quantitatively available, TEOS-10 has offered a
329 thermodynamically improved option for defining OHC (McDougall et al. 2021), in the form of the
330 integral over the ocean volume,

331 $OHC = \int h^{\mathrm{SW}}(S, \eta, p_0)\, \rho^{\mathrm{SW}}(S, \eta, p)\mathrm{d}V$ .    (4)

332 Here, $h^{\mathrm{SW}}(S, \eta, p_0)$ is the *potential enthalpy* (McDougall 2003) relative to the surface pressure, $p_0$,
333 and $\rho^{\mathrm{SW}}(S, \eta, p)$ is the in-situ mass density at the pressure $p$ of a parcel with salinity and entropy
334 equal to those before. This definition can be understood in terms of both, a specified process of heat
335 exchange, and a reference state relative to which OHC is counted, as follows (Feistel 2024):

336  (i)  A virtual **heat exchange process** supporting the definition (4) is sketched in Fig. 6. In turn,
337    each ocean parcel with in-situ properties $(S, \eta, p)$ is lifted to the surface pressure $p_0$,
338    keeping its salinity and entropy constant. There, it reversibly exchanges heat, $\mathrm{d}h = T\mathrm{d}\eta$,
339    with a measuring device until the parcel's entropy has reached a certain reference value,
340    $\eta_{\mathrm{ref}}$, while the parcel's salinity remains unchanged. Subsequently, the heat is reversibly
341    put back to the parcel which is then returned to its original location. The work required
342    to lift and lower the parcel is balanced because the parcel's thermodynamic state is
343    exactly the same before and after the balanced reversible heat exchange across the
344    surface. The "heat content" defined this way for a single parcel is added up then over all
345    ocean parcels to result in its total OHC value.
346  (ii)  The **reference state** relative to which OHC is measured may freely be specified at will,
347    but beneficially be chosen with respect to its convenience or usefulness. In the case of
348    eq. (4), the OHC reference state is zero potential enthalpy (or zero Conservative
349    Temperature, McDougall 2003) of all ocean parcels.

The process depicted in Fig. 6 measures the total heat flux $\int dh = \int T d\eta$ which changes the entropy
of the given sample from the current value, $\eta$, to some arbitrary reference value, $\eta_{\mathrm{ref}}$, and this way,
the process also changes the parcel's enthalpy from $h^{\mathrm{SW}}(S, \eta, p_0)$ to $h^{\mathrm{SW}}(S, \eta_{\mathrm{ref}}, p_0)$. Integration
over all ocean samples results in an OHC value of
$$OHC^* = \int \left[ h^{\mathrm{SW}}(S, \eta, p_0) - h^{\mathrm{SW}}(S, \eta_{\mathrm{ref}}, p_0) \right] \rho^{\mathrm{SW}}(S, \eta, p) \mathrm{d}V. \qquad (5)$$
While the choice of the OHC reference state is - in principle - entirely arbitrary, such as simply putting
$\eta_{\mathrm{ref}} = 0$, it is reasonable to better adapt this selection to the purpose of the OHC definition. The
main purpose of estimating OHC is keeping track of the ocean's long-term energy balance, in
particular of the ocean's share of global warming. Three conditions appear immediately plausible in
order to achieve this goal,
(i)  *The OHC definition should ensure that OHC differences represent a suitable spatial*
*integral over the heat fluxes crossing the ocean's boundaries.* As discussed in more detail
in Section 5.3, production of entropy, $\mathrm{d}_i \eta$, caused by irreversible processes between
different parcels within the ocean, does not affect the ocean's total enthalpy budget.
This is quite in contrast to entropy exchange, $\mathrm{d}_e \eta$, of the given sample in the form of
reversible heat flux across its boundary. Such irreversible processes affect the ocean's
total potential enthalpy much less than its total entropy (McDougall et al. 2021). For this
reason the OHC reference state should explicitly be defined in terms of potential
enthalpy, $h^{\mathrm{SW}}(S, \eta_{\mathrm{ref}}, p_0)$, and this way only implicitly in terms of entropy by specifying
$\eta_{\mathrm{ref}}(S)$.
(ii)  *Provided that the ocean's mass remains the same between any two ocean states (1) and*
*(2), the difference OHC(1) – OHC(2) should depend only on the surface heat flux balance*
*during the time in between. In particular, differences OHC(1) – OHC(2) should not depend*
*on the OHC reference state.* For this reason, the OHC reference value should be
independent of changes occurring in the density distribution, $\rho^{\mathrm{SW}}(S, \eta, p)$. This can be
achieved by assigning to each ocean parcel the same reference potential enthalpy,
$h^{\mathrm{SW}}(S, \eta_{\mathrm{ref}}, p_0) = const$, even though such a state may hardly ever be observed in the
real ocean.
(iii)  *Quantitatively, OHC values estimated at different times or places should be mutually*
*comparable without estimation bias resulting from possibly changing methods of OHC*
*calculation.* For this reason, resulting OHC values should be independent of the inevitable
arbitrary, physically irrelevant reference-state conditions imposed on energy and
entropy, such as eqs. (1)-(3). This can be achieved by assigning to each ocean parcel the
same standard-ocean enthalpy as its reference potential enthalpy, $h^{\mathrm{SW}}(S, \eta_{\mathrm{ref}}, p_0) =$
$h_{\mathrm{SO}}$. In the special case of TEOS-10 enthalpy, this value is defined by eq. (2), $h_{\mathrm{SO}} = 0$.
This choice is implicitly made by the definition (4) but needed to be considered explicitly
as soon as alternative equations for seawater enthalpy or entropy are employed, such as
those of Millero and Leung (1976) and Millero (1982, 1983).

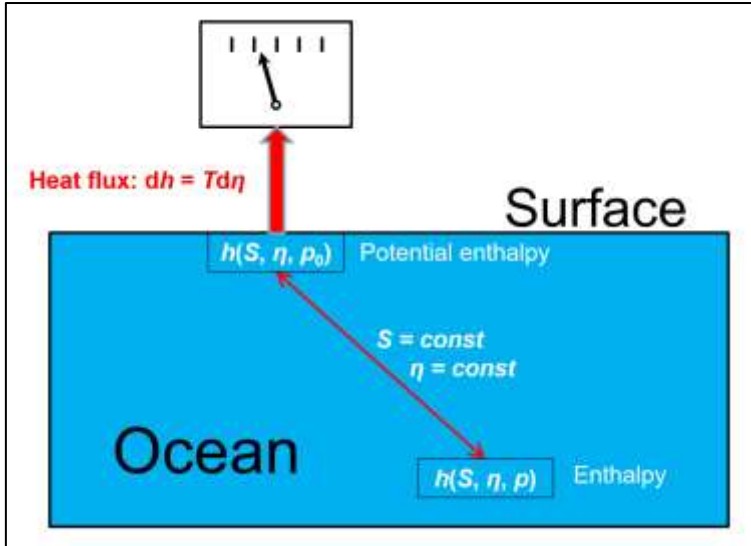


Fig. 6: Schematic of a conceptual process defining the ocean heat "content" (OHC) by measuring heat
flux across the ocean boundary according to eqs. (4) and (5).

In a sense consistent with the previous OHC definition (Abraham et al. 2013), also a climatological
average state could in principle be chosen as the OHC reference. However, this option includes the
problem that the salinity distribution of the current ocean may differ from the reference ocean, and
that thermodynamically properly treating the required salt exchange processes at the surface may
turn the issue unnecessarily complicated. A detailed comparison of the OHC definition (4) with its
precursor prior to TEOS-10 is provided by McDougall et al. (2021). OHC as a part of the total energy
balance of the ocean is analysed by Tailleux (2010, 2018) and Tailleux and Dubos (2024).

**4    Relative Fugacity and Ocean Evaporation Rate**
"The global water cycle and the exchange of freshwater between the atmosphere and ocean is
poorly understood. … It has been predicted that increasing global temperatures will lead to an
enhanced global water cycle" (Holliday et al. 2011: p. 34). In the past, several climate researchers
have argued that along with global warming the marine evaporation has or will be "amplified" or
"intensified" (Feistel and Hellmuth 2021). However, it was not always made clear whether this may
mean that (a) in the course of a year, more water vapour is transferred from the global ocean to the
atmosphere, or (b) that the global mean evaporation rate remains unchanged while locally or
temporally, evaporation is more intense, or (c) any combination of the two variants. Conclusions of
kind (a) were drawn by renowned climatologists such as Budyko (1984), Flohn et al. (1992), Yu
(2007), Randall (2012), Francis (2021) or Zhang et al. (2021).
By contrast, in favour of option (b), the currently observed ocean warming at a rate about 1 W m$^{-2}$
does not support assumptions of an enhanced hydrological cycle with related latent-heat cooling,
rather, it more likely suggests a slight reduction of evaporation. Two decades ago, Held and Soden
(2006: p. 5687-5689) had already clearly stated that "it is important that the global-mean
precipitation or evaporation, commonly referred to as the strength of the hydrological cycle, does
not scale with Clausius–Clapeyron. … We can, alternatively, speak of the mean residence time of
water vapor in the troposphere as increasing with increasing temperature." Subsequent observations
have underpinned their statement.

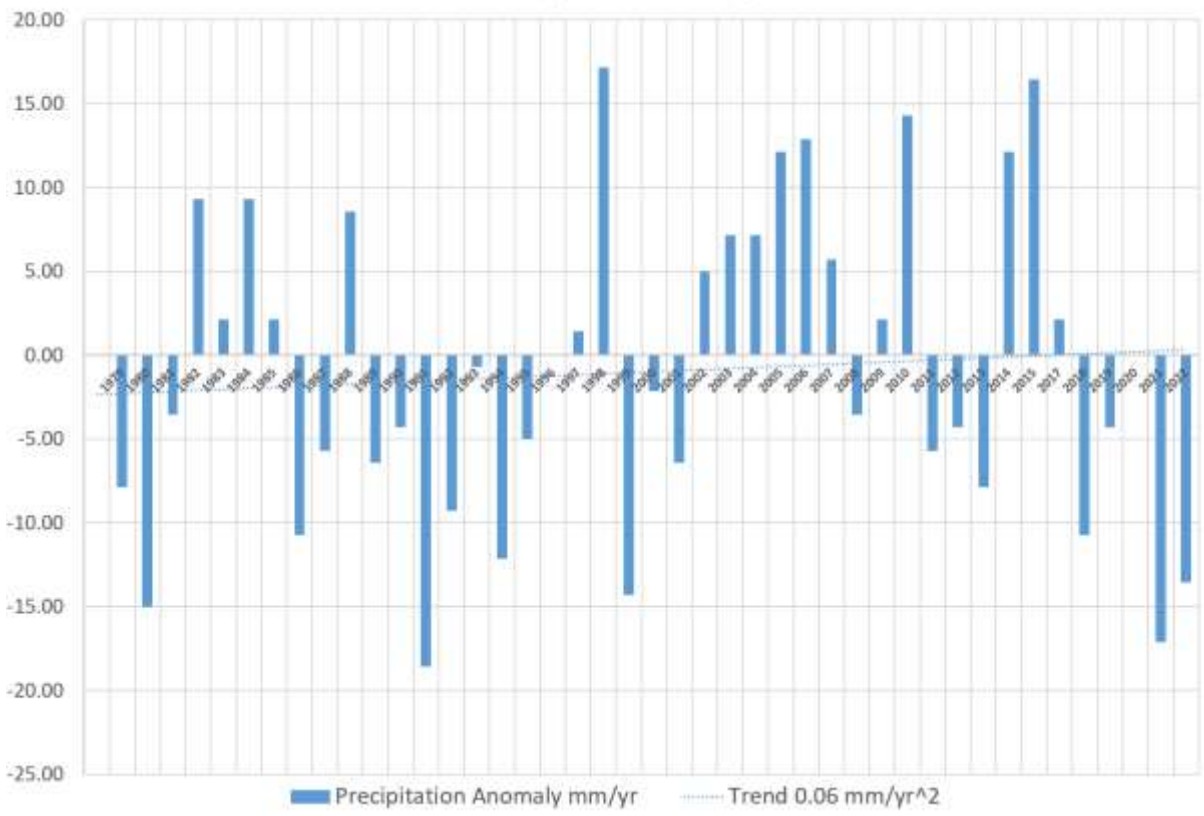

Fig. 7: Global mean precipitation anomaly 1979-2022 in mm yr$^{-1}$. The values displayed exhibit a minor increasing trend (dotted line) of 0.06 mm yr$^{-2}$. Data from Vose et al. (2023)

Between 1979 and 2022, annual mean global precipitation values, see Fig. 7, fluctuated by about $\pm 10$ mm yr$^{-1}$, in particular due to La Niña events, but do not exhibit a significant long-term trend (Vose et al. 2023). Under the common assumption that global precipitation is balanced against evaporation, no substantial strengthening of the hydrological cycle may be observed yet.

Probably, the minor trend of 0.06 mm yr$^{-2}$ of the data displayed in Fig. 7 is statistically insignificant. Associated with this apparent trend, the latent heat transferred to the troposphere can be estimated to a negligible putative warming rate of additional 0.5 mW m$^{-2}$ per year, which could explain only 10 % of observed atmospheric warming by 1.7 °C per century (Morice et al. 2012, Feistel and Hellmuth 2021).

The thermodynamic driving force for evaporation is the difference between the chemical potentials of water in humid air and in seawater at the two sides of the sea-air interface (Kraus and Businger 1994). TEOS-10 has made this difference numerically available in the form of the water mass evaporation rate (Feistel and Hellmuth 2022, 2023)

$$J_{\mathrm{W}} = -D_f(u) \ln \frac{\psi_f}{x_{\mathrm{W}}}. \tag{6}$$

Here, $x_{\mathrm{W}}$ is the mole fraction of water in seawater. Consistent with Wüst (1920), for the standard ocean with Reference Composition, this fraction is (Millero et al. 2008: Table 4),

$$x_{\mathrm{W}} = \frac{53.556\,514\,4}{54.676\,283\,8} = 0.979\,52, \qquad \ln x_{\mathrm{W}} = -0.0206926. \tag{7}$$

In eq. (6), $D_f(u)$, the *Dalton coefficient*, is an empirical transfer coefficient as a function of the wind
speed, $u$, as a parameterisation of the turbulent transport processes of water in the vicinity of the
interface. Applications of the *Monin-Obukhov Similarity Theory* (MOST) in order to estimate the
Dalton coefficient are reviewed by Liu et al. (1979), Foken and Richter (1991), Foken (2004, 2016) and
in the Digital Supplement of Feistel and Hellmuth (2024). A review of empirical Dalton coefficients is
given by Debski (1966); historical evaporation experiments are summarised by Biswas (1969).
In eq. (6), the sea-surface humidity is expressed by the *relative fugacity* (RF), $\psi_f$, defined by the ratio
of the water-vapour fugacity in humid air, $f_V$, to that fugacity at saturation, $f_V^{\mathrm{sat}}$ (Feistel and Lovell-
Smith 2017), see eq. (49). In ideal-gas approximation, RF equals conventional RH (Lovell-Smith et al.
452    2016)

$$\psi_f \equiv \frac{f_V}{f_V^{\mathrm{sat}}} \approx \psi_x \equiv \frac{x}{x^{\mathrm{sat}}}.$$    (8)
Here, the mole fraction of water vapour in humid air is $x$, and its value at saturation is $x^{\mathrm{sat}}$. Further,
$\psi_x$ is the conventional definition of RF in metrology and meteorology which, however, is inconsistent
with alternative definitions such as the one employed in climatology (Lovell-Smith et al. 2016).
Independent of ideal-gas conditions, but sufficiently close to saturation, such as near the sea surface,
RF can be estimated in excellent approximation from the Clausius-Clapeyron formula (Feistel et al.
459    2022),

$$\psi_f \approx \exp\left\{\frac{L(T_{\mathrm{dp}},p)}{R_W}\left(\frac{1}{T} - \frac{1}{T_{\mathrm{dp}}}\right)\right\}.$$    (9)
The evaporation enthalpy of pure water (IAPWS SR1-86 1992) at the dewpoint $T_{\mathrm{dp}}$ is $L$, and $R_W =$
$461.523\ \mathrm{J\ kg^{-1}\ K^{-1}}$ is the specific gas constant of water. The typical marine RF is
$$\psi_f \approx 80\ \%\mathrm{rh},$$    (10)
and is fairly independent of region, season or global warming (Dai 2006, Randall 2012, Rapp 2014,
MetOffice 2020). Indeed, observed ocean surface RH has no significant climatological trend (Willett
et al. 2023). Similarly, observed ocean wind speeds seem to be unaffected by global warming (Azorin-
Molina et al. 2023). Eq. (6) for the evaporation rate depends only on wind speed and RF, so that it
may be concluded that also the global mean evaporation rate has no significant climatic trend. In
turn, as far as the release of latent heat is the main driving force of marine tropospheric dynamics,
without increase of that release the mean wind speed is not expected to grow. "Latent heat is the
main fuel that powers hurricanes, thunderstorms and normal bouts of lousy weather" (Francis 2021).
Hence, the TEOS-10 approach in the form of eq. (6) appears to be consistent with the prediction of
Held and Soden (2006) that the global evaporation does not increase along with temperature.
Various empirical evaporation equations, commonly known as *Dalton equations*, are found in the
literature (Wüst 1920, Sverdrup 1936, 1937, Montgomery 1940, Debski 1966, Biswas 1969,
Baumgartner and Reichel 1975). Several numerical climate models estimate evaporation from the
formula (Stewart 2008, Pinker et al. 2014),
$$J_W = D_q(u)(q_0 - q_{10}),$$    (11)
where $q_0$ is the specific humidity at the sea surface and $q_{10}$ is that at 10 m height, or from (Josey et
al. 1999, 2013)
$$J_W = D_q(u)(0.98\ q^{\mathrm{sat}} - q).$$    (12)
Here, $q$ is the near-surface specific humidity, and $q^{\mathrm{sat}}$ is the saturation value at the same
temperature and pressure. The factor 0.98 accounts for the salinity, see eq. (7). After a few
approximation steps (Feistel and Hellmuth 2023), these Dalton equations can be derived from the
TEOS-version, eq. (6), however, there is an important qualitative difference. At constant RH, due to
global warming, specific humidities such as $q$ and $q^{\mathrm{sat}}$, as well as their difference, are increasing
following the Clausius-Clapeyron saturation formula. Accordingly, eq. (12) implies that also the
evaporation rate $J_{\mathrm{W}}$ is growing this way, by contrast to eq. (6). This virtual acceleration of the
hydrological cycle is evidently inconsistent with the prediction of Held and Soden (2006). This
parameterisation-caused additional latent heat flux implies a spurious ocean cooling that may
contribute to the finding that many numerical climate models tend to underestimate the observed
ocean warming (Weller et al. 2022).
From eq. (6), the sensitivity of the latent heat flux, $LJ_{\mathrm{W}}$, with respect to RH variations is easily
estimated. For a mean evaporation rate of 1200 mm per year, the corresponding mass flux is about
$J_{\mathrm{W}} \approx 3.8 \times 10^{-5}\ \mathrm{kg\ m^{-2} s^{-1}}$ and the related heat flux is $LJ_{\mathrm{W}} \approx 95\ \mathrm{W\ m^{-2}}$ with respect to the ocean
surface area and a specific evaporation enthalpy of $L = 2\,501\ \mathrm{kJ\ kg^{-1}}$. At a surface humidity of $\psi_f =$
0.8, a value of $D_f(u) \approx 1.87 \times 10^{-4}\ \mathrm{kg\ m^{-2} s^{-1}}$ can be concluded for the mass transfer coefficient,
and of $LD_f(u) \approx 468\ \mathrm{W\ m^{-2}}$ for that of latent heat. Then, from
$$\Delta(LJ_{\mathrm{W}}) = L\frac{\partial J_{\mathrm{W}}}{\partial \psi_f}\Delta\psi_f = -LD_f(u)\frac{\Delta\psi_f}{\psi_f} \qquad\qquad (13)$$
it follows that an increase by $\Delta\psi_f = 1\ \%\mathrm{rh}$ results in a heat flux reduction by $\Delta(LJ_{\mathrm{W}}) =$
$5.85\ \mathrm{W\ m^{-2}}$. So, the currently observed ocean warming (Cheng et al. 2024) of $1.3\ \mathrm{W\ m^{-2}}$ could
theoretically be caused already by a minor marine humidity increase of $\Delta\psi_f = 0.2\ \%\mathrm{rh}$, a value far
below the present measurement uncertainty between 1 and 5 %rh of relative humidity. The
resolution of climate models and observation seems to be insufficient yet to identify the possible role
of RH for the unclear explanation of the warming ocean.

**5   Sea Air as a Two-Phase Composite**
Gibbs' (1873) method of using potential functions can be applied to any systems possessing stable
thermodynamic equilibria and obeying energy conservation, without being restricted to merely
homogeneous or single-phase samples. The intentionally strict mutual consistency of the different
TEOS-10 potential functions permits a mathematical description of multi-phase composites such as
sea ice, consisting of ice with included brine pockets (Feistel and Hagen 1998, Feistel and Wagner
2005), or clouds, where liquid water or ice is floating in saturated humid air (Hellmuth et al. 2021).
Another important model is that of *sea air*, a sample consisting of a mass $m^{\mathrm{SW}}$ of seawater in
thermodynamic equilibrium with a mass $m^{\mathrm{AV}}$ of humid air (Feistel et al. 2010d, Feistel and Hellmuth
2023). Such a model may serve as a mathematical description for certain thermodynamic properties
of ocean-atmosphere interaction.
Extensive thermodynamic functions such as Gibbs energy or enthalpy are additive with respect to the
two separate phases of the sample. Equilibrium between those parts requires equal temperatures
and pressures. For this reason, a Gibbs function of sea air is an appropriate potential for the
composite system with the TEOS-10 Gibbs functions $g^{\mathrm{SW}}(S,T,p)$ describing the liquid part and
$g^{\mathrm{AV}}(A,T,p)$ the gas part. Let the masses of the substances in the parts be $m^{\mathrm{W}}$ of liquid water, $m^{\mathrm{S}}$ of
dissolved salt, $m^{\mathrm{A}}$ of dry air and $m^{\mathrm{V}}$ of water vapour. Note that TEOS-10 neglects solubility of dry air
constituents in liquid water. From combinations of the partial masses follow the liquid mass, $m^{\mathrm{SW}} =$
$m^{\mathrm{S}} + m^{\mathrm{W}}$, the gas mass, $m^{\mathrm{AV}} = m^{\mathrm{A}} + m^{\mathrm{V}}$, the total mass $m = m^{\mathrm{SW}} + m^{\mathrm{AV}}$, the total water mass
$m^{\mathrm{WV}} = m^{\mathrm{W}} + m^{\mathrm{V}}$, the salinity $S = m^{\mathrm{S}}/m^{\mathrm{SW}}$ and the dry-air fraction $A = 1 - q = m^{\mathrm{A}}/m^{\mathrm{AV}}$.
The Gibbs energies of the two phases of sea air are additive,
$G^{\mathrm{SA}} = G^{\mathrm{SW}} + G^{\mathrm{AV}} = mg^{\mathrm{SA}}$ ,                                   (14)
and, accordingly, the Gibbs function of sea air, $g^{\mathrm{SA}}$, may be constructed from that of seawater,
$g^{\mathrm{SW}}(S,T,p)$, with a liquid mass fraction of $w^{\mathrm{SW}} = m^{\mathrm{SW}}/m$ and that of humid air, $g^{\mathrm{AV}}(A,T,p)$, with
a gaseous mass fraction of $w^{\mathrm{AV}} = m^{\mathrm{AV}}/m = 1 - w^{\mathrm{SW}}$,
$g^{\mathrm{SA}}(S,A,w^{\mathrm{SW}},T,p) = w^{\mathrm{SW}}g^{\mathrm{SW}}(S,T,p) + (1 - w^{\mathrm{SW}})g^{\mathrm{AV}}(A,T,p)$.          (15)
If the two phases are assumed to be at mutual equilibrium, they possess the same temperature,
pressure and chemical potentials, see eq. (B.11) in Appendix B, $\mu_{\mathrm{W}}^{\mathrm{SW}} = \mu_{\mathrm{V}}^{\mathrm{AV}}$, namely that of water in
seawater,
$\mu_{\mathrm{W}}^{\mathrm{SW}}(S,T,p) = g^{\mathrm{SW}} - S\left(\dfrac{\partial g^{\mathrm{SW}}}{\partial S}\right)_{T,p}$,                 (15)
equalling that of water vapour in humid air,
$\mu_{\mathrm{V}}^{\mathrm{AV}}(A,T,p) = g^{\mathrm{AV}} - A\left(\dfrac{\partial g^{\mathrm{AV}}}{\partial A}\right)_{T,p}$.                 (16)

*5.1 Sea Air as a Model for Latent Heat of Evaporation*
Water evaporated from the ocean surface drives the climate system. "The by far largest part of heat
conveyed to the air is in the form of latent heat during subsequent condensation along with cloud
formation. The heat budget over the sea is mainly controlled by the latent heat released to the air"
(Albrecht 1940). It is the "*heat source for a gigantic steam engine*", as Heinrich Hertz had put it in his
1885 inaugural lecture at Karlsruhe (Mulligan and Hertz 1997). The latent heat of evaporation of pure
liquid water into pure water vapour is numerically well known from experiments (IAPWS SR1-86
1992, Harvey 1998, Wagner and Pruß 2002). Slightly differing values are reported in various
textbooks on hydrology (Debski 1966: p. 332), meteorology (Linke and Baur 1970) or geophysics (Gill
1982, Kraus and Businger 1994). TEOS-10, however, permits the computation of evaporation
properties from seawater into humid air, based on the first-time availability of standard equations
for enthalpies and chemical potentials of those non-ideal mixtures.

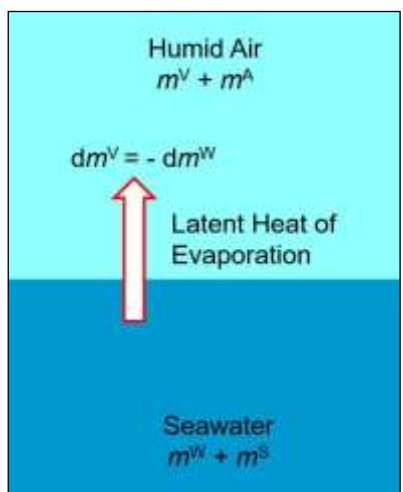


Fig. 8: Conceptual thermodynamic "sea air" model of ocean-atmosphere interaction as a two-phase
composite of seawater and humid air

"Latent heat is the quantity of heat which must be communicated to a body in a given state in order
to convert it into another state without changing its temperature" (Maxwell 1888: p.73). If an
infinitesimal amount of water is transferred from the liquid to the gas phase (Fig. 8), while
temperature and pressure remain at their equilibrium values, and the total masses of salt, $m^S$, dry
air, $m^A$, and water, $m^{WV}$, are not affected, the isobaric-isothermal latent heat of evaporation may be
defined by
$$L^{SA} \equiv \left(\frac{\partial H^{SA}}{\partial m^V}\right)_{T,p,m^S,m^A,m^{WV}}. \tag{17}$$

This latent heat accounts for the loss of total heat of the sea-air sample associated with the loss of
liquid water and equal gain of water vapour,
$$\frac{\partial m^V}{\partial T} = -\frac{\partial m^W}{\partial T}. \tag{18}$$

Here, $H^{SA}$ is the enthalpy of sea air, available from the Gibbs function (15) through the sum
$$H^{SA} \equiv m^{SW}h^{SW} + m^{AV}h^{AV}. \tag{19}$$

Here, the specific enthalpies of seawater,
$$h^{SW} = g^{SW} - T\left(\frac{\partial g^{SW}}{\partial T}\right)_{S,p}, \tag{20}$$

and of humid air,
$$h^{AV} = g^{AV} - T\left(\frac{\partial g^{AV}}{\partial T}\right)_{A,p}, \tag{21}$$

are defined in terms of the related Gibbs functions.

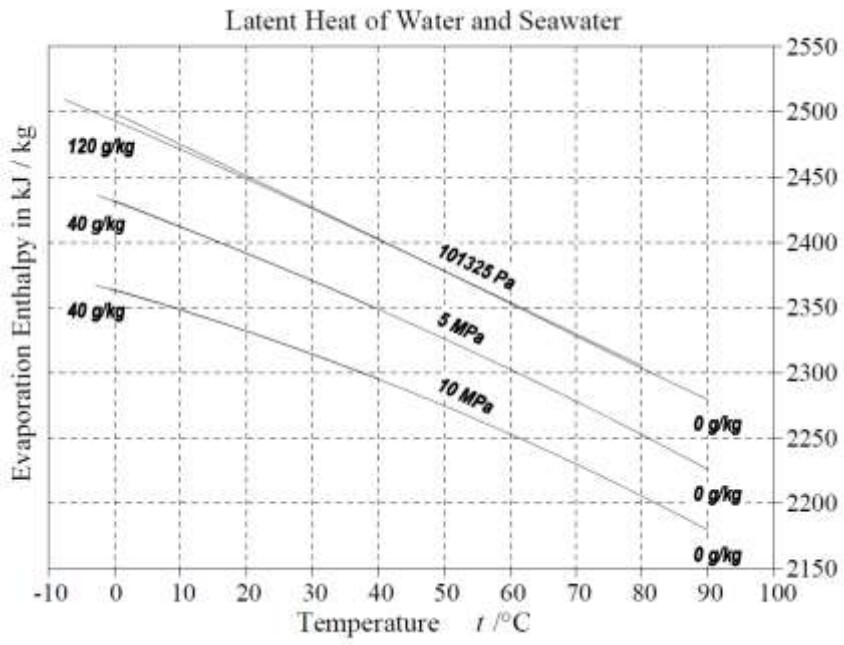


Fig. 9: Evaporation enthalpy, eq. (23), of seawater in equilibrium with humid air at different
temperatures, pressures and salinities. The dependence on salinity is very weak; graphically, the
related curves are hardly distinguishable. The nonlinear dependence on temperature is more
pronounced at elevated pressures. Figure from Feistel et al. (2010a: p. 105)

The derivative (17) is carried out in the form
$L^{\mathrm{SA}} = -h^{\mathrm{SW}} - m^{\mathrm{SW}}\left(\frac{\partial h^{\mathrm{SW}}}{\partial S}\right)_{T,p}\left(\frac{\partial S}{\partial m^{\mathrm{V}}}\right)_{m^{\mathrm{S}},m^{\mathrm{WV}}} + h^{\mathrm{AV}} + m^{\mathrm{AV}}\left(\frac{\partial h^{\mathrm{AV}}}{\partial A}\right)_{T,p}\left(\frac{\partial A}{\partial m^{\mathrm{V}}}\right)_{m^{\mathrm{A}}},$     (22)
which results in the TEOS-10 latent-heat equation (Feistel et al. 2010a, Feistel and Hellmuth 2023),
$L^{\mathrm{SA}} = h^{\mathrm{AV}} - A\left(\frac{\partial h^{\mathrm{AV}}}{\partial A}\right)_{T,p} - h^{\mathrm{SW}} + S\left(\frac{\partial h^{\mathrm{SW}}}{\partial S}\right)_{T,p},$     (23)
with typical values shown in Fig. 9. If seawater is in mutual equilibrium with humid air at given
temperature and pressure, salinity and humidity of the parts of sea air satisfy the condition $\mu_{\mathrm{W}}^{\mathrm{SW}} =$
$\mu_{\mathrm{V}}^{\mathrm{AV}}$, given by eqs. (15) and (16),
$\Delta\mu \equiv g^{\mathrm{SW}} - S\left(\frac{\partial g^{\mathrm{SW}}}{\partial S}\right)_{T,p} - g^{\mathrm{AV}} + A\left(\frac{\partial g^{\mathrm{AV}}}{\partial A}\right)_{T,p} = 0$     (24)
At given masses of salt, $m^{\mathrm{S}}$, of dry air, $m^{\mathrm{A}}$, and of total water, $m^{\mathrm{WV}} = m^{\mathrm{W}} + m^{\mathrm{V}}$, eq. (24) controls
the value of either $m^{\mathrm{W}}$ or $m^{\mathrm{V}}$, and this way also of $S$ and $A$ as functions of $T, p, m^{\mathrm{S}}, m^{\mathrm{A}}$ and $m^{\mathrm{WV}}$.
Related numerical solutions are readily implemented in the TEOS-10 SIA library; the latent heat of
sea air can be computed by calling the function sea_air_enthalpy_evap_si(), see Wright et al.
591    (2010).

Latent heat of eq. (23) is valid regardless of the equilibrium condition, eq. (24), being satisfied or not.
The non-equilibrium case is considered separately in Section 5.3.

*5.2 Sea Air as a Model of Sea Spray*
As a special form of air-sea interaction, sea spray is typically ejected from the crest of a breaking
wave, which may happen all along oceanic coasts but also wherever whitecaps are produced from
swell or stormy sea state, see Fig. 10. In contrast to fresh-water haze, droplets of sea spray cannot
completely evaporate for the salt they contain, and rather develop into a floating persistent Köhler
(1936) equilibrium between droplet size, droplet salinity and ambient relative fugacity (Hellmuth and
Shchekin 2015, Pöhlker et al. 2023). This equilibrium can be described by the TEOS-10 model of sea
air if the additional Kelvin pressure caused by the surface tension is allowed for.
In the infrared spectral range, sea spray as well as other aerosols (Carlon 1970, 1980) may be
considered as a black absorber and emitter of thermal radiation. The resulting "gray atmosphere" is a
conveniently simple conceptual model for the long-wave radiative effects of dust or haze in the
climate system (Emden 1913). When heated from below, as in the case of the clear-sky marine
troposphere, a theoretical finding is that the thermally stratified gray troposphere exhibits a special
critical value of the isobaric heat capacity at $c_p = 4R$ (Pierrehumbert 2010: p. 201), $R$ being the
molar gas constant. Vertical stability may be lost at $c_p > 4R$ and turbulent mixing is expected to
commence (Feistel 2011b: eq. 58 therein). Such a kinetic phase transition could substantially modify
the thermal radiation balance between troposphere and ocean surface.

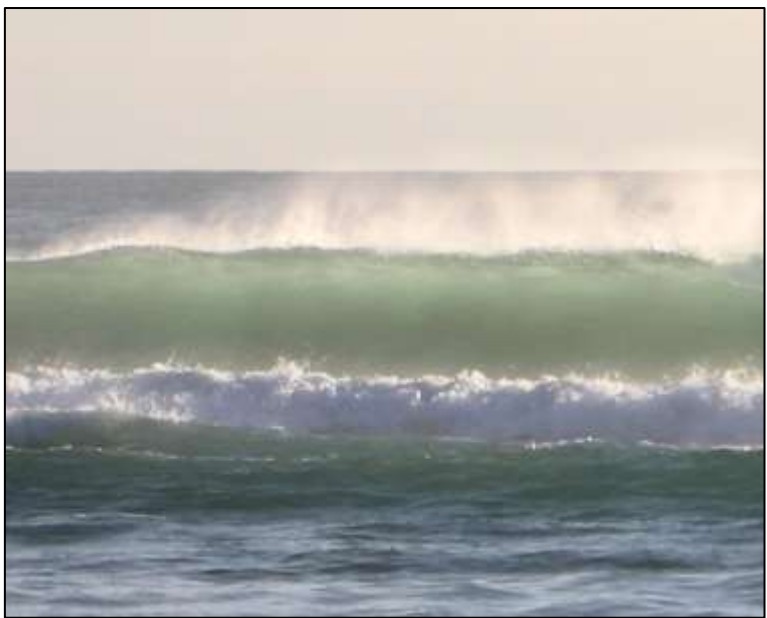


Fig. 10: Sea spray ejection from a breaking wave crest of Atlantic swell. Photo taken at Cabo Trafalgar
in March 2011.

The terrestrial atmosphere is dominated by the two-atomic gases $N_2$ and $O_2$ with heat capacities
about $3.5\,R$ which prevent the putative radiative vertical instability to occur. This situation may
change, however, in the presence of haze or sea spray. To investigate this effect theoretically, in this
section a TEOS-10 equation for the heat capacity of equilibrium sea air is derived from the definition
$$c_p^{\mathrm{SA}} \equiv \frac{1}{m}\left(\frac{\partial H^{\mathrm{SA}}}{\partial T}\right)_{p,m^{\mathrm{S}},m^{\mathrm{A}},m^{\mathrm{WV}}}. \qquad (25)$$
The enthalpy of sea air is given by eq. (19). Taking into account water conservation upon
evaporation, $m^{\mathrm{WV}} = \mathrm{const}$, that is,
$$\frac{\partial m^{\mathrm{V}}}{\partial T} = -\frac{\partial m^{\mathrm{W}}}{\partial T}, \qquad (26)$$
and of eq. (23), the isobaric heat capacity of sea air is concluded to be
$$c_p^{\mathrm{SA}} = w^{\mathrm{SW}} c_p^{\mathrm{SW}} + w^{\mathrm{AV}} c_p^{\mathrm{AV}} + L^{\mathrm{SA}} \frac{1}{m}\frac{\partial m^{\mathrm{V}}}{\partial T}. \qquad (27)$$
To the additive contributions of the partial heat capacities of the liquid and the gas part, there
appears the latent heat of the water mass that evaporates from the liquid as vapour. This
evaporation rate is governed by the mutual equilibrium between seawater and humid air.
During the temperature change, sea-air equilibrium, eq. (24), is assumed to be maintained by water
transfer between the phases, changing $S$ and $A$ along with $T$,
$$\left(\frac{\partial \Delta\mu}{\partial T}\right)_{p,m^{\mathrm{S}},m^{\mathrm{A}},m^{\mathrm{WV}}} = 0. \qquad (28)$$
Carrying out the derivative, this condition reads
$$\left(\frac{\partial g^{\mathrm{SW}}}{\partial T}\right)_{S,p} - S\left(\frac{\partial^2 g^{\mathrm{SW}}}{\partial S \partial T}\right)_p - S\left(\frac{\partial^2 g^{\mathrm{SW}}}{\partial S^2}\right)_{T,p}\left(\frac{\partial S}{\partial T}\right)_{m^{\mathrm{S}}}$$
$$= \left(\frac{\partial g^{AV}}{\partial T}\right)_{A,p} - A\left(\frac{\partial^2 g^{AV}}{\partial A\partial T}\right)_p - A\left(\frac{\partial^2 g^{AV}}{\partial A^2}\right)_{T,p}\left(\frac{\partial A}{\partial T}\right)_{m^A}. \tag{29}$$
On the other hand, from combining eq. (23) with eq. (24) it follows that the latent heat may be
expressed by,
$$L^{SA} = T\left\{\left(\frac{\partial g^{SW}}{\partial T}\right)_{S,p} - S\left(\frac{\partial^2 g^{SW}}{\partial S\partial T}\right)_p - \left(\frac{\partial g^{AV}}{\partial T}\right)_{A,p} + A\left(\frac{\partial^2 g^{AV}}{\partial A\partial T}\right)_p\right\}, \tag{30}$$
so that eq. (29) may be written as
$$L^{SA} = T\left\{S\left(\frac{\partial^2 g^{SW}}{\partial S^2}\right)_{T,p}\left(\frac{\partial S}{\partial T}\right)_{m^S} - A\left(\frac{\partial^2 g^{AV}}{\partial A^2}\right)_{T,p}\left(\frac{\partial A}{\partial T}\right)_{m^A}\right\}. \tag{31}$$
Further, the total water mass balance, eq. (26), implies that
$$\left(\frac{\partial S}{\partial T}\right)_{m^S} = \left(\frac{\partial S}{\partial m^W}\right)_{m^S}\frac{\partial m^W}{\partial T} = \frac{S}{m^{SW}}\frac{\partial m^V}{\partial T}, \tag{32}$$
and similarly,
$$\left(\frac{\partial A}{\partial T}\right)_{m^A} = \left(\frac{\partial A}{\partial m^V}\right)_{m^A}\frac{\partial m^V}{\partial T} = -\frac{A}{m^{AV}}\frac{\partial m^V}{\partial T}. \tag{33}$$
Inserting those expressions into eq. (31), the equation for the isobaric evaporation rate of sea air is
$$\frac{\partial m^V}{\partial T} = \frac{L^{evap}}{T}\left\{\frac{S^2}{m^{SW}}\left(\frac{\partial^2 g^{SW}}{\partial S^2}\right)_{T,p} + \frac{A^2}{m^{AV}}\left(\frac{\partial^2 g^{AV}}{\partial A^2}\right)_{T,p}\right\}^{-1}. \tag{34}$$

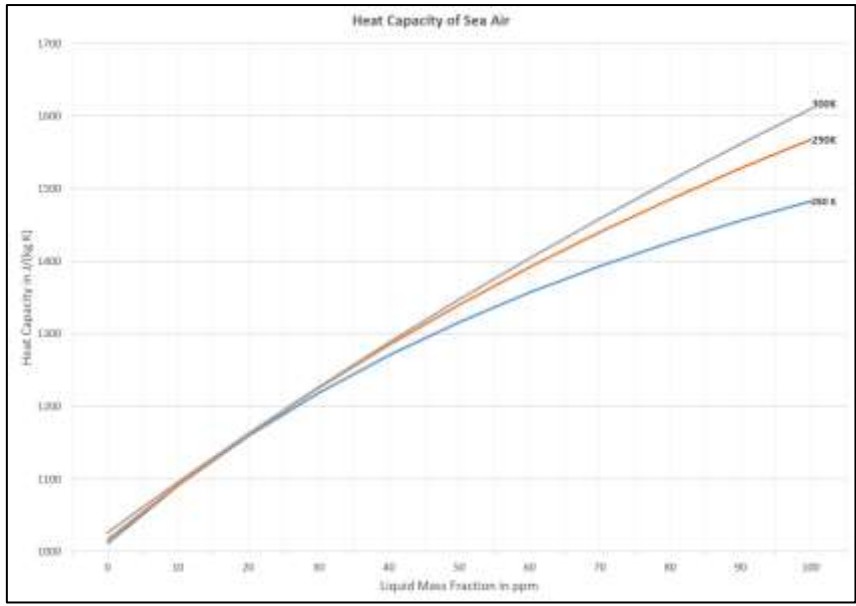


Fig. 11: TEOS-10 values for the isobaric specific heat capacity, eq. (35), of sea air at atmospheric
pressure and sea-spray standard-ocean salinity, $S$ = 35.165 04 g kg$^{-1}$, at temperatures of 280 K (lower
curve), 290 K (middle curve) and 300 K (upper curve) as functions of the liquid mass fraction, $w^{SW}$,
up to 100 ppm.

Together with eq. (34), the desired formula for the isobaric heat capacity (27) of sea air finally
becomes (Feistel et al. 2010a: eq. 6.22 therein),
$$c_p^{\mathrm{SA}} = w^{\mathrm{SW}} c_p^{\mathrm{SW}} + \left(1 - w^{\mathrm{SW}}\right) c_p^{\mathrm{AV}} + \frac{\left(L^{\mathrm{SA}}\right)^2}{T} \left\{ \frac{S^2}{w^{\mathrm{SW}}} \left(\frac{\partial^2 g^{\mathrm{SW}}}{\partial S^2}\right)_{T,p} + \frac{A^2}{\left(1 - w^{\mathrm{SW}}\right)} \left(\frac{\partial^2 g^{\mathrm{AV}}}{\partial A^2}\right)_{T,p} \right\}^{-1}. \quad (35)$$
Of the *liquid water content*, expressed in form of the liquid mass fraction, $w^{\mathrm{SW}}$, realistic values may
typically range between $10^{-6}$ and $10^{-4}$ in the troposphere. Growing along with this fraction, related
heat capacities of sea air, eq. (35), may substantially exceed that of liquid-free humid air, $c_p^{\mathrm{AV}}$, see
Fig. 11.

*5.3 Sea Air as a Model for Irreversible Evaporation*
The climate system functions far from thermodynamic equilibrium, permanently producing and
exporting entropy at an average rate about 1 W m$^{-2}$ K$^{-1}$ per global surface area (Ebeling and Feistel
1982, Feistel and Ebeling 2011). By contrast, TEOS-10 is a mathematical description of equilibrium
properties (Appendix B). The latter is applicable to states away from thermodynamic equilibrium
under the assumption of *local equilibrium* as introduced by Ilya Prigogine (1947, 1978). This
assumption means that spatially extended substances such as ocean or atmosphere consist of
sufficiently small volume elements that may reasonably be described as macroscopic equilibrium
states, homogeneous in temperature, pressure and chemical potentials. TEOS-10 thermodynamic
potentials can be used to describe those local states.
By definition, if a volume at equilibrium is divided into partial volumes, each of those parts is at
equilibrium itself, and each pair of those is at mutual equilibrium also. By contrast, the combination
of several local-equilibrium elements forms a non-equilibrium state if pairs of elements exist that are
out of mutual equilibrium. Extensive properties such as mass, energy, entropy or enthalpy can be
added up to give correct values of the entire system. When exchange processes between those
elements occur, gains and losses of masses, energies or enthalpies are mutually balanced by
conservation laws, however, this is not the case for entropy.
A tutorial case of a local equilibrium system may be the model of sea air (Feistel and Hellmuth 2024a)
depicted in Fig. 8. It consists of a mass $m^{\mathrm{SW}} = m^{\mathrm{S}} + m^{\mathrm{W}}$ of seawater in contact with a mass $m^{\mathrm{AV}} =$
$m^{\mathrm{A}} + m^{\mathrm{A}}$ of humid air. Both fluids are assumed to be at internal equilibrium themselves but not
necessarily in mutual equilibrium with one another. This is a natural geophysical situation – marine
RH has typical values of 80 %rh while the equilibrium of humid air with seawater, eq. (24), is
established at about 98 %rh. For simplicity, let all parts have equal temperatures and pressures.
If evaporation takes place, the partial water masses involved will change by a mass flux across the
sea surface,
$$J_m \equiv \frac{\mathrm{d}m^{\mathrm{AV}}}{\mathrm{d}t} = \frac{\mathrm{d}m^{\mathrm{V}}}{\mathrm{d}t} = -\frac{\mathrm{d}m^{\mathrm{SW}}}{\mathrm{d}t} = -\frac{\mathrm{d}m^{\mathrm{W}}}{\mathrm{d}t}. \quad (36)$$
The change of the total enthalpy of the sea-air sample is available from eqs. (17) and (23),
$$\frac{\mathrm{d}H^{\mathrm{SA}}}{\mathrm{d}t} = \left(\frac{\partial H^{\mathrm{SA}}}{\partial m^{\mathrm{V}}}\right)_{T,p,m^{\mathrm{S}},m^{\mathrm{A}},m^{\mathrm{WV}}} \frac{\mathrm{d}m^{\mathrm{V}}}{\mathrm{d}t} = L^{\mathrm{SA}} J_m. \quad (37)$$
This expression of energy conservation, the 1$^{\mathrm{st}}$ law of thermodynamics, is similarly valid for
equilibrium and non-equilibrium conditions of the sample. For comparison, of the total entropy
defined by,
$$N^{\mathrm{SA}} \equiv m^{\mathrm{SW}} \eta^{\mathrm{SW}} + m^{\mathrm{AV}} \eta^{\mathrm{AV}}, \quad (38)$$
the change is given by
$$\frac{dN^{SA}}{dt} = \left(\frac{\partial N^{SA}}{\partial m^V}\right)_{T,p,m^S,m^A,m^{WV}} \frac{dm^V}{dt}.$$  (39)
In terms of its two parts, eq. (38), this change takes the form,
$$\frac{dN^{SA}}{dt} = \left[\eta^{AV} - A\left(\frac{\partial \eta^{AV}}{\partial A}\right)_{T,p} - \eta^{SW} + S\left(\frac{\partial \eta^{SW}}{\partial S}\right)_{T,p}\right]J_m.$$  (40)
In oceanography, the symbol $N$ for entropy was suggested by Fofonoff (1962) to avoid confusion
with salinity $S$. Making use of their local equilibria, specific entropy of each part can be expressed by
the difference, eq. (B.6),
$$\eta = \frac{h-g}{T},$$  (41)
between specific enthalpy, $h$, and specific Gibbs energy, $g$, so that the entropy change (40) becomes
$$T\frac{dN^{SA}}{dt} = \left(L^{SA} + \Delta\mu\right)J_m.$$  (42)
Here, the latent heat, $L^{SA}$, is given by eq. (23), and the distance from mutual equilibrium, $\Delta\mu$, by eq.
703  (24).

The first term,
$$T\frac{d_e N^{SA}}{dt} \equiv L^{SA}J_m,$$  (43)
is the *external* entropy change (subscript e) in the form of the heat flux required to maintain the
sample's temperature, in the sense of Maxwell's (1888) definition of latent heat, compensating the
storage of latent heat by emitting water vapour.
The second term,
$$T\frac{d_i N^{SA}}{dt} \equiv J_m\Delta\mu.$$  (44)
is the *internal* entropy change (subscript i), or *entropy production*, of the non-equilibrium sea-air
sample. It represents the additional entropy gain of humid air compared to the entropy loss of
seawater. This production happens at the air-sea interface and disappears as soon as mutual
equilibrium, $\Delta\mu = 0$, is approached.
It is important to be aware that the external part, $\frac{d_e N^{SA}}{dt}$, *always* constitutes a contribution to the
system's energy balance while, by contrast, the internal part, $\frac{d_i N^{SA}}{dt}$, is *never* any such contribution.
The irreversible production of entropy is an internal conversion or redistribution of energy rather
than a change of it. This implies that irreversible processes violate Gibbs' fundamental equation (B.8)
in the sense that
$$\frac{dH^{SA}}{dt} = -T\frac{d_e N^{SA}}{dt} + V^{SA}\frac{dp}{dt} + \sum_i \mu_i \frac{dm_i}{dt} > -T\frac{dN^{SA}}{dt} + V^{SA}\frac{dp}{dt} + \sum_i \mu_i \frac{dm_i}{dt},$$  (45)
even though each of its local-equilibrium elements strictly satisfies the related fundamental equation
(B.13), valid for reversible processes only,
$$dh = -Td\eta + vdp + \sum_{i=1}^{n-1}(\mu_i - \mu_0)dw_i.$$  (46)
Entropy production appears wherever a flux is passing its driving gradient. Near equilibrium, this flux
is proportional to its driving force (Glansdorff and Prigogine 1971, Landau and Lifschitz 1974, Kraus
and Businger 1994, Feistel and Hellmuth 2024a), usually termed *Onsager force*. For example, the
evaporation mass flux of water, eq. (6),
$$J_m = C\,\Delta\mu \tag{47}$$
may be assumed as being proportional to the difference between the chemical potentials of water
across the air-sea interface. The related *Dalton equation* (6) was discussed in Section 4. The
associated entropy production, eq. (44), obeys the 2$^{nd}$ law of thermodynamics by the inequality
$$\frac{d_i N^{SA}}{dt} = C\,(\Delta\mu)^2 \geq 0, \tag{48}$$
while the total entropy change, eq. (42) may possess any sign. In other words, the 2$^{nd}$ law forbids that
*Onsager fluxes* may be directed against their causing Onsager forces. The *Prigogine Theorem* predicts
that in linear irreversible thermodynamics, entropy production approaches minimum values at
steady states (Glansdorff and Prigogine 1971).
Processes accompanied by entropy production are termed *irreversible* ones, since entropy once
created may never be destroyed again. Related processes cannot be reversed unless lasting changes
are left behind in the external world. By contrast, processes which transform an equilibrium state
into another equilibrium state may *reversibly* be performed without producing entropy. Entropy
production is possible only under non-equilibrium conditions.
Under typical marine circumstances, the entropy production density of ocean evaporation can be
estimated to about 4 mW K$^{-1}$ m$^{-2}$, contributing roughly 0.4 % to the global entropy production
(Feistel and Ebeling 2011, Feistel and Hellmuth 2024a).


**6  Cloudiness and Ocean Warming**
"Cloud feedback on climate represents the largest uncertainty in our ability to understand the
sensitivity of the planet to radiative forcing" (Gettelman and Sherwood 2016). On the long-term
average, cloudiness is particularly strong in the low-pressure belts of the global tropospheric
circulation, where air is ascending and its humidity is condensing, see Fig. 12. Except for the
equatorial zone, those spatial cloudiness pattern correlate visibly with those of recent ocean
warming, compare Fig. 1. It is a plausible working hypothesis that this correlation could also indicate
a causal relation between the two phenomena. However, such correlations imply chicken-and-egg
problems (Rapp 2014): putative causality relations between those trends cannot be derived from
observation but only be concluded from reliable prediction models (Feistel 2023). May the observed
systematic reduction of global cloudiness (Fasullo and Trenberth 2012) actually be responsible for
the currently recorded excessive ocean warming (You 2024)? Unfortunately, and somewhat
surprisingly, this assumption can apparently not be underpinned yet by closer investigation. Some
related issues will be discussed in this section.

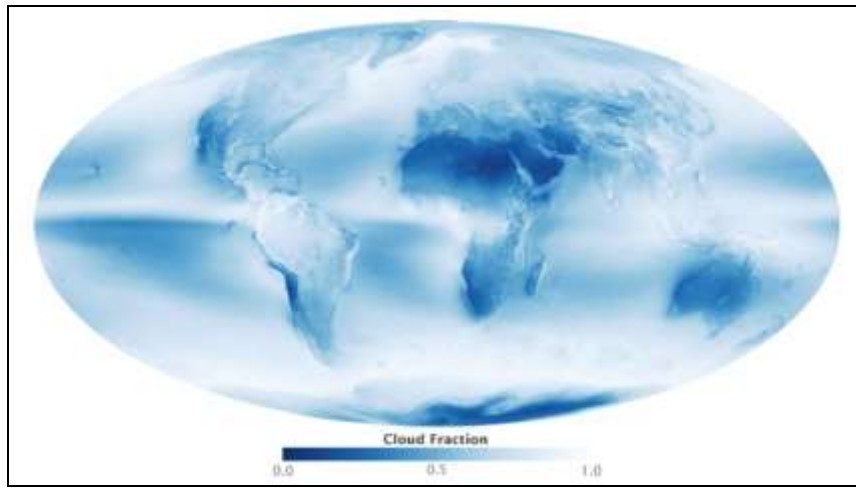

Fig. 12: Global distribution of cloudiness July 2002 – April 2015 (Allen and Ward 2015). Image reproduction permitted by NASA Copyright.

### 6.1 Cloudiness Trend

Global cloud-covered surface area fraction $C$ has systematically been reducing by about 6 % per century, see Fig. 13, from $C \approx 67.5$ % in 1980 to $C \approx 65$ % in 2022 (Foster et al. 2023, Phillips and Foster 2023). Observed cloudiness values depend strongly on the way clouds are defined (Spänkuch et al. 2022) and on the measurement technology applied. For example, Rapp (2014: Fig. 6.20) reported a decrease in cloudiness in 30 years from 70 % in 1983 down to 63.5 % in, likely, 2013. This reduction rate of more than 20 % per century is three times as fast as that given in Fig. 13 and may result from different observation techniques.

Assuming that this shrinking occurred in a similar way above both land and sea, the ocean is expected to receive increasingly more solar irradiation. This phenomenon is known as the *short-wave cloud radiative effect* (SW CRE), see Fig. 14.

On the other hand, clouds are opaque with respect to oceanic upward thermal radiation and emit themselves downward infrared radiation. This phenomenon is known as the *long-wave cloud radiative effect* (LW CRE), see Fig. 15. Radiation models show that on the global average these two effects cancel each other almost completely up to minor residual of $-1$ mW m$^{-2}$ yr$^{-1}$, so that the continuously shrinking cloudiness may be assumed to have practically no net effect on the ocean's radiation balance (Phillips and Foster 2023, Feistel and Hellmuth 2024b). However, more detailed investigations in the future may reveal more rigorous results for the ocean than this simplified picture.

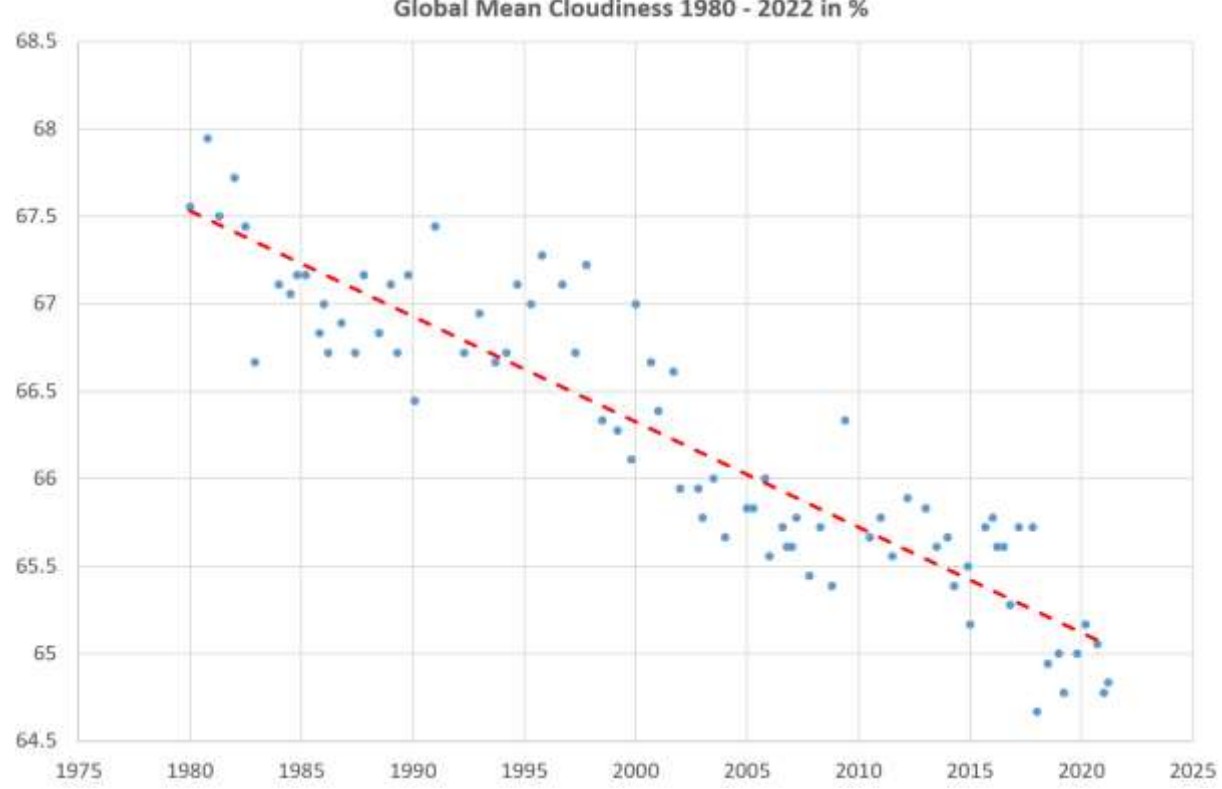

Fig. 13: Dots: satellite-derived global mean cloud area fractions 1980-2022 in percent. Data from
Foster et al. (2023). Dashed line: present cloudiness is 65 % with a climatological linear shrinking
trend of −6.2 % per century.

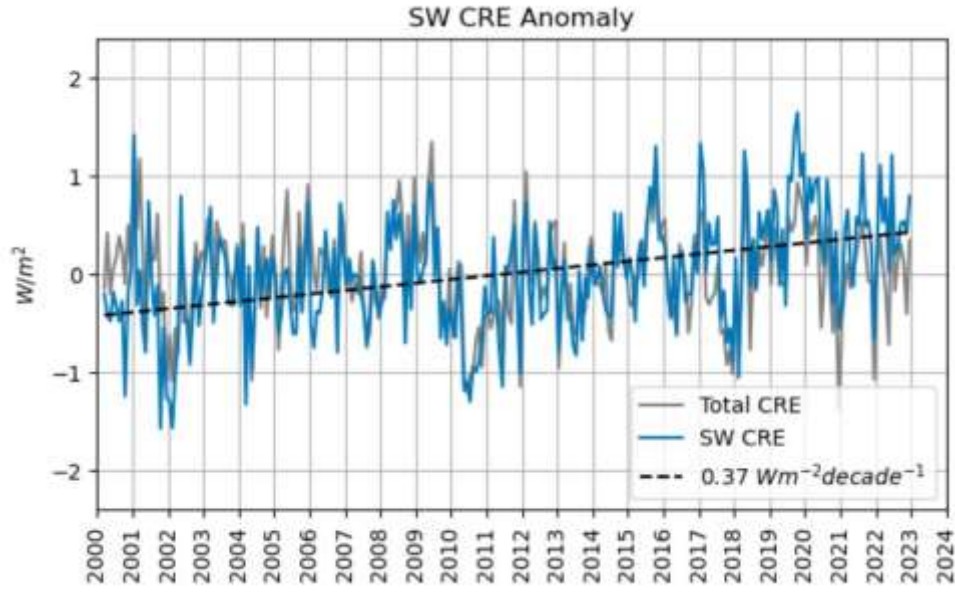

Fig. 14: Short-wave cloud radiative effect (SW CRE) of increasing solar irradiation. Image kindly
provided by Coda Phillips (priv. comm.), with minor correction compared to the similar previous
publication (Phillips and Foster 2023). Total CRE is the net effect of SW and LW CRE, see Fig. 15

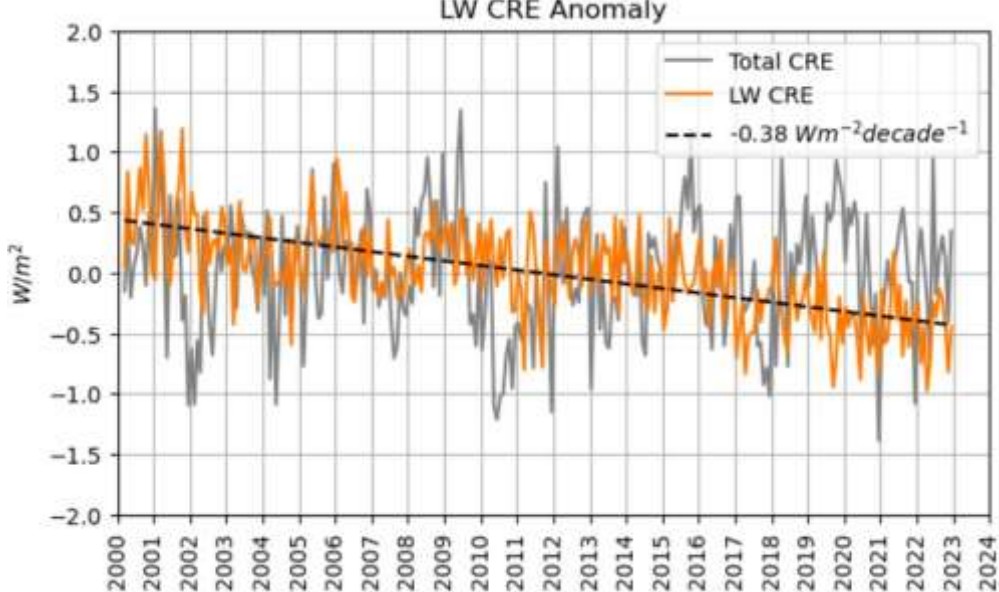


Fig. 15: Long-wave cloud radiative effect (LW CRE) of decreasing net thermal radiation. Image kindly
provided by Coda Phillips (priv. comm.), with minor correction compared to the similar previous
publication (Phillips and Foster 2023). Total CRE is the net effect of SW and LW CRE, see Fig. 14.

*6.2 Cumulus Clouds*
Cumulus clouds are often formed in the course of diurnal convection by isentropic uplift of humid air
parcels from the sea surface to the condensation level, mostly located at low heights between 200
and 500 m. This process permits a thermodynamic description of such clouds (Romps 2014) by
calculating the *lifted condensation level* (LCL) as the cumulus cloud base. In distinction to previous
studies, as the first such international geophysical standard, TEOS-10 provides explicit equations for
entropy, enthalpy and chemical potentials of humid air which may be used to derive reference
equations and values of the LCL (Feistel and Hellmuth 2024b).
At the sea surface pressure, $p_{SS}$, the air parcel may possess the temperature $T_{SS}$ and the relative
fugacity $\psi_f$, which is a real-gas definition of relative humidity (Feistel and Lovell-Smith 2017) in terms
of the chemical potential of water vapour in humid air, $\mu_V^{AV}$, and that of liquid water, $\mu_W$,
$$R_W T_{SS} \ln \psi_f = \mu_V^{AV}(A, T_{SS}, p_{SS}) - \mu_W(T_{SS}, p_{SS}).$$ (49)
Here, $R_W = 461.523 \text{ J kg}^{-1} \text{ K}^{-1}$ is the specific gas constant of water, and $A = 1 - q$ is the dry-air
mass fraction of the parcel, to be determined from $\psi_f$ by this condition.
At the LCL, the parcel is saturated at $\psi_f = 1$, i.e.,
$$0 = \mu_V^{AV}(A, T_{LCL}, p_{LCL}) - \mu_W(T_{LCL}, p_{LCL}).$$ (50)
During uplift, $A$ is assumed to remain constant, as well as the parcel's entropy, $\eta^{AV}$,
$$\eta^{AV}(A, T_{SS}, p_{SS}) = \eta^{AV}(A, T_{LCL}, p_{LCL}).$$ (51)

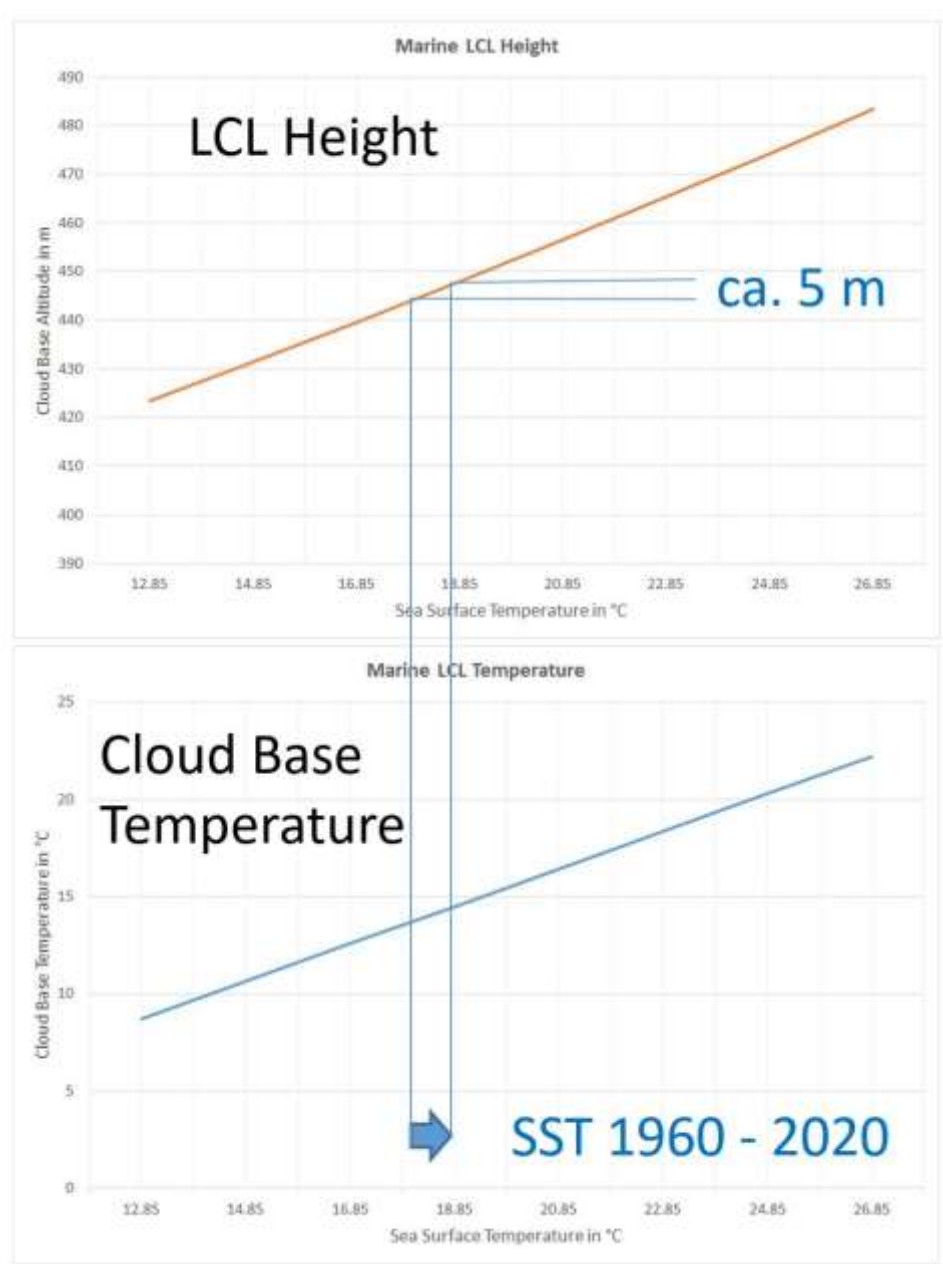

Fig. 16: As a function of typical low-latitude sea-surface temperatures, LCL height (top) and LCL
temperature (bottom) are computed from the TEOS-10 equations (49) – (52) at a typical marine
surface RH of 80 %rh. The added interval indicates the global mean SST change between 1960 and
2020 which has resulted in an increase of the cloud base altitude by about 5 m.

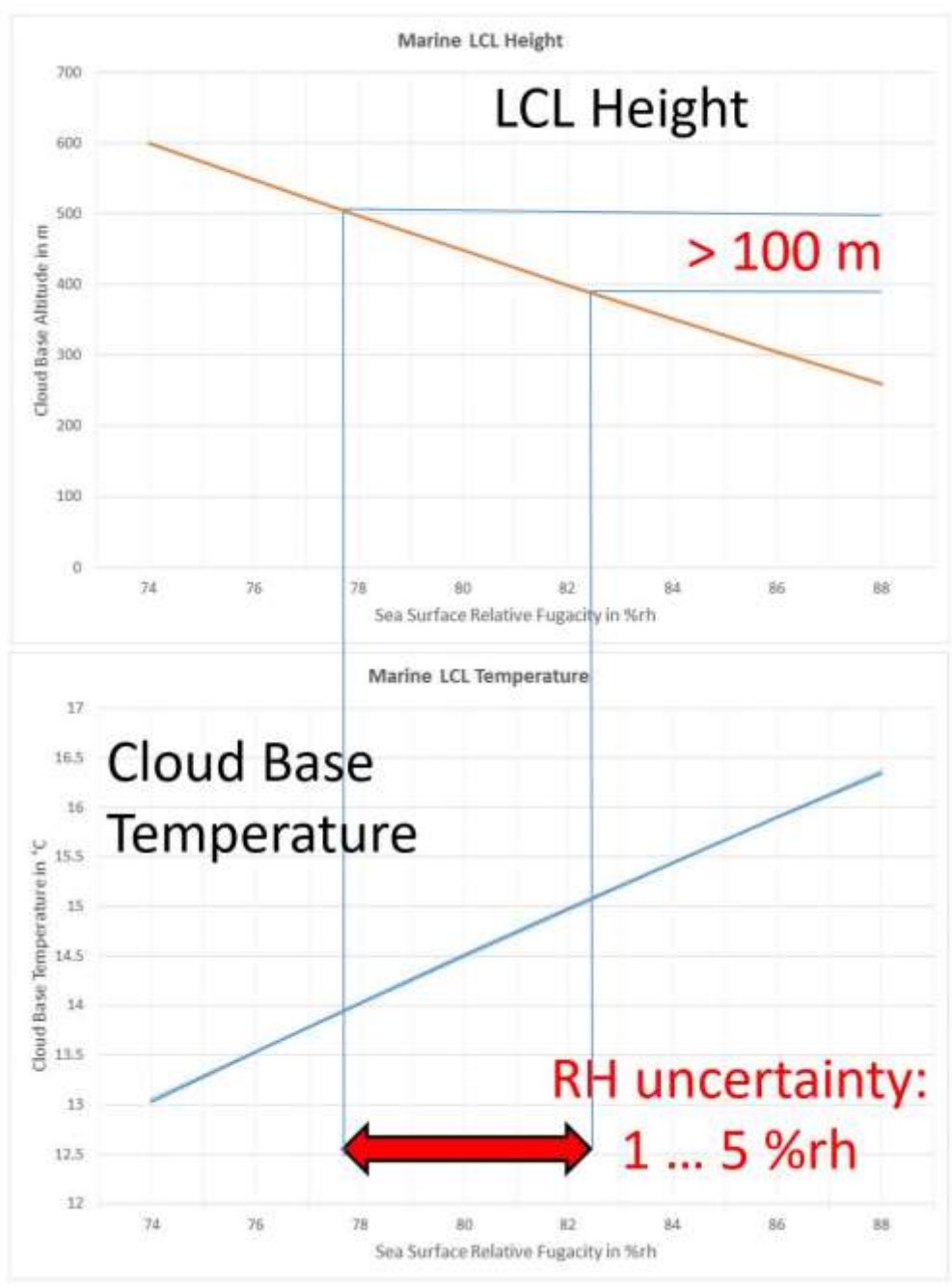

Fig. 17: As a function of typical marine RH values, LCL height (top) and LCL temperature (bottom) are computed from the TEOS-10 equations (49) – (52) at a sea surface temperature of 292 K, close to the current global mean SST of 18.8 °C, see Fig. 18. The added interval indicates the observation uncertainty of sea surface RH which corresponds to an uncertainty of the cloud base altitude of more than 100 m.

Finally, the LCL altitude, $z_{LCL}$, above sea level follows from the isentropic integral of the hydrostatic equation in terms of the enthalpy, $h^{AV}$, of humid air,

$$z_{LCL} = \frac{1}{g_E}\left[h^{AV}(A, \eta^{AV}, p_{SS}) - h^{AV}(A, \eta^{AV}, p_{LCL})\right]. \tag{52}$$

The gravity acceleration is $g_E = 9.81 \text{ m s}^{-2}$. The functions $\mu_V^{AV}$, $\eta^{AV}$, $h^{AV}$ and $\mu_W$ can be expressed by partial derivatives of the TEOS-10 thermodynamic potentials of humid air and liquid water, and

are numerically available from the *Sea-Ice-Air* (SIA) *library* (Feistel et al. 2010d, Wright et al. 2010).
Solving eqs. (49) – (52) numerically, the LCL properties $(A, T_{\mathrm{LCL}}, p_{\mathrm{LCL}}, z_{\mathrm{LCL}})$ are obtained from the
given surface properties, $(\psi_f, T_{\mathrm{SS}}, p_{\mathrm{SS}})$.
Table 1: LCL cloud-base temperatures, $T_{\mathrm{LCL}}$, pressures, $p_{\mathrm{LCL}}$, and heights, $z_{\mathrm{LCL}}$, as functions of the
SST, $T_{\mathrm{SS}}$, at marine surface relative fugacity of $\psi_f = 80$ %rh, computed from TEOS-10 eqs. (49) –
(52), as well as climatic LCL sensitivities, $\alpha, \beta, \gamma$, eq. (53), with respect to increasing SST (Feistel and
Hellmuth 2024). The row printed in bold approximates the current global mean SST, see Fig. 18.

| $T_{\mathrm{SS}}$ K | $T_{\mathrm{LCL}}$ K | $p_{\mathrm{LCL}}$ hPa | $z_{\mathrm{LCL}}$ m | $\alpha$ % K$^{-1}$ | $\beta$ K K$^{-1}$ | $\gamma$ hPa K$^{-1}$ |
|---|---|---|---|---|---|---|
| 286 | 281.883 | 963.093 | 423.468 | −0.0483 | 0.9634 | −0.2742 |
| 288 | 283.810 | 962.542 | 431.481 | −0.0542 | 0.9629 | −0.2773 |
| 290 | 285.735 | 961.984 | 439.660 | −0.0608 | 0.9624 | −0.2806 |
| **292** | **287.659** | **961.419** | **448.017** | **−0.0680** | **0.9619** | **−0.2841** |
| 294 | 289.583 | 960.847 | 456.561 | −0.0759 | 0.9614 | −0.2878 |
| 296 | 291.505 | 960.268 | 465.305 | −0.0846 | 0.9608 | −0.2917 |
| 298 | 293.426 | 959.680 | 474.263 | −0.0942 | 0.9603 | −0.2959 |
| 300 | 295.346 | 959.084 | 483.449 | −0.1047 | 0.9597 | −0.3004 |



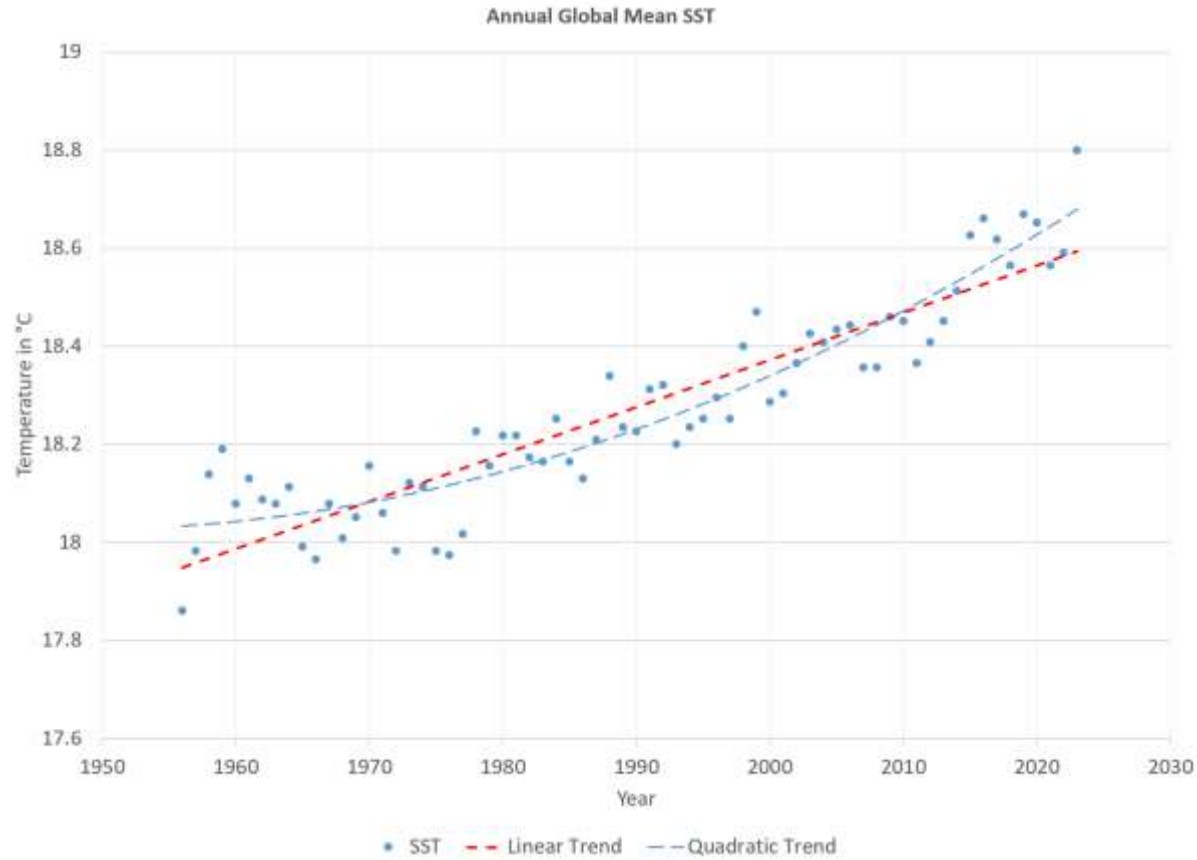


Fig. 18: Estimated increase 1957 – 2023 of global annual mean sea-surface temperatures (source:
Cheng et al. 2024). The linear trend (red) is $t/°\mathrm{C} \approx 18 + 0.01 \times (yr - 1961)$. The quadratic trend
curve (blue) suggests an acceleration of warming.

As solutions of eqs. (49) – (52), height and base temperature of marine cumulus clouds, as a function
of the sea-surface temperature $T_0$ at a sea-surface relative fugacity of $\psi_f = 80$ %rh, are displayed in
Fig. 16. Similarly, height and base temperature, as a function of the sea-surface relative fugacity RF of
$\psi_f$ at a sea-surface temperature $T_{SS} = 292$ K, close to the current global mean SST, are displayed in
Fig. 17. It is obvious that the LCL effect of the range of RF uncertainty exceeds significantly the effect
caused by global SST rise, so that unknown minor systematic RF changes may easily disguise the
thermal effects on marine cumulus clouds.
Global mean sea-surface temperature has risen from about 17.9 °C in 1956 to 18.8 °C in 2023 (Cheng
et al. 2024), see Fig. 18. This pronounced climatic trend is expected to let the cumulus cloud base lift
up while at the same time warming it, see Fig. 16, but not as much as the SST itself is increasing. The
related climatic sensitivities possess complicated dependencies but may directly be derived by taking
the related derivatives
$$\alpha \equiv \left(\frac{\partial A}{\partial T_{SS}}\right)_{p_{SS},\psi_f} = -\left(\frac{\partial q}{\partial T_{SS}}\right)_{p_{SS},\psi_f}, \beta \equiv \left(\frac{\partial T_{LCL}}{\partial T_{SS}}\right)_{p_{SS},\psi_f} \text{ and } \gamma \equiv \left(\frac{\partial p_{LCL}}{\partial T_{SS}}\right)_{p_{SS},\psi_f}, \quad (53)$$
of the TEOS-10 LCL equations (49) – (52) with respect to the surface temperature while keeping
surface RH fixed (Feistel and Hellmuth 2024). Selected results for those sensitivities are given in Table
1 relative to 1 °C rise of SST, similar to that in the past 70 years (Fig. 16). Here, $\alpha \approx -0.07$ % K$^{-1}$
describes the rate of increase of specific humidity at the sea surface, often dubbed the "Clausius-
Clapeyron effect". The value of $\beta \approx 0.96$ indicates that the cumulus cloud base warms up slower
that the ocean by about 4 %, and $\gamma \approx -0.28$ hPa K$^{-1}$ is the LCL pressure lowering caused by ocean
warming, corresponding to ascending clouds. The value $\beta < 1$ implies that the thermal downward
radiation from the cloud base does not keep pace with the ocean upward radiation, so that the net
climatic feedback of cumulus clouds is negative and acts against ocean warming. These clouds do not
provide a physical explanation for the observed enhanced ocean warming.

*6.3 Stratocumulus and Other Clouds*
"Marine low clouds strongly cool the planet" (Myers et al. 2021). Over the Atlantic, "the strongest
surface longwave cloud effects were shown in the presence of low level clouds" (Kalisch and Macke
2012). "Low-cloud feedbacks are also a leading cause of uncertainty in future climate prediction
because even small changes in cloud coverage and thickness have a major impact on the radiation
budget" (Wood 2012: p. 2373).
Generally, however, the dominating cloud type over the ocean is stratocumulus (Eastman et al.
2011). "They are common over the cooler regions of subtropical and midlatitude oceans where their
coverage can exceed 50% in the annual mean" (Wood 2012: p. 2373) with a typical thickness about
320 m and "a tendency for thicker clouds (median 420 m) in mid- and high latitudes" (Wood 2012: p.
2378). "Stratocumuli tend to form under statically stable lower-tropospheric conditions" (Wood
2012: p. 2374). On the annual average, stratocumulus is particularly frequent (up to 60 % coverage)
at the subtropical coastal upwelling regions such as the cold Benguela, Humboldt and California
Currents (Wood 2012: Fig. 4a, Muhlbauer et al. 2014: Fig. 2). However, in those areas there is no
obvious correlation of cloud cover with ocean warming (Fig. 1). Stratocumulus also forms large cloud
cover (about 20 % coverage) in the boreal and austral west-wind bands (Wood 2012: Fig. 4a) where
the ocean is strongly warming up (Fig. 1).
"Only small changes in the coverage and thickness of stratocumulus clouds are required to produce a
radiative effect comparable to those associated with increasing greenhouse gases" (Wood 2012: p.
2374). Marine stratocumulus cloud feedback is still a major challenge and source of uncertainty of
climate models (Hirota et al. 2021). However, "similar to other low-cloud types in the marine
boundary layer, the impact of stratocumulus clouds on the outgoing longwave radiation is marginal
due to the lack of contrast between the temperature of stratocumulus cloud tops and the
temperature of the sea surface over which they form. Thus, the net radiative effect of stratocumulus
clouds is primarily controlled by factors influencing their shortwave cloud forcing such as the cloud
albedo and the cloud coverage" (Muhlbauer et al. 2014: p. 6695).
Following this argumentation and assuming that the short-wave cloud effect of stratocumuli on the
ocean radiation balance by far outweighs their long-wave effects, then the short-wave warming
effect (Fig. 14) of decreasing cloudiness may dominate over the long-wave cooling (Fig. 15). Possibly,
this could make stratocumulus a potential candidate for causing the unclear recent ocean warming.
Similarly, in the diurnal cycle, short-wave effects (Fig. 14) have an impact at daytime only, while long-
wave effects (Fig. 15) are present all 24 hours. In this respect, Luo et al. (2024) report that low-level
cloudiness has an asymmetric day/night trend which enhances ocean warming. Regionally, where in
spring the days get longer, and the heavy cloudiness of the west-wind belt becomes replaced by
fewer subtropical clouds (see Fig. 12), the systematic reduction of cloudiness may be expected to
produce local excess warming such as near the subtropical fronts (see Fig. 1). Only dedicated future
model studies, however, may reliably verify such speculations. As a recent example for the
complexity of SST warming "by suppressing the evaporative cooling" of the ocean, Wang et al. (2024)
explain dramatic but yet elusive warming events in the North-East Pacific by changes in ocean-
atmosphere mechanisms caused by reduced Chinese aerosol emissions. Also, Berthou et al. (2024)
describe cloud cover feedback over the sea during an unprecedented marine heatwave off northwest
Europe in 2023.

## 916    7   Summary

Substantial uncertainties of estimated heat fluxes at the ocean-atmosphere interface, such as the
"ocean heat budget closure problem", prevent reliable model predictions and causal explanations of
climate phenomena that may take place within the range of those uncertainties. Among such
"surprises" is the currently registered excessive ocean warming, but are also the subsequent
consequences of this warming, such as those expected for global weather processes.
Intending to reduce model uncertainties of thermal energies and heat fluxes in the climate system
associated with the global circulation of water in its different phases and mixtures, the new
geophysical thermodynamic standard TEOS-10 had been adopted internationally in 2009 and 2011.
Meanwhile, the uptake of TEOS-10 by the scientific community is mainly focussed on ocean
observations and modelling, as the related publication metrics are suggesting (Appendix A).
TEOS-10 is advanced over previous similar standards and various collections of tailored empirical
property equations by (i) its completeness in describing all thermodynamic properties of seawater,
humid air and ice, including their entropies, enthalpies and chemical potentials, by (ii) its perfect
mutual consistency between different phases and mixtures, and by (iii) its minimum uncertainty over
maximum ranges of validity. Among its particularly favourable fields of application are composite
systems with internal phase boundaries such as air sea interaction or cloud formation.
In addition to entropies, enthalpies and chemical potentials, TEOS-10 has made available certain new
quantities for the description and modelling of climate processes, such as (i) Absolute Salinity of the
ocean with a specified Reference Composition, (ii) Conservative Temperature as a measure of

Potential Enthalpy of seawater representing a definite heat content, and (iii) Relative Fugacity as the thermodynamic driving force of evaporation, suggesting an improved full-range definition of relative humidity as a substitute for mutually inconsistent and restricted such definitions in practical use in climatology, meteorology and physical chemistry.

This paper explains some tutorial examples for the application of TEOS-10 to selected current climate problems. There is (i) the two-phase conceptual model of "sea air" which provides rigorous equations for the latent heat of evaporation, for the heat capacity of humid air including salty aerosols (sea spray), and for the irreversible production of entropy by evaporation into the marine troposphere. There is also (ii) the formation of low marine cumulus clouds by isentropic thermal convection up to their condensation level, and their climatic feedback to surface temperature and humidity concerning their infrared radiation effects.

It is currently unclear why and how the ocean warming is intensifying, and when and how the related enormous amount of heat may transfer to the atmosphere. The observed systematic reduction of cloudiness may play an important role in this process, but responsible details and theoretical causes are unknown. Marine surface relative humidity is an important and rather sensitive "control valve" for the supply of the troposphere with latent heat, however, the common assumption of constant relative humidity during climate change lacks rigorous explanation and leaves open the question of its possible trends below the insufficiently high level of observational uncertainty. TEOS-10 may further assist climate modellers to address such issues.

Ocean Science has proved a scientifically well-reputable, reliable and successful partner journal for the publication of advanced results and methods in oceanography and geophysics. Cooperation with international bodies such as IUGG, UNESCO/IOC, IAPSO, SCOR, IAPWS and BIPM has made possible the development and international introduction of TEOS-10. The established standing committee JCS remains active with respect to related fundamental problems yet to be solved. It is hoped and expected that TEOS-10 may constitute a reliable long-term thermodynamic basis for interdisciplinary climate research.

**Appendix A: Summary and Metrics of Selected Publications Related to TEOS-10**

Between December 2008 and December 2012, supporting the activities of SCOR/IAPSO WG127, *Ocean Science* had published 16 articles open-access in its Special Issue #14, "Thermophysical properties of seawater" (Feistel et al. 2008a). From February 2013 on, monthly metrics have been recorded by the journal. Table A1 reports those metrics of the last decade.

For comparison, metrics – as far as published elsewhere by 04 April 2024 – of selected TEOS-10 articles listed at www.teos-10.org are reported in Table A2.

**Table A1**: Metrics of articles in the *Ocean Science* Special Issue #14, "Thermophysical properties of seawater" (Feistel et al. 2008a), from February 2013 till March 2024. "SIA" stands for the TEOS-10 Sea-Ice-Air open source code library.

| Reference | Topic | Accessed | PDF Downloads | Cited |
|---|---|---|---|---|
| Millero and Huang (2009) | Seawater at High $T,S$ | 16 462 | 11 061 | 79 |
| Feistel et al. (2010c) | Baltic Sea Density/Salinity | 15 435 | 11 385 | 92 |
| Pawlowicz et al. (2011) | Seawater Biogeochemistry | 9 663 | 6 444 | 47 |
| McDougall et al. (2012) | Global Absolute Salinity | 9 290 | 5 489 | 116 |
| Feistel et al. (2010a) | Humid Air Helmholtz Function | 8 737 | 5 346 | 31 |

| Safarov et al. (2009) | Seawater at High *T*,*p* | 7 356 | 4 308 | 68 |
|---|---|---|---|---|
| Wright et al. (2011) | Density Salinity | 5 268 | 2 891 | 49 |
| Marion et al. (2009) | CaCO$_3$ Solubility | 5 169 | 3 170 | 36 |
| Pawlowicz (2010) | Composition Variation | 4 471 | 2 666 | 27 |
| Feistel et al. (2010d) | SIA Library Equations | 4 255 | 2 416 | 23 |
| Wright et al. (2010) | SIA Library Routines | 4 049 | 1 733 | 19 |
| Feistel et al. (2008b) | Consistent New Potentials | 3 585 | 1 527 | 27 |
| Seitz et al. (2011) | Salinity Traceability | 3 363 | 1 705 | 24 |
| Feistel et al. (2010b) | Baltic Property Anomalies | 3 183 | 1 500 | 12 |
| Tailleux (2009) | Mixing Efficiency | 2 752 | 1 303 | 11 |
| Millero and Huang (2010) | Seawater at High *T*,*S* (corrig.) | 2 189 | 909 | 1 |


**Table A2**: Metrics published by March 2024 of selected TEOS-10 related articles apart from *Ocean*
*Science* Special Issue #14. "Ice Ih" is the ambient, hexagonal ice I phase of water.

| Reference | Topic | Accessed | PDF Downloads | Cited |
|---|---|---|---|---|
| Wagner and Pruß (2002) | Water Helmholtz Function | 7 516 | 7 516 | 3 457 |
| Jackett et al. (2006) | Algorithms for Seawater | 2 877 | 2 364 | 119 |
| Feistel (2005) | Seawater Gibbs Function | 2 584 | 1 126 | 10 |
| Feistel et al. (2005) | Ice Ih Gibbs Function | 2 288 | 1 015 | 5 |
| Lemmon et al. (2000) | Dry Air Helmholtz Function | 2 279 | 2 279 | 381 |
| McDougall (2003) | Potential Enthalpy | 1 970 | 1 367 | 50 |
| Wagner et al. (2011) | Ice Melting/Sublimation | 1 467 | 510 | 102 |
| Seitz et al. (2010) | Salinity Determination | 1 332 | | 15 |
| Feistel (2008b) | IAPWS-06 and IAPWS-08 | 1 279 | | 4 |
| Millero et al. (2008) | Seawater Composition | 970 | | 780 |
| Feistel and Wagner (2006) | Ice Ih Gibbs Function | 843 | 843 | 286 |
| Feistel and Wagner (2005) | Ice Ih Gibbs Function | 833 | | 58 |
| Graham and McDougall (2013) | Conservative Temperature | 651 | 467 | 28 |
| Feistel (2012) | New TEOS-10 Standard | 436 | | 27 |
| Spall et al. (2013) | TEOS-10 for oceanography | 230 | 128 | 3 |
| Feistel (2008a) | Seawater Gibbs Function | 134 | | 133 |
| Roquet et al. (2015) | TEOS-10 Polynomials | 111 | | 97 |
| Feistel and Wagner (2007) | Ice Ih Sublimation >20 K | 105 | | 112 |
| Feistel (2003) | Seawater Gibbs Function | 100 | | 105 |
| McDougall et al. (2013) | Thermodynamics of Seawater | 35 | | 10 |
| Feistel and Marion (2007) | Seawater Gibbs-Pitzer | 25 | | 32 |
| Valladares et al. (2011) | Replacement of EOS-80 | 14+5 | | 4+1 |
| Feistel et al. (2006) | New Seawater Equation | | | |


**Table A3**: IAPWS documents supporting TEOS-10, openly accessible at www.iapws.org. IAPWS
documents are independently and painstakingly verified before they may become adopted at an
annual meeting. No metrics available.

| Document | Code | Topic | Meeting | Year |
|---|---|---|---|---|
| Release | R06-95 | Water Helmholtz Function | Dresden | 2016 |
| Release | R10-06 | Ice Ih Gibbs Function | Doorwerth | 2009 |
| Release | R13-08 | Seawater Gibbs Function | Berlin | 2008 |
| Release | R14-08 | Ice Melting/Sublimation | Pilsen | 2011 |

| Suppl. Release | SR1-86 | Water Saturation Properties | St. Petersburg | 1992 |
|---|---|---|---|---|
| Suppl. Release | SR6-08 | Liquid Water at 0.1 MPa | Pilsen | 2011 |
| Suppl. Release | SR7-09 | Liquid Water Gibbs Function | Doorwerth | 2009 |
| Guideline | G05-01 | Fundamental Constants | Virtual Online | 2020 |
| Guideline | G08-10 | Humid Air Helmholtz Function | Niagara Falls | 2010 |
| Guideline | G09-12 | Cold Water Vapour < 130 K | Boulder | 2012 |
| Guideline | G11-15 | Fugacity Virial Equation | Stockholm | 2015 |
| Guideline | G12-15 | Supercooled Water | Stockholm | 2015 |
| Advisory Note | AN4-09 | IAPWS/CIPM Water Density | Doorwerth | 2009 |
| Advisory Note | AN5-13 | Industrial Seawater | Dresden | 2016 |
| Advisory Note | AN6-16 | IAPWS support for TEOS-10 | Dresden | 2016 |


**Table A4:** Numbers of unique internet downloads 2011-2023 of supporting material from the TEOS-
10 homepage at www.teos-10.org. "GSW" stands for the TEOS-10 Gibbs Seawater open source code
library. Data from Pawlowicz (2023)

| Item | 2011 -13 | 2013 -14 | 2014 -15 | 2015 -16 | 2016 -17 | 2017 -18 | 2018 -19 | 2019 -20 | 2020 -21 | 2021 -22 | 2022 -23 |
|---|---|---|---|---|---|---|---|---|---|---|---|
| TEOS-10 Manual | 920 | 360 | 535 | 552 | 418 | 427 | 349 | 472 | 479 | 482 | 530 |
| Getting Started | 879 | 362 | 558 | 547 | 427 | 475 | 349 | 444 | 460 | 483 | 479 |
| Lecture Slides | 704 | 284 | 374 | 318 | 219 | 248 | 204 | 272 | 272 | 231 | 272 |
| TEOS-10 Primer | 584 | 197 | 289 | 297 | 222 | 217 | 187 | 253 | 260 | 226 | 268 |
| GSW MATLAB | 1920 | 1102 | 1485 | 1814 | 1235 | 1552 | 1233 | 1556 | 1504 | 1747 | 1897 |
| GSW FORTRAN | 366 | 222 | 171 | 162 | 127 | 116 | 82 | 98 | 83 | 92 | 87 |
| GSW C | 202 | 84 | 133 | 151 | 85 | 96 | 59 | 81 | 58 | 49 | 57 |
| GSW PHP | - | 55 | 61 | 43 | 29 | 60 | 28 | 52 | 22 | 22 | 21 |
| SIA VB | 72 | 100 | 46 | 45 | 45 | 48 | 43 | 47 | 47 | 38 | 30 |
| SIA FORTRAN | 59 | 118 | 58 | 44 | 36 | 42 | 37 | 42 | 31 | 33 | 31 |


**Table A5**: Selected additional TEOS-10 related readings, metrics by March 2024

| Reference | Topic | Accessed | PDF Downloads | Cited |
|---|---|---|---|---|
| Turner et al. (2016) | Seawater Pitzer Model | 13 780 | 1 175 | 21 |
| Lovell-Smith et al. (2016) | Relative Humidity Challenges | | 6 502 | 27 |
| Schmidt et al. (2018) | Density-Salinity Relation | 9 421 | 5 481 | 28 |
| Feistel et al. (2016a) | Challenges beyond TEOS-10 | | 5 023 | 49 |
| Dickson et al. (2016) | Seawater pH Challenges | | 2 818 | 43 |
| Pawlowicz et al. (2016) | Seawater Salinity Challenges | | 2 738 | 40 |
| Smythe-Wright et al. (2019) | IAPSO's history and roles | 5 893 | 384 | 3 |
| Foken et al. (2021) | Atmospheric Measurements | 5 709 | | 2 |
| Feistel (2018) | TEOS-10 Review | 5 441 | 1 632 | 38 |
| Feistel and Hellmuth (2023) | Dalton Equation | 5 068 | | 1 |

| | | | | |
|---|---|---|---|---|
| McDougall et al. (2021) | Ocean Heat Flux and Content | 4 993 | 1 425 | 5 |
| Hellmuth et al. (2020) | Ice-Crystal Nucleation | 4 811 | | 6 |
| Uchida et al. (2019) | Optical Density Sensor | 3 513 | | 19 |
| Hellmuth et al. (2021) | Mass Density of Humid Air | 2 643 | | 4 |
| Feistel and Lovell-Smith (2017) | Relative Fugacity Part 1 | | 1 335 | 18 |
| Le Menn et al. (2018) | Seawater Salinity Measurands | | 1 136 | 13 |
| Schmidt et al. (2016) | Seawater Density up to 1 ppm | | 950 | 21 |
| Ji et al. (2021) | Absolute Salinity off China | 2 462 | 856 | 3 |
| Von Rohden et al (2016) | Baltic Sound Speed | 2 122 | 784 | 1 |
| Feistel et al. (2016b) | Uncertainty of Correlation Eqs. | | 662 | 14 |
| Martins and Cross (2022) | TEOS-10 Excel Code | 2 087 | 542 | 2 |
| Hellmuth and Feistel (2020) | Low-Density Subcooled Water | 1 827 | | 1 |
| Feistel (2011a) | Stochastic Potential Functions | 1 217 | | 6 |
| McDougall et al. (2014) | Sea Ice Formation | 1 124 | 771 | 16 |
| Feistel and Hellmuth (2024a) | Evaporation Entropy | 1 038 | | 0 |
| Young (2010) | Boussinesq Approximation | 928 | 724 | 56 |
| Harvey et al. (2023) | Water Properties | 874 | 369 | 9 |
| Tailleux (2018) | Local Available Potential Energy | 807 | 409 | 11 |
| Uchida et al. (2020) | Seawater Intercomparison | 707 | 764 | 6 |
| Sharkawy et al. (2010) | Review of Seawater Correlations | 701 | | 946 |
| Feistel (2019a) | Relative Fugacity Part 2 | | 267 | 3 |
| Feistel et al. (2022) | Relative Fugacity Part 3 | | 252 | 4 |
| McDougall et al. (2023) | Seawater Potential of (S, CT, p) | 629 | 122 | 1 |
| Feistel et al. (2015) | Virial Fugacity Equation | 581 | | 17 |
| Nayar et al. (2016) | Seawater Property Review | 553 | | 366 |
| Marion et al. (2011) | Seawater pH | 491 | | 170 |
| Feistel and Lovell-Smith (2023) | Systematic Error in Regression | 428 | 41 | |
| Holzapfel and Klotz (2024) | $H_2O$ and $D_2O$ Ice Ih | 285 | 58 | |
| Ji et al. (2024) | Bohai Sea Salinity Anomaly | 254 | 56 | |
| Holzapfel and Klotz (2021) | Thermal Expansion of Ice Ih | 245 | 77 | 1 |
| Pawlowicz and Feistel (2012) | TEOS-10 in Limnology | | | 22 |
| Kretzschmar et al. (2015) | Industrial Seawater Equation | 104 | | 0 |
| Ebeling et al. (2020) | Individual Ionic Activities | 99 | | 10 |
| Marion et al. (2010) | FREZCHEM Solution Model | 82 | | 74 |
| Sun et al. (2008) | Saline Thermal Fluid Equations | 79 | | 84 |
| Almeida et al. (2018) | TEOS-10 Atlantic Impact | 53 | | 5 |
| Safarov et al. (2012) | High-Salinity Seawater | 42 | | 21 |
| Woosley et al. (2014) | World Ocean Absolute Salinity | 39 | | 16 |
| Safarov et al. (2013) | Brackish Seawater Properties | 35 | | 15 |
| Ulfsbo et al. (2015) | Seawater Activity Coefficients | 34 | | 11 |
| Feistel and Hagen (1998) | Sea Ice Gibbs Function | 24 | | 31 |
| Feistel (2010) | Seawater Gibbs Function | 23 | | 24 |
| Tailleux (2010) | Buoyancy Power Input | | | 20 |
| Millero and Huang (2011) | Seawater Compressibility | 19 | | 19 |
| Tchijov et al. (2008) | Ice at High $p$ and Low $T$ | 19 | | 6 |
| Von Rohden et al. (2015) | Seawater Sound Speed 0.1 MPa | | | 18 |
| Budéus (2018) | TEOS-10 Density Bias ? | 8 | | 5 |
| Lago et al. (2015) | Seawater Sound Speed < 70 MPa | 8 | | 4 |
| Manaure et al. (2021) | Individual Ionic Activities | 8 | | 2 |
| Weinreben and Feistel (2019) | Anomalous Salinity Density | 8 | | 1 |
| Pawlowicz & Yerubandi (2024) | Water as a Substance | 4 | | 0 |

| Ebeling et al. (2022) | Individual Ionic Activities | 2 | | 6 |
|---|---|---|---|---|
| Waldmann et al. (2022) | Uncertainty of Ocean Variables | 2 | | |
| Tailleux and Dubos (2024) | Seawater Static Energy | 1 | | 1 |
| Pawlowicz (2013): | Physical Variables in the Ocean | | | |
| Laliberte (2015) | TEOS-10 Python Code | | | |
| Thol et al. (2024) | $N_2$-$O_2$-Ar Helmholtz Function | | | |


**Appendix B: Thermodynamic Potentials**

This Appendix provides a short introduction to thermodynamic potentials, supporting the equations and topics discussed in this article. Alternative presentations from different perspectives are available from numerous textbooks such as Guggenheim (1949), Margenau and Murphy (1964), Landau and Lifschitz (1966) or Kittel (1969). For seawater, the use of a Gibbs thermodynamic potential was first suggested theoretically by Fofonoff (1958, 1962), see also Craig (1960).

A key theoretical tool for the physical investigation of the globally warming climate and the related energy balances is *thermodynamics*. It is known from experience that there exists a distinguished state of various ambient substances that is known as a *thermodynamic equilibrium state*. If a sample of matter is in this state, it may never spontaneously alter its measurable macroscopic properties unless it becomes disturbed by external contact and exchange of energy or matter with its surrounding. Typical properties which characterise a particular equilibrium state are the total mass of a sample, $m$, its volume, $V$, its temperature, $T$, or its pressure, $p$. Of a given sample, different equilibrium states may exist that differ in those quantities, but there exists a specific relation between those variables, known as an *equation of state*, which is characteristic for the given substance and remains universally valid at any of its possible equilibrium states. The most general and comprehensive equation of state of a given substance is a *thermodynamic potential* of that substance.

Thermodynamics is a mathematical theory for the construction and exploitation of equations of state and of properties derived therefrom for the prediction or verification of observations or experiments. Depending on the properties of interest, equations of state may be formulated in various different mathematical forms. It was discovered by J. Willard Gibbs (1873) that from a suitable thermodynamic potential all thermodynamic properties of a given substance at any of its equilibrium states can be derived by appropriate mathematical methods.

For theoretical reasons (namely, the statistical so-called *canonical ensemble*, Landau and Lifschitz 1966: §31; Kittel 1969: Ch. 18), a preferred thermodynamic potential of a pure substance is its *Helmholtz Energy*, or *Free Energy*, $F(m, T, V)$, expressed in terms of the sample's mass, $m$, its temperature und volume. For mixtures, the single mass must be replaced by the set of partial masses of the species involved. Here, mass is used as a measure for the amount of substance, rather than particle or mole numbers, for the practical reason that in oceanography masses are easier measured than moles, and so TEOS-10 is following that tradition and is a mass-based description. Classical empirical thermodynamics of Clausius and Gibbs was formulated independently of the existence and properties of atoms or molecules which presently define the mole (BIPM 2019).

To the *Internal Energy $E$* of the sample, the Helmholtz energy is related by the Helmholtz Differential Equation,

$$E = F - T \left( \frac{\partial F}{\partial T} \right)_{m,V} \tag{B.1}$$

Note that IOC et al. (2010) uses the symbol $U$ for the Internal Energy rather than $E$ in eq. (B.1). This
replacement is done here for denoting with $u$ the wind speed, eq. (6), rather than specific internal
energy, which is defined here by $e = E/m$, eqs. (1) and (B.3). The symbol $E$ is frequently used in the
thermodynamic literature, for example by Gibbs (1873a) and Landau and Lifschitz (1966).
The potential function $F$ is extensive, which means that for instance $F(2\,m, T, 2\,V) = 2\,F(m, T, V)$ is
valid for an equilibrium sample of twice the mass. It follows that the mass-specific Helmholtz
function, $F/m \equiv f(T, \rho)$, depends on two variables only, $T$ and the mass density, $\rho \equiv m/V$, and is
mathematically simpler and more convenient than $F$, which may always be retrieved from a given $f$
by
$$F(m, T, V) = m \times f\left(T, \frac{m}{V}\right). \tag{B.2}$$
The quantitative description of a substance of interest in the form of a thermodynamic potential such
as $f(T, \rho)$ has axiomatic properties. The description is *complete*, i.e., all thermodynamic properties of
that substance are available, it is *consistent*, i.e., for any property one and only one result can be
derived, and it is *independent*, i.e., no part of this description may be omitted without loosing the
completeness. It is obvious that such axiomatic properties are very desirable for the description of
geophysical substances, however, such thermodynamic potentials are rarely found in the
corresponding literature. In particular in climate research which combines results and data from
different disciplines, such as meteorology and oceanography, from research carried out all over the
globe and over the years by subsequent generations of specialists, international binding standards
such as the International System of Units (SI) are required that ensure mutual consistency and
metrological comparability of any involved data produced from experiments, observations and
models.
Gibbs' (1873a) original potential function was (internal) energy, $e = E/m$. It is known that a sample's
energy can be increased by compression, $-p\,dv$, where $v = 1/\rho$ is the specific volume, or by input of
heat, $T\,d\eta$, where $\eta = N/m$ is the specific entropy. As an extensive quantity, entropy introduced by
Clausius (1865, 1976) is denoted here by $N$ to avoid confusion with seawater salinity, $S$. Energy
conservation implies that
$$de = T\,d\eta - p\,dv. \tag{B.3}$$
Any such change between different equilibrium states of the same sample takes place along a
definite, substance-specific surface $e(\eta, v)$ so that $de$ in eq. (B.3) is mathematically an exact
differential and the partial derivatives of $e$ possess the physical meanings that
$$T = \left(\frac{\partial e}{\partial \eta}\right)_v, \qquad -p = \left(\frac{\partial e}{\partial v}\right)_\eta. \tag{B.4}$$
Gibbs (1873b) also demonstrated that for several equilibrium samples in contact with one another, in
absence of gravity or accelerated motion, the samples are in mutual equilibrium only if they have
equal values of the coefficients $T$ and $p$ of eq. (2.3).
In the geophysical practice, the quantities $\eta$ and $v$ are difficult to measure, in contrast to, say, $T$ or $p$.
Mathematically equivalent to $e(\eta, v)$, thermodynamic potentials in terms of the other three possible
pairs of independent variables are formally obtained from so-called Legendre transforms (Alberty
2001), namely the *Helmholtz function* $f(T, v) \equiv e - T\eta$ with the differential
$$df = -\eta\,dT - p\,dv, \tag{B.5}$$
the *Gibbs function* $g(T, p) \equiv f + pv = e - T\eta + pv$ with
$\mathrm{d}g = -\eta\mathrm{d}T + v\mathrm{d}p,$                                                                    (B.6)
and the specific *enthalpy* $h(\eta, p) \equiv g + T\eta = f + T\eta + pv = e + pv$ with
$\mathrm{d}h = T\mathrm{d}\eta + v\mathrm{d}p.$                                                                   (B.7)
Depending on the application purpose, each of these potential functions has certain advantages and
disadvantages, and having all of them optionally at hand in mutually consistent versions is most
useful.
Gibbs (1874-78) also considered a situation in which a given sample may exchange substance with its
surrounding. If the exchanged mass of substance *i* is $\mathrm{d}m_i$, the related change of the sample's
(extensive) energy $E$ at constant entropy and volume is termed the *chemical potential* $\mu_i$ of that
substance,
$\mathrm{d}E = T\mathrm{d}N - p\mathrm{d}V + \sum_i \mu_i\, \mathrm{d}m_i,$                                                      (B.8)
so that this exact differential implies that the chemical potential is obtained from
$\mu_i \equiv \left(\frac{\partial E}{\partial m_i}\right)_{N,V,m_{j\neq i}} = \left(\frac{\partial F}{\partial m_i}\right)_{T,V,m_{j\neq i}} = \left(\frac{\partial G}{\partial m_i}\right)_{T,p,m_{j\neq i}} = \left(\frac{\partial H}{\partial m_i}\right)_{N,p,m_{j\neq i}}.$     (B.9)
Equilibrium of a spatially extended substance, in absence of gravity or accelerated motion, requires
that in addition to $T$ and $p$, also the chemical potential $\mu_i$ separately for each present substance
needs to possess the same value anywhere in the volume. "The potential for each component
substance must be constant throughout the whole mass" (Gibbs 1874-78: p. 119).
As intensive properties, the specific energies cannot depend on the total mass but only on the mass
fractions, $w_i \equiv m_i/m$. Because by definition $\sum w_i = 1$, only $(n - 1)$ different fractions may be
independent variables describing the $n$ components of a mixture. For example, one of the
components may be chosen as a master species, "0", such as a solvent, and the remaining ones, $i =$
$1, \dots, n - 1$, may denote the solutes.
In terms of $T$ and $p$, chemical potentials are computed from the Gibbs function, $g$, through the Gibbs
energy, $G$, of eq. (B.9). Because the Gibbs function depends only on the independent intensive
variables, $g(w_1, \dots, w_{n-1}, T, p)$, the solutes' chemical potentials, $i > 0$, are
$\mu_i = \left(\frac{\partial G}{\partial m_i}\right)_{T,p,m_{j\neq i}} = \left(\frac{\partial(m\,g)}{\partial m_i}\right)_{T,p,m_{j\neq i}} = g + \left(\frac{\partial g}{\partial w_i}\right)_{T,p,w_{j\neq i}} - \sum_{j=1}^{n-1} w_j \left(\frac{\partial g}{\partial w_j}\right)_{T,p,w_{k\neq j}}$   (B.10)
Similarly, the solvent's chemical potential is
$\mu_0 = \left(\frac{\partial G}{\partial m_0}\right)_{T,p,m_{j>0}} = \left(\frac{\partial(m\,g)}{\partial m_0}\right)_{T,p,m_{j>0}} = g - \sum_{j=1}^{n-1} w_j \left(\frac{\partial g}{\partial w_j}\right)_{T,p,w_{k\neq j}}.$   (B.11)
Therefore, the *relative chemical potentials* of the solutes are simply the partial derivatives,
$\mu_i - \mu_0 = \left(\frac{\partial g}{\partial w_i}\right)_{T,p,w_{j\neq i}}.$                                                   (B.12)
For mixtures, $n > 1$, the differential (B.6) of the Gibbs function takes the more general form
$\mathrm{d}g = -\eta\mathrm{d}T + v\mathrm{d}p + \sum_{i=1}^{n-1}(\mu_i - \mu_0)\mathrm{d}w_i.$                                        (B.13)
It follows straightforwardly from (B.10), (B.11) that the sum,
$\sum_{i=0}^{n-1} \mu_i\, m_i = m\,g = G,$                                                      (B.14)
equals the Gibbs energy itself (Gibbs 1874-78: eq. (96) therein, Guggenheim 1949, Landau and
Lifschitz 1966, Kittel 1969). In particular, if $n = 1$, the Gibbs function $g$ of a pure substance
represents its chemical potential,
$$g = \mu. \hspace{6cm} \text{(B.15)}$$
Where two phases of a pure substance are in contact at mutual equilibrium, such as saturated water
vapour at the liquid water surface, the mathematically distinct Gibbs functions of those phases take
equal values. This indispensable condition for mutual consistency between the thermodynamic
potentials of TEOS-10 is rigorously obeyed by virtue of appropriate reference-state conditions (Feistel
et al. 2008b).
*Code/Data availability*. TEOS-10 library code used for this paper is available from www.teos-10.org
*Competing interests*. The author has declared that he has no competing interests.
*Acknowledgements.* The author is grateful to Karen Heywood for her kind invitation to write this
Ocean Science Jubilee article. This paper contributes to the tasks of the International Joint
SCOR/IAPWS/IAPSO Committee on the Properties of Seawater (JCS) and was presented at the 18[th]
International Conference on the Properties of Water and Steam (ICPWS) at Boulder, Co. (Feistel
1113 2024).

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
