# Peer review of "Dated 26 July 2024 – submitted to Ocean Science"

_EGUsphere, 2024_

## Referee Comment (RC1)

**Review of the manuscript "TEOS-10 and the Climatic Relevance of Ocean-Atmosphere Interaction" by Rainer Feistel**

This is an important manuscript that should certainly be published. Much of it is a review, and this is well written and is needed, given the large number of papers that have appeared on ocean, ice and air thermodynamics in the past 20 years. This review is also very appropriately submitted to *Ocean Science* since many of the recent papers on ocean thermodynamics have been published in *Ocean Science*.

There is an emphasis in the Abstract and Introduction on the alarming and unexplained global warming of 2023-2024. In the context of an academic review paper on ocean thermodynamics I think that this might be overdone and might make the review seem a bit dated when read in say 10 years' time. I recommend playing this down somewhat by delaying its discussion so that, while this issue is still there to gain the reader's attention, it does not distract from what is essentially a scholarly work.

Despite the theory of thermodynamics being 150 years old, it fell to the author, Rainer Feistel, to accurately derive the melting enthalpy and evaporation enthalpy (the so-called latent heats associated with these phase changes). These formulae were published by Rainer in his 2010 paper in Ocean Science, but since these papers are quite hard to penetrate for the uninitiated, the beauty of Rainer's derivations have been largely unappreciated. The enthalpy of evaporation is derived again in this review as Eqs. (23) and a careful reader can now follow and appreciate all the steps involved.

The review places a lot of emphasis on what thermodynamics alone can say about the causes of the increasing ocean heat content. This part of the review reflect recent research by Rainer Feistel, often published in conjunction with Olaf Hellmuth, which points out if the common assumption of the relative humidity of the atmosphere being constant was just a little in error, then there are rather large consequences for the rate of evaporation, and hence for what is typically meant by the "acceleration of the hydrologic cycle". The inclusion of this material in the review will give this idea much needed exposure to both the observational and the climate modelling community. This part of the review then extends into the trends in global-averaged cloudiness. I think that these topics which explain what thermodynamic theory says about evaporation at the air-sea interface, need to be understood by many more climate scientists, and I am very glad to see them appear in this review paper.

I identify one small part of this manuscript that I think is wrong and should be deleted, namely the part around Eq. (5). I will now describe this before then going on to list rather minor typographical things that might be corrected. Lines 290-307 of the manuscript discuss what is currently used (since TEOS-10) as the definition of ocean heat content (OHC). But then, between lines 308-323 a possible alternative definition of heat content is discussed, namely the one described by Eqn. (5). It is these lines 308-323, and Eq. (5), that I think should be deleted, because Eq. (5) does not describe a quantity that changes only due to the air-sea flux of heat, even for a very shallow ocean. To be specific, let us consider an ocean that is very shallow. In this case the TEOS-10 definition of heat content, Eq. (4), is exact in the sense that its increase or decrease is equal to the area-integrated air-sea flux of heat (air-sea enthalpy flux) [apart from the extra interior Joule heating caused by the dissipation of turbulent kinetic energy]. This is true even though the ocean is being subject to mixing and advection processes continually. Hence, we may speak of the definition Eq. (4) of OHC as being "conservative". However, this is not the case of the variable described by Eq. (5). Rather, this variable is "non-conservative", because even in the absence of any air-sea fluxes anywhere, lateral mixing of different "reference water parcels", each of which have zero entropy, but which have a variety of salinities, will result in an increase in entropy. This increase in entropy of the mixed "reference parcels" would require heat to come from the real seawater parcels in order to bring the "reference water parcels" back to zero entropy. This effect is not trivial. Looking at Figure A.16.1 of the TEOS-10 Manual (IOC et al (2010)), consider the mixing of two fluid parcels that both have zero entropy. To be specific, consider two fluid parcels both at zero Conservative Temperature and zero entropy, but one with Absolute Salinity just slightly greater than zero and the other with S_SO. When equal masses of these two fluid parcels are subject to interior ocean mixing, the well-mixed fluid will have the average Absolute Salinity and the average Conservative Temperature (which is zero) but will have a non-zero entropy (about 0.2K worth of entropy in temperature equivalent units). It is this non-conservative mixing of entropy, even when the entropy of the whole "reference state ocean" has zero entropy, that causes the non-conservation of Eq. (5) and hence invalidates it from being considered as a viable choice for "heat content".

[Figure]

**Figure A.16.1.** Contours (in °C ) of a variable which illustrates the non-conservative production of entropy $\eta$ in the ocean.

Now for the small suggested changes.

Line 12.  Replace "Here, history and properties ..." with "Here, the history and properties ..."

Lines 22-27.  I discourage the inclusion of religion in scientific papers.  First, the bible is not a scientific book, nor is it scientifically correct since its discussion of the arrival of humans on earth (in its first chapter) contradicts the known science of evolution.  Second, scientific papers should be able to be read by authors of all religions without them encountering quotes which somehow endorses the basic textbook of one religion.  Hence, I think that biblical quotes, just like quotes from the textbooks of any religion, should not be allowed in *Ocean Science*.  Please delete these lines.

Line 48.  "and is apparently even counting."  I'm not sure what this means.  Is it a typo?

Line 73. Replace "85% result from" with "85% results from"

Lines 202-203.  Surely the water which IAPWS-95 describes is not Standard Mean Ocean Seawater.

Line 201 and in hundreds of places throughout the paper, Absolute Salinity and Conservative Temperature are used without their upper-case letters.  This goes against what IOC et al. (2010) and Valladares, J., Fennel, W., and Morozov, E.G (2011) and Spall et al (2013) [see below] dictate.  I think the field should stick with the upper-case letters, simply because there are many different possible definitions of absolute salinity and of conservative temperature, but there is only one definition (each) of Absolute Salinity and Conservative Temperature.  Having said that, I have no problem with skipping on the subscript A in the symbol for Absolute Salinity if an author wishes to do so.

Line 244.  Replace "thereof" with "thereby"

Line 272. Replace "entered of left" with "entered or left"

Line 335. Replace "suggests its" with "suggests a"

Line 397.  Replace "eq. (e4.6)" with "eq. (12)"

Line 447. Replace "equating" with "equalling"

Line 654.  Replace "hen-and-egg" with "chicken-and-egg" [since this is the more common expression in English].

Line 688.  Should "minor rest" be "minor residual"?

Line 706.  I think here T_0 is being used as a general reference temperature, rather than 273.15K.  If so, perhaps the subscript could be changed.

Line 811.  "but are, expectedly, also the".  I'm not sure what this means.  Please reword.

Lines 816-817.  Replace "mainly focussed on ocean modelling," with "mainly focussed on ocean observations and modelling,"

Line 857.  Is "Feistel et al. 2008" really "Feistel et al. 2008a"?

Line 896. Replace "derived thereof" with "derived therefrom"

Line 909.  Replace "To the total energy $E$ of the sample," with "To the internal energy $U$ of the sample,"

Line 911.  Replace $E$ with $U$ in this and subsequent equations.  IOC et al (2010) has used $U$ and $u$ for internal energy (extensive and intensive), and this review paper should do the same.

Line 957. Replace "(extensive) energy $E$" with "extensive internal energy $U$"

Line 974, equation (B.10) has a sign error.  The last term should be added, not subtracted.

Line 974, equation (B.10).  The things that are held constant during the differentiation in the last term in this equation are not correct.  They should be the same as the corresponding term in the next equation. Eq. (B.11).

**EDITORIAL**

In 2010, the Intergovernmental Oceanographic Commission (IOC; Chairman Javier Valladares), with the endorsement of the Scientific Committee on Oceanic Research (SCOR; President Wolfgang Fennel) and the International Association for the Physical Sciences of the Oceans (IAPSO; President Eugene G. Morozov), adopted the International Thermodynamic Equation Of Seawater—2010 (TEOS-10) as the official description of seawater and ice properties in marine science, to replace the 1980 International Equation of State of Seawater (EOS-80). The commission has urged all oceanographers to use the new TEOS-10 algorithms and variables to report their work. The TEOS-10 computer software, the official TEOS-10 manual, and other background and explanatory documents are available online (http://www.TEOS-10.org). The advantages of TEOS-10 relative to EOS-80 and guidelines on its usage are contained in those documents.

A prominent part of TEOS-10 is the adoption of a quantity referred to in the standard as "Absolute Salinity" (abbreviated in the standard as "$S_A$") to describe the salinity of seawater. An associated quantity, which replaces potential temperature $\theta$ and accurately describes the heat content per unit mass of seawater, is referred to in the standard as "Conservative Temperature" ("$\Theta$"). The leading uppercase letters in these two terms are a defined and integral part of the printed and approved TEOS-10 standard, as is the roman font for the subscripted A in the Absolute Salinity symbol. To foster proper usage of the TEOS-10 standard and to minimize confusion in the community, the American Meteorological Society (AMS) Publications Department and the field editorial staff of the *Journal of Physical Oceanography* (*JPO*) have opted, for these specific terms, to make an exception to the long-practiced AMS scientific-journal style rules that prohibit capitalization of variable names and stipulate that single-character subscripts will be typeset in italic font. This style policy will take effect with the publication in this issue of *JPO* of the article by Graham and McDougall titled "Quantifying the nonconservative production of Conservative Temperature, potential temperature, and entropy" and will be applied to all AMS journals from that point forward. Prior to now and since its introduction in McDougall's 2003 *JPO* paper, Conservative Temperature had been typeset as "conservative temperature" in AMS journals.

It is hoped that these revisions to our editorial style will help to promote usage of the TEOS-10 standard in the scientific community and will encourage continuing publication of state-of-the-art physical oceanographic research in the AMS journals.

*Michael A. Spall*
Chief Editor

*Karen Heywood, William Kessler, Eric Kunze, Parker MacCready, Jerome A. Smith, and Kevin Speer*
Editors

*Mark E. Fernau*
Managing Technical Editor

DOI: 10.1175/JPO-D-13-082.1

---

## Referee Comment (RC2)

Review of:

**TEOS-10 and the climatic relevance of ocean-atmosphere interaction**
**By Rainer Feistel**

**Summary and recommendation:** The main aim of this paper is to present an overview of some of the main historical developments that led to the new TEOS-10 and discuss some of its applications to examine the potential relevance of irreversible evaporation and changes in the cloud condensation level of cumulus for understanding the anomalous ocean heat uptake associated with global warming. Overall, I found the paper a very interesting and stimulating read that should be eventually published. Before that, however, I have several concerns on several aspects of the paper that I think need to be addressed before the paper can be accepted for publication.

**Major points**

Section 3. Unlike the other sections, this section uses persuasive writing rather than scientific writing to convince the reader of the legitimate and rigorous character of the TEOS10 approach to defining ocean heat content. In essence, this amounts to providing a solution to a question that has not been properly formulated first; as result, the reader is not given the scientific elements necessary to assess the legitimacy of the author's assertions. Moreover, the topic is not properly reviewed or discussed in the context of past research on the issue. As a result, this section does not conform to accepted scientific standards, and therefore should either be significantly improved, or removed from the paper.

Yet, if one looks at the literature, it appears possible to identify the scientific question to be resolved. If one goes back to Bryan (1962), one realises that the problem of how to define heat was originally defined as the problem of how to separate the total energy transport into a dynamical and thermodynamic part. Mathematically, one general way of doing that is by writing the Bernoulli head $B = E_k + \Sigma$, the sum of kinetic energy Ek and static energy $\Sigma + \Phi$

$$E_k + \Sigma = E_k + \Sigma_{dyn} + \Sigma_{heat}$$

Now, the meridional transport of B through some latitude must be balanced by the sum of the wind power input and net heat flux between one pole and the latitude considered. Historically, Bryan (1962) appeared to have considered that the thermodynamic part of the static energy should be defined in terms of the non-elastic part of the internal energy, which is how the quantity $c_{p0}\theta$ is often interpreted (see discussion in Warren 1999). However, while the surface flux of internal energy is the net heat flux Q, this is only accurately the case for $c_{p0}\theta$. If one considers that the goal of the exercise is to define mechanical energy as the quantity absorbing the work transfer by the wind and the heat as the quantity absorbing the heat transfer, then agreed, potential enthalpy

more accurately does so than $c_{p0}\theta$. However, if one considers that part of the heat transfer contributes to the production of mechanical energy (which is equivalent to say that the ocean heat engine has a non-zero thermodynamic efficiency, as predicted by the theory of available potential energy or Carnot heat engine theory, e.g., Tailleux (2010)), then clearly, potential enthalpy is less satisfactory, and can only be regarded as some kind of zero-thermodynamic-efficiency limit or approximation of heat, which should be explicitly stated. The physical basis for decomposing total energy into dynamical and thermodynamic components was recently discussed in Tailleux and Dubos (2024), part of it being rooted in the local theory of available potential energy of Tailleux (2018). To achieve consensus, what is still needed is to agree on the objective criteria that one should use to assess the relative merits of different viewpoints.

I believe that the above presentation is more satisfactory than the one given by the author because: 1) it clearly identifies the scientific question to be resolved; 2) it connects the problem to past and recent research on the topic, by re-situating it in the context in which it was originally developed; 3) it gives the reader the necessary scientific elements for assessing the relative merits of different viewpoints on the matter. In contrast, TEOS-10 or the author's section gives the impression that there is only a unique way to address the problem and that there is nothing left to be solved, when this is clearly not the case.

**Minor points**

Line 65. Typically, present numerical climate models suffer from an "ocean heat budget closure problem" (Josey et al. 1999) and describe the m–2 m–2 ocean-atmosphere heat flux only to within uncertainties between 10 W and 30 W (Josey et al. 2013).
I find this statement confusing because my understanding of the Josey et al papers relate to the `observational' closure problem arising from the technical difficulties of measuring the different heat fluxes component reliably enough and with the desired accuracy. The closure problem in numerical ocean models is a completely different thing. Numerical ocean models will in general exhibit drift depending on many different factors, such as model resolution, and various model errors. The author needs to review the literature more carefully to avoid confusing observational and modelling issues.

Lines 70-72. While that may be the case, countless climate projections have been published that reproduce ocean warming like that observed. Presumably, air-sea interactions in such simulations have been analysed. It would therefore be useful if the author could summarise the state of knowledge on the matter, including discussions of the nature of uncertainties, rather than just speculate on the matter.

Lines 78-80: it would be useful to the reader if the author could translate these numbers in terms of implied change in net evaporation or precipitation, assumes that the two balance on average. May be the author could also discuss the fact that global warming is expected to heat up land area faster than ocean area. As a result, this may decrease

relative humidity, with a possible compensating effect over the ocean like the one suggested by the author.

Line 95. About modelling the global heat engine. I agree with the author that improved thermodynamic formulations are useful to that end. Note, however, that a key part of understanding the functioning of a heat engine is to identify the relative fraction of the heat transfer going into driving the dynamics (the thermodynamic efficiency) compared to that passively as heat. It seems to me that while TEOS10 is clearly a success in providing such improved formulations, it is unclear how it can claim to contribute to the understanding of the functioning of the ocean heat engine. Indeed, by assuming that all the heat transfer into the ocean goes into heat, with none contributing to the dynamics, TEOS10 implicitly assumes that that the thermodynamic efficiency of the ocean engine is zero, which is inconsistent with studies such as Tailleux (2010) and many others. Moreover, if atmospheric scientists had a way of defining atmospheric heat as proposed by TEOS-10 in terms of a variable absorbing all heat transfer, then this would also imply a zero thermodynamic efficiency for the atmospheric heat engine, which I am not sure would be very popular.

Figures 3 and 4. Shouldn't credit or copyright for the photo be indicated? Can these be re-used by others?

Lines 129-134. The question is whether the TEOS-10 definition of heat is as rigorous as the author claims, as the definition seems an ad-hoc one to me. TEOS-10 proposes a solution to a question that they never define in the first place. See my comments in the major points section.

Line 200-203. Can the author provide some explanation about why a Helmholtz potential is preferred in that case rather than a Gibbs function? The use of a Gibbs function as the basis for TEOS10 is generally understood from the fact that S, T, and p are variables that are the most easily measured/fixed in practice. We are also told that density is a variable that is very hard to measure in practice, which makes the usefulness of a Helmholtz function hard to understand. So, what are the physical arguments in favour of it?

Line 259-260. Preferred by whom? Many scientists consider that the choice of prognostic variable is a matter of personal preference and essentially subjective. Anybody trained as a physicist will prefer to use a variable that is as close as possible to measured or measurable quantities, which is what most physicists consider to be the best practice. Conservative Temperature may have some desirable features, but it has many undesirable ones as well, as it remains a non-measurable ad-hoc energy-like quantity that does not separate thermal from saline effects as well as potential temperature. It remains a puzzle to me why TEOS-10 found the need to legislate on an essentially subjective matter when potential temperature is clearly advantageous to Conservative Temperature in many important and fundamental ways. Atmospheric scientists retain absolute freedom in using whatever potential temperature variable they want, and many have been developed, the jury being still out on the relative merits of each one. Why should oceanographers have less freedom than atmospheric

scientists to choose whatever they consider to be best for their own applications? In this regard, TEOS-10 feels very autocratic.

Line 358, Equation 6: Can you be more specific as to the form of the transfer coefficient Df(u) by providing examples from the literature? I am confused by the author's statement that such a coefficient only depends on u, because my understanding is that such a coefficient also depends on many other things, such as a sea surface roughness, nature of the boundary layer, and so on...

Lines 396-398. This sounds like an important result warranting further attention. However, can the author guarantee that Dq(u) does not depend indirectly on q in a way that would compensate the effect discussed? Change in q may modify the nature of the turbulent boundary layer and the transfer coefficient.

Lines 637. The author only discusses irreversibility associated with non-zero relative humidity under the assumption that the oceans and atmosphere have the same temperature. In reality, the latter may also have different temperatures. Can the author comment as to the implications that this would have for his theory?

Lines 723-725. My understanding is that the Zlcl is to be obtained by integrating the hydrostatic relationship, which can only lead to the author's formula (52) if the entropy and specific humidity are perfectly uniform from the surface to the bottom of the cloud. Is that really the case in reality?

**Appendix**

Lines 930-934. I am surprised to see the quantities $-pdV$ and $Td\eta$ equated with the work and heat transfers $\delta W$ and $\delta Q$, because this is only true for reversible and quasi-static transfers. As far as I am aware, the exact relations are $T\,d\eta \geq \delta Q$ and $-p\,dV \leq \delta W$. This can be verified for an adiabatic expansion of a piston in a vacuum. In that case, $\delta Q = 0$ yet the entropy increase; moreover, $\delta W = 0$, yet V increases so that $-p\,dV < 0$. Moreover, note that p, V, T and eta relates to internal properties of the fluid, while the concepts of heat and work transfers relate to external properties describing the interactions of the fluid with its environment, so that it is dangerous and confusing to equate internal and external properties without further discussion.

Lines 962-963. I thought that this condition was also true in the presence of gravity. Can the author explain how gravity affects these conditions, given that this is obviously relevant to the oceanic case?

**References**

Bryan, K. (1962). Measurements of meridional heat transport by ocean currents. JGR, 67, 3403—3414.

Tailleux, R. (2010). Entropy versus APE production: on the buoyancy power input in the oceans energy cycle. GRL. https://doi.org/10.1029/2010GL044962

Tailleux, R. (2018) Local available energetics of multicomponent compressible stratified fluids. JFM. 842, https://doi.org/10.1017/jfm.2018.196

Tailleux, R. and T. Dubos (2024). A Simple and transparent method for improving the energetics and thermodynamics of seawater approximations: Static energy asymptotics (SEA). Ocean Modelling. https://doi.org/10.1016/j.ocemod.2024.102339

Warran, B. (1999). Approximating the energy transport across oceanic sections. JGR, 140, 7915—7919.

---

## Community Comment (CC1)

TABLE 12-3. THE ENTROPIES OF SOME COMMON SUBSTANCES AT 298.15°K, CAL/DEG MOLE†

Solids:

| | |
|---|---|
| Ag | 10.20 |
| AgCl | 23.00 |
| AgBr | 25.60 |
| AgI | 27.6 |
| Al | 6.77 |
| As | 8.4 |
| C(gr) | 1.37 |
| Ca | 9.95 |
| CaO | 9.5 |
| Cd | 12.37 |
| Cu | 7.97 |
| Fe | 6.49 |
| $I_2$ | 27.76 |
| S(rh) | 7.62 |
| Si | 4.51 |
| $SiO_2$(q) | 10.00 |

Liquids:

| | |
|---|---|
| $Br_2$ | 36.4 |
| $HNO_3$ | 37.19 |
| $H_2O$ | 16.73 |
| Hg | 18.17 |

Gases:

| | |
|---|---|
| $CH_4$ | 44.47 |
| CO | 47.20 |
| $CO_2$ | 51.08 |
| $Cl_2$ | 53.29 |
| $F_2$ | 48.49 |
| $H_2$ | 31.21 |
| HCl | 44.64 |
| $H_2S$ | 49.13 |
| $N_2$ | 45.77 |
| NO | 50.34 |
| $O_2$ | 49.01 |

† See K. K. Kelley, *U.S. Bur. Mines Bull.* 477, 1950, and revision by K. K. Kelley and E. G. King, *U.S. Bur. Mines Bull.* 592, in press, for a critical summary of entropy data and references to original sources.

(Lewis and Randall, 1961, p.137)

TABLE 27-1. ENTROPIES OF MONATOMIC GASES AT 298.15°K†

| | $\int C_P\, d \ln T$ | Eq. (27-2) |
|---|---|---|
| Ne | $35.01 \pm 0.10$ | $34.95 \pm 0.01$ |
| Ar | $36.95 \pm 0.2$ | $36.99 \pm 0.01$ |
| Kr | $39.17 \pm 0.1$ | $39.20 \pm 0.01$ |
| Xe | $40.7 \pm 0.3$ | $40.54 \pm 0.01$ |

† K. K. Kelly, *U.S. Bur. Mines Bull.* 477, 1950, reviews critically the available data and gives original references.

(Lewis and Randall, 1961, p.421)

Figure 15: *The molar entropies $\overline{S}^o$ at 25°C for liquid water ($H_2O$) and the main atmospheric gases ($N_2$, $O_2$, $CO_2$, Ar) published by Lewis and Randall (1961).* **Top:** *the Table 12-3 of (p.137) for the molar entropy of liquid water $H_2O$ ($16.73 \times 4.184 \approx 70.00\ J\ K^{-1}\ mol^{-1}$), together with the molar entropy for three of the main dry-air atmospheric gases ($N_2$, $O_2$, $CO_2$).* **Bottom:** *the Table 27-1 (p.421) for the molar entropy of the Argon noble gaz ($36.99 \times 4.184 \approx 154.77 \pm 0.8\ J\ K^{-1}\ mol^{-1}$).*

TABLE 25-7. THERMODYNAMIC DATA FOR AQUEOUS IONS AT 298.15°K, CAL/DEG MOLE OR KCAL/MOLE

| Ion | $\bar{C}_P^o$ | $\bar{S}^o$ | $\Delta \bar{H}_f^o$ | $\Delta \bar{F}_f^o$ |
|---|---|---|---|---|
| H+ | 0 | 0 | 0 | 0 |
| OH⁻ | −32.0 | −2.52 | −54.96 | −37.59 |
| F⁻ | −29.5 | −2.3 | −78.66 | −66.08 |
| Cl⁻ | −30.0 | 13.2 | −40.02 | −31.35 |
| ClO2⁻ | | 24.1 | −17.18 | 2.74 |
| ClO3⁻ | −18 | 39 | −23.5 | −0.6 |
| ClO4⁻ | | 43.2 | −31.41 | −2.47 |
| Br⁻ | −31 | 19.29 | −28.90 | −24.57 |
| I⁻ | −31 | 26.14 | −13.37 | −12.35 |
| I3⁻ | | 57.1 | −12.4 | −12.31 |
| S⁻ | | −4 | 7.8 | 20.6 |
| HS⁻ | | 15.0 | −4.10 | 3.00 |
| SO4⁻⁻ | −66 | 4.1 | −216.90 | −177.34 |
| HSO4⁻ | | 30.52 | −211.70 | −179.94 |
| SeO3⁻ | | 3.9 | −122.39 | −89.33 |
| SeO4⁻ | | 5.7 | −145.3 | −105.42 |
| HSeO4⁻ | | 22.0 | −143.1 | −108.2 |
| NH4+ | 16.9 | 26.97 | −31.74 | −19.00 |
| N2H6++ | | 19 | −4 | 22.5 |
| N2H5+ | | 31 | −1.7 | 21.0 |
| NH2OH2+ | | 37 | −30.7 | −13.54 |
| NO2⁻ | | 29.9 | −25.4 | −8.25 |
| NO3⁻ | −18 | 35.0 | −49.37 | −26.43 |
| PO4³⁻ | | −52 | −306.9 | −245.1 |
| HPO4⁻ | | −8.6 | −310.4 | −261.5 |
| H2PO4⁻ | | 21.3 | −311.3 | −271.3 |
| HAsO4⁻ | | 0.9 | −214.8 | −169 |
| H2AsO4⁻ | | 28 | −216.2 | −178.9 |
| HCOO⁻ | | 21.9 | −98.0 | −80.0 |
| HCO3⁻ | | 22.7 | −165.18 | −140.31 |
| CO3⁻ | | −12.7 | −161.63 | −126.22 |
| CH3COO⁻ | | | | |

(Lewis and Randall, 1961, p.400)

TABLE 25-7. THERMODYNAMIC DATA FOR AQUEOUS IONS AT 298.15°K, CAL/DEG MOLE OR KCAL/MOLE (Continued)

| Ion | $\bar{C}_P^o$ | $\bar{S}^o$ | $\Delta \bar{H}_f^o$ | $\Delta \bar{F}_f^o$ |
|---|---|---|---|---|
| PtCl4⁻ | | 42 | −123.4 | −91.9 |
| PtCl6⁻ | | 52.6 | −167.4 | −123.1 |
| Fe++ | | −27.1 | −21.0 | −20.30 |
| Fe3+ | | −70.1 | −11.4 | −2.53 |
| Fe(OH)++ | | −23.2 | −67.4 | −55.91 |
| FeNO++ | | −10.6 | −9.7 | 1.5 |
| Mn++ | | −20 | −53.3 | −54.4 |
| MnO4⁻ | | 45.4 | −129.7 | −107.4 |
| H2BO3⁻ | | 7.3 | −251.8 | −217.6 |
| BF4⁻ | | 40 | −365 | −343 |
| Al3+ | | −74.9 | −125.4 | −115 |
| Gd3+ | | −43 | −168.8 | −165.8 |
| Mg++ | 4 | −28.2 | −110.41 | −108.99 |
| Ca++ | −9 | −13.2 | −129.77 | −132.18 |
| Sr++ | | −9.4 | −130.38 | −133.2 |
| Ba++ | −11 | 3 | −128.67 | −134.0 |
| Li+ | 14.2 | 3.4 | −66.55 | −70.22 |
| Na+ | 7.9 | 14.4 | −57.28 | −62.59 |
| K+ | 2.3 | 24.5 | −60.04 | −67.46 |
| Rb+ | −8.7 | 28.7 | −59.4 | −67.65 |
| Cs+ | −18.7 | 31.8 | −62.6 | −70.8 |
| UO2++ | | −17 | −250.4 | −236.4 |

(Lewis and Randall, 1961, p.401)

TABLE X

Calculation of the Partial Molal Entropy of Sea Salt at 25°C (a)

| Species | $S_i^o$ | $N_i$ | $N_i \bar{S}_i^o$ |
|---|---|---|---|
| Na+ | 60.2 | 0.83619 | 50.34 |
| Mg2+ | −118.0 | 0.09509 | −11.22 |
| Ca2+ | −55.2 | 0.01834 | −1.01 |
| K+ | 102.5 | 0.01822 | 1.87 |
| Sr2+ | −39.3 | 0.00016 | −0.01 |
| Cl⁻ | 55.2 | 0.97461 | 53.81 |
| SO4²⁻ | 17.2 | 0.05042 | 0.87 |
| HCO3⁻ | 95.0 | 0.00345 | 0.33 |
| Br⁻ | 80.8 | 0.00151 | 0.01 |
| B(OH)3 | ∼50 | 0.0006 | 0.03 |
| CO3²⁻ | −53.1 | 0.0004 | −0.02 |
| B(OH)4⁻ | ∼100 | 0.00015 | 0.01 |
| F⁻ | −9.6 | 0.0001 | −0.001 |
| | | | 95.01 |

(a) Based on $\bar{S}^o(H^+) = 0$ and taken from Lewis and Randall (1961). The values of B(OH)3 and B(OH)4⁻ have been estimated by comparison with solutes of similar size and charge.

(Millero, 1983, p.35)

Table 3.3

Conventional Ionic Entropies at 25°C (298·16°K), computed relative to $\bar{S}_{H^+}^o = 0$ in the hypothetical standard state of one gram-ion per kg of water.

| Ion | $\bar{S}^o$ cal deg⁻¹ mole⁻¹ | Ion | $\bar{S}^o$ cal deg⁻¹ mole⁻¹ | Ion | $\bar{S}^o$ cal deg⁻¹ mole⁻¹ |
|---|---|---|---|---|---|
| H+ | (0·00) | Mg++ | −28·2 | Al+++ | −74·9 |
| Li+ | 3·4 | Ca++ | −13·2 | Cr+++ | −73·5 |
| Na+ | 14·4 | Sr++ | −9·4 | Fe+++ | −70·1 |
| K+ | 24·5 | Ba++ | 3·0 | Ga+++ | −83 |
| Rb+ | 29·7 | Mn++ | −20 | In+++ | −62 |
| Cs+ | 31·8 | Fe++ | −27·1 | Gd+++ | −43 |
| Tl+ | 30·4 | Cu++ | −23·6 | U+++ | −36 |
| Ag+ | 17·67 | Zn++ | −25·45 | Pu+++ | −39 |
| F⁻ | −2·3 | Cd++ | −14·6 | U++++ | −78 |
| Cl⁻ | 13·17 | Sn++ | −5·9 | Pu++++ | −87 |
| Br⁻ | 19·25 | Hg++ | −5·4 | | |
| I⁻ | 26·14 | Pb++ | 5·1 | | |
| OH⁻ | −2·5 | S⁻⁻ | −6·4 | | |
| SH⁻ | 14·9 | | | | |

Data from POWELL, R. E. and LATIMER, W. M., *J. chem. Phys.*, 19 (1951) 1139.

(Robinson and Stokes, 1970, 1955, p.67)

Figure 16: **Top:** *The molar entropy* $\bar{S}^o$ *and specific heat at constant pressure* $\bar{C}_p^o$ *listed in the Tables 25-7 of Lewis and Randall (1961, p.400-401) for most of the sea-salt anions and cations (at 298.15 K and relative to* $\bar{S}_{H^+}^o = 0$). **Bottom-left:** *The same sea-salts entropies as in Lewis and Randall (1961, p.400-401), but used in Millero (1983, p.35) with 1 cal = 4.184 J;* **Bottom-right:** *values of the "Conventional Ionic Entropies" (at 298.16 K and relative to* $\bar{S}_{H^+}^o = 0$) *given in the Table 3.3 of Robinson and Stokes (1970, 1955, p.67) and given for most of the sea-salt ions (in particular* $Na^+$, $K^+$, $Cl^-$, $Br^-$, $Mg^{2+}$) *from Powell and Latimer (1951).*

[Figure]

**Figure 1.** Temperature (T) and salinity (S) profiles from SCICEX'96, cast 43

Figure 17: *A study of some thermodynamic properties of the SCICEX'96 (cast 43) CTD vertical profiles available on* `https://www.nodc.noaa.gov/archive/arc0021/0000568/1.1/data/0-data/SCICEX-96/Exported%20Data/CTD043.EDF` *and corresponding to the Fig. 1 of Steele et al. (2004) with the draft profiles recalled in the top right panel.*

[Figure]

Figure 18: *A study of the (std and abs) seawater entropies for 6 CTD profiles available: 1) for "CTD-1-arctic" "CTD-2-arctic" "CTD-3-arctic" and "CTD-1" profiles in the subroutine* `gsw_mod_check_data.f90` *in Feistel and TEOS10 (2010) software; 2) for "CTD-E88-61" and "CTD-E88-64" in the document UNESCO-JPOTS (1991, August 1982 tropical profiles, Endeavour cruise 88, stations 61 and 64, Tables 2.1, 2.2, 2.3, p.49, 51, 53).*

[Figure]

Figure 19: *The "bulk salinity parts" of the seawater entropies for the same vertical profiles shown in the Figs. 5, computed by removing at each level the quantity* $4217.4 \times \ln\left[\left(t + 273.15\right) / 273.15\right]$*, namely* $c_w \ln\left(T / T_0\right)$*.*

[Figure]

Figure 20: *The vertical profiles (from 0 to 800 m depth) for the temperature (t, in °C), the absolute salinity ($S_A = 1.00488 \times S_p$), and the "bulk salinity parts" of the seawater entropies for both the standard and absolute (with a threshold of $-1$) versions of TEOS10 computed from the Fig. 19 (p.79) of Sverdrup et al. (1942) and the dataset recalled (up to 800 m depth) in the Table 8.*

[Figure]

Figure 21: *A study of the Salinity parts of the seawater entropies for various formulations.*

[Figure]

Figure 22: *Entropies for dry-air ($N_2$, $O_2$, Ar, $CO_2$ and water $H_2O$) species plotted against the absolute temperature and computed at $1000\,\text{hPa}$. The calorimetric method $\left(\int_0^T c_p(T')\,d\ln(T') + \sum_j L(T_j)/T_j\right)$ corresponds to the coloured solid lines. The third-law hypothesis is applied at $0\,K$ with zero entropies for all the solid phases, but with the residual entropy of $189\,\text{J kg}^{-1}\,\text{K}^{-1}$ for ice-Ih. The vertical jumps correspond to phase changes at $T_j$ with the phase-change enthalpies $L(T_j)$ between solids phases (for $N_2$ and $O_2$), then from solid to liquid phases, then from liquid to vapour phases. The statistical-physics values (black dashed lines) are computed from $S = k\,\ln(W)$ and $F = -\,k\,T\,\ln(Z)$ for the vapour phases according to the method described in Chase (1998) for translational, rotational, vibrational and electronic partition functions (Z).*

[Figure]

Figure 23: *A study of thermodynamic vertical profiles computed from observations collected during the DYCOMS-II (RF01) Stratocumulus (from the Figs. 1 of Zhu et al., 2005, p.2742, see the dataset in the Table 10).* **(a):** *for the specific water contents in g kg⁻¹ ($q_t = q_v + q_l$, $q_v$ and $10 \times q_l$);* **(b):** *for the absolute temperature in K (T, in green), the dry-air potential temperature in K ($\theta$, in violet) and other moist-air potential temperatures in K (absolute in red and orange, "liquid-water" in black, "equivalent" in blue);* **(c):** *for several moist-air entropies in J K⁻¹ kg⁻¹ (absolute in red and orange, "liquid-water" in black, "equivalent" in blue, with the corresponding TEOS10 values in grey and green).*

[Figure]

Figure 24: *The Fig. 5 of Marquet and Stevens (2022, p.1099). Vertical profiles of $\theta_e$, $\theta_l$, and $\theta_s$ plotted for half of the observed sounding of the first ASTEX Lagrangian experiment (with a shift of 2 K or 2 g/kg between each profiles). Stratocumulus (Sc) profiles are colored blue, whereas cumulus (Cu) profiles are colored black. Transition profiles between the two regimes are colored red, with the purple arrow indicating the deepening of the PBL associated with the transition. The green arrows show the sign of the top-PBL jump for each variable and for each regime: positive if tilted to the right, null if vertical, negative otherwise. The blue and red dashed boxes have been added to highlight the isentropic regions where $\theta_s$ (and not $\theta_e$) is constant despite the opposite vertical gradients in $\theta_l$ and $q_t$ which compensate with the special value of $\lambda_r$ given by the third law.*

[Figure]

Figure 25: *A study of the Figs. 3 and 4 of Feistel et al. (2010a, p.101-103)...*

[Figure]

Figure 26: *A study of the Figs. 19 and 20 of Feistel et al. (2010a, p.117-118)...*

---

## Community Comment (CC2)

[supplement omitted: unrelated document]

---

## Community Comment (CC3)

Comments of O. Hellmuth on
**"TEOS-10 and the Climatic Relevance of Ocean-Atmosphere Interaction"**
**by Rainer Feistel**
(https://doi.org/10.5194/egusphere-2024-1243, Preprint)

The author of the paper is member of a very productive and powerful community which established a new international seawater standard, called TEOS-10. As can be seen from the reference list of this study, the author itself serves as the leading author of many of the TEOS-10 relevant papers and can be considered as the main driver of this development. The adoption TEOS-10 by the responsible international authorities is a result of intensive scientific research over a period of more than a decade (starting in 2004) including its technical implementation according to the high-quality evaluation standards of IAPWS. TEOS-10 can be considered as a real gem of both oceanographic and atmospheric thermodynamics.

Based on a review of present challenges in the description of ocean-atmosphere interactions the author identified a number of unresolved open questions to which TEOS-10 can provide an important contribution.

Of top priority for the fate of Earth's climate is the diagnosis of a hitherto unexplained ocean warming rate that is equivalent to an excess heat flux of about 1.3 W m$^{-2}$. The key problem in the elucidation of the physical reason for this excess heat flux is of metrological origin: for the time being the uncertainties (noise) in the heat flux estimation is by at least one order of magnitude larger than the physical signal. Therefore, root-cause analysis based on empirical data must be replaced by model-based deductions. This cumbersome situation will hold on until the required degree of metrolocigal certainty in the heat flux determination is achieved. Based on my own examinations of the authors considerations on the role of the relative humidity (RH) in the marine surface layer, I confirm authors estimation that a tiny increase in RH of only 0.2 %rh might be suffice to reduce the ocean evaporation by the observed 1.3 W m$^{-2}$ and, as a consequence, to increase the ocean heat content. Feistel's consideration is based on a generalised Dalton equation (see Section 4) which uses a physically rigorous formulation of the thermodynamic driving force but still inheres several approximations of the "kinetic" prefactor in Daltons formula (see the empirical transfer coefficient as a function of wind speed in Eq. (6)). Classical approaches of the heat flux estimation relying on the Monin-Obukhov similarity theory in combination with a sophisticated treatment of the molecular layer at the ocean-atmosphere interface such as the renewal theory (e.g., Liu et al. 1979) will further complicate the situation: next to variations in the marine-surface layer RH also air and water temperature as well as wind variations may essentially contribute to the total heat flux uncertainty. A dedicated flux intercomparison analysis between the generalised Dalton approach, Eq. (6), and classical approaches is recommendable for subsequent studies.

Owing to its large heat capacity, the ocean serves as a superthermostat ensuring habitable temperatures on Earth. However, due to the superposition of multiscale waves in the ocean, the thermostatic performance of the ocean appears rather "erratic" which has implications, e.g., for the near-surface bucket temperature and the SST. As ocean currents are driven by atmospheric motions, changes in the atmospheric circulation (wind fields) may have a large impact on the water temperature and so on oceanic evaporation too. For me there is no doubt that a refined treatment of the ocean-atmosphere heat transfer is one of the keys to the elucidation of the reasons for the sudden ocean warming.

The author initiated and significantly contributed to the formulation of a roadmap toward a generalised SI-compatible RH definition which is based on rigorous thermodynamic notions. These basic notions are the four thermodynamic potentials presented in Section 2, allowing for a self-consistent formulation of the mutual thermodynamic equilibrium conditions for water in humid air, seawater and ice. A key role in the proposed generalised definition of RH plays the concept of fugacity. This notion allows a highly accurate full range-definition of RH in terms of relative fugacity with consideration of real gas effects. The classical formula for RH such as employed by WMO can be recovered from the relative fugacity for the limit of atmospheric pressure (ideal-gas limit). Even though the scientific background of relative fugacity and their approximations for practical use at atmospheric pressure were completely worked out and published in a series of peer-reviewed papers over many years, the formal adoption as a SI standard by the responsible international authorities remained, unfortunately, unrequited so far. The realisation of the final step of this roadmap remains desirable. I do not say a better way to define RH than by relative fugacity.

In Section 5.2 the author mentioned that evaporating sea spray droplets will "rather develop into a floating persistent Köhler (1936) equilibrium between droplet size, droplet salinity and ambient relative fugacity. This equilibrium can be described by the TEOS-10 model of sea air if the additional Kelvin pressure caused by the surface tension is allowed for." Just for authors information: more than a decade ago my former colleague Prof. A. K. Shchekin from Fock Institute of State University of St. Petersburg and his co-workers developed a full thermodynamic theory to describe the growth of soluble nanoparticles in a solvent vapor. The description is based on a generalisation of the Gibbs-Kelvin-Köhler theory of condensation and a generalisation of the Ostwald-Freundlich theory of solutions with consideration of the thermomechanic concept of disjoining pressure. Lateron this very sophisticated theory was extented to non-ideal solutions, combined with the theory of homogeneous salt crystallisation, and applied to the evolution of sodium chloride particles (as a proxy for sea spray) in ambient humid air (Hellmuth and Shchekin 2015, see references therein). This theory allows a full description of the size evolution of such hygroscopic particles from the dry initial state until the deliquescence threshold and back to the efflorescence threshold with disclosure of hysteresis effects. Independent of Rainer Feistels suggestions, the availability of TEOS-10 opens the challanging way to revise the approach of Hellmuth and Shchekin within the framework of a sensitivity analysis by replacing several properties of the theory with TEOS-10 based properties, among others by replacement of classical RH definition with the relative fugacity.

With reference to radiation models the author argued in Section 6 that on the global average the short-wave and long-wave cloud radiative effects cancel each other almost completely up to minor rest of $-1$ mW m$^{-2}$ yr$^{-1}$, so that the observed continuously shrinking cloudiness (see Fig. 13) may be assumed to have practically no net effect on the ocean's radiation balance. Feistel added: "However, more detailed investigations in the future may reveal more rigorous results for the ocean than this simplified picture." In the context of authors argumentation it might be helpful to consider also a very recent study of Luo et al. (2024), who showed that in a warming climate low-level cloudiness exhibits diurnally asymmetric trends. According to them cloud fraction decreases more during the day than at night. Based on climate modelling, the diurnally asymmetric cloud cover variation was argued to be driven by trends in the lower tropospheric stability as a consequence of increasing greenhouse gas concentrations. This

asymmetry turns out to be an amplifier of surface warming, by both decreasing the daytime cloud shortwave albedo effect and increasing the nighttime cloud longwave greenhouse effect. To complete the picture it might be useful to give reference to this study too.

In view of the high number of peer-reviewed TEOS-10 relevant papers which reflect the historical evolution of the development of this standard, the entry to this extraordinary achievement by different users is not easy. Although every physical detail is on the table, the time seems to be ripe for a user-friendly handbook to support the further application of TEOS-10 especially in the atmospheric community.

I found the study of Feistel very comprehensive and instructive and would like to see it accepted by the referees and the editor. My congrats to this further building stone in the TEOS-10 housing.

Technical

- Line 397: eq. (e4.6) is undefined

**References**

Hellmuth, O. and A. K. Shchekin, 2015: Determination of interfacial parameters of a soluble particle in a nonideal solution from measured deliquescence and efflorescence humidities. Atmos. Chem. Phys., 15, 3851–3871, www.atmos-chem-phys.net/15/3851/2015/ doi:10.5194/acp-15-3851-2015

Liu, W. T., K. B. Katsaros, and J. A. Businger, 1979: Bulk parameterization of air–sea exchanges of heat and water vapor including the molecular constraints at the interface. J. Atmos. Sci., 36, 1722–1735, doi:10.1175/1520-0469(1979)036<1722:BPOASE>2.0.CO;2, URL https://journals.ametsoc.org/view/journals/atsc/36/9/1520-0469_1979_036_ 1722_bpoase_2_0_co_2.xml.

Luo, H., J. Quaas, Y. Han, 2024: Diurnally asymmetric cloud cover trends amplify greenhouse warming. Science Advances 10, eado5179 (2024)

---

## Author Comment (AC4)

**Final reply to the comments** on:

"TEOS-10 and the Climatic Relevance of Ocean-Atmosphere Interaction"

submitted to Ocean Science 2024 by Rainer Feistel

In the **review of Trevor McDougall**, the main criticism was the request to remove from this paper the Section on Potential Enthalpy and Ocean Heat Content. From my perspective, this was a misunderstanding, and as a consequence, this section has been enlarged in order to better explain what it is intended for.

Already in 1888, in his "Theory of Heat", J.C. Maxwell had clearly stated that "We have … a right to speak of heat as a measurable quantity, … however, … we have no right to treat heat as a substance". In the recent oceanographic and climatological literature, terms like "ocean heat content" are frequently used, giving the impression that "heat content" is something of the same kind as "salt content", "water content" or "$CO_2$ content" of the ocean. But, thermodynamically, this impression is definitely wrong.

What can properly be measured, in principle, is the amount of heat that goes into or out of the ocean across its interface with the outside world. "Heat content" may be defined by a specific process that transforms the ocean from a certain reference state to the state of interest. Different processes carried out between those states may be associated with different amounts of heat. Remember that heat engines are systems that are permanently supplied with heat even though they return periodically to the same state over and over again. State quantities, by definition, take the same values again if a system returns to its previous state. There cannot exist any ocean state quantity that may properly be identified with the thermodynamic quantity "heat".

Section 3 of this paper presents a conceptual proposal for defining a reference state and a measurable heat exchange process of the ocean which is consistent with the common understanding of "heat content" in the context of TEOS-10. This proposal should be understood as an additional physical justification of the current formal mathematical definitions of "heat content".

The **review of Remi Tailleux** raises a number of specific questions which have already been addressed in the direct response. In addition to that, a key issue again is the definition of heat, its uniqueness and way of description in oceanography. With this respect, I like to refer to the above response to Trevor McDougall's comments.

The way "heat content" is discussed in this paper is certainly only one possible option of doing so. It is the aim here to raise awareness of the fundamental character and ambiguity of the heat problem involved, and to offer a specific proposal as a suggested solution consistent with the common published definitions of "heat content" in the context of TEOS-10.

The **comments of Pasqual Marquet** have the form of an extended counter publication. The numerous technical arguments raised there must be left to be discussed in detail by the scientific community. None of those, however, is capable of rebutting the general physical key statement that residual entropies are neither available from thermodynamic measurements, nor do their quantitative values affect any results of thermodynamic measurements.

Regarding TEOS-10, note that:

No user of TEOS-10 is committed to work with the reference state definitions actually implemented. TEOS-10 equations and source code are open and well documented. It takes only a few numerical

constants to be modified, mutually consistently, in order to install arbitrary other residual entropies. This will not affect, though, results for any measurable thermodynamic properties of the climate system, and is simply unnecessary therefore. However, care must be taken with respect to special quantities such as Ocean Heat Content or Conservative Temperature which are presently defined under the assumption that the enthalpy of the standard ocean state is zero by definition of the TEOS-10 reference state conditions. The current choice of TEOS-10 reference state conditions is optimum with respect to uncertainties and has been supported by the expert group of TEOS-10 developers as well as various IAPWS experts.

Some key arguments are:

- No scientific study has ever revealed a climatic relevance of residual entropies in ocean-atmosphere interaction

- No atmospheric measurement has ever revealed the exact value of the residual entropy of ice Ih

- No atmospheric measurement has ever revealed whether ice Ih (with residual entropy) or ice XI (without) is the proper zero-point equilibrium phase of water

- No technical or scientific application of the IAPWS-95 equation has ever been reported to be quantitatively in conflict with the IAPWS reference state conditions

- Among all the experimental thermodynamic data available and exploited for the development of TEOS-10, none of those permitted the determination of the adjustable coefficients representing the absolute entropies of the substances involved

- Only little is known about possible residual entropies of the various substances contained in dry air and dissolved sea salt. Published "standard molar entropies" may be considered as agreed reference state definitions as their assumed perfect equilibrium state at 0 K is usually not exactly known (and unnecessary to be exactly known)

- Clausius' empirical entropy definition was derived from cyclic processes which permit an arbitrary additive constant of entropy. Heat exchange is defined in terms of entropy differences only

- Isentropic parcel trajectories defined by $S(\mathbf{x}) = S(\mathbf{x}_0)$ remain the same as $S'(\mathbf{x}) = S'(\mathbf{x}_0)$ under the transformation $S' = S + \text{const}$. Such trajectories are physically meaningful only if along the trajectory the parcel does not exchange matter with its surrounding

- Residual entropy has only theoretically been concluded from the equation $S = k \log W$ of the statistical model of Boltzmann, Planck and Pauling, which requires counting of theoretically possible alternative microscopic (molecular) configurations that are consistent with exactly one and the same given macroscopically observed state

Thankfully, the **comments of Olaf Hellmuth** offer valuable additional aspects of the submitted paper.

- Some references to reviews of MOST have been added to the context of eq. (6)

- Reference to Hellmuth & Shchekin (2015) has been added in Section 5.2

- Work of Luo et al. (2024) has additionally been mentioned in Section 6.3

---

## Author Response (AR1)

**Point by point response to reviewer comments**

**Reviewer 1:**

**RF:** Thanks for indicating the various minor issues. They have been fixed.

1. **Recent warming 2023**

**Review 1:** „There is an emphasis in the Abstract and Introduction on the alarming and unexplained global warming of 2023-2024. In the context of an academic review paper on ocean thermodynamics I think that this might be overdone and might make the review seem a bit dated when read in say 10 years' time. I recommend playing this down somewhat by delaying its discussion so that, while this issue is still there to gain the reader's attention, it does not distract from what is essentially a scholarly work."

**RF:** The 2023 warming is taken as a current example for the urgency and importance of the questions raised. This "teaser" is presented as just a single quotation. To make this more obvious, the section has been split into 2 paragraphs:

"…the reported ocean's average warming rate amounts to 1.3 W m$^{-2}$, and is apparently even increasing.

The currently observed *ocean heat content* (OHC) represents a merely transient maximum after a decade-long systematic warming process in the past, see Fig. 18 in **Section 6**, which may proceed to even higher values in the future. In **Section 3**, thermodynamic aspects of related OHC definitions will be considered. Regarding the long-term period since 1971, "the drivers of a larger Earth energy imbalance in the 2000s than [before] are still unclear. … Future studies are needed to further explain the drivers of this change" (von Schuckmann et al. 2023: p. 1694). Laterally, the observed heat excess is unevenly distributed over the world ocean (Fig. 1), in contrast to what naively may be expected from rising atmospheric $CO_2$ concentrations. Rather, warming seems to be most pronounced in the austral and boreal west-wind belts. Selected thermodynamic relations between OHC and cloudiness are briefly discussed in **Section 6**."

2. **OHC**

**Review 1:** „ I identify one small part of this manuscript that I think is wrong and should be deleted, namely the part around Eq. (5)."

**RF:** I have described my intention behind eq. (5) in more detail now:

"The process depicted in Fig. 6 measures the total heat flux $\int dh = \int T d\eta$ which changes the entropy of the given sample from the current value, $\eta$, to some arbitrary reference value, $\eta_{\text{ref}}$, and this way, the process also changes the parcel's enthalpy from $h^{\text{SW}}(S, \eta, p_0)$ to $h^{\text{SW}}(S, \eta_{\text{ref}}, p_0)$. Integration over all ocean samples results in an OHC value of

$$OHC^* = \int \left[ h^{\text{SW}}(S, \eta, p_0) - h^{\text{SW}}(S, \eta_{\text{ref}}, p_0) \right] \rho^{\text{SW}}(S, \eta, p) dV. \tag{5}$$

While the choice of the OHC reference state is - in principle - entirely arbitrary, such as simply putting $\eta_{\text{ref}} = 0$, it is reasonable to better adapt this selection to the purpose of the OHC definition. The main purpose of estimating OHC is keeping track of the ocean's long-term energy balance, in particular of the ocean's share of global warming. Three conditions appear immediately plausible in order to achieve this goal,

(i) *The OHC definition should ensure that OHC differences represent a suitable spatial integral over the heat fluxes crossing the ocean's boundaries*. As discussed in more detail

in Section 5.3, production of entropy, $d_i\eta$, caused by irreversible processes between different parcels within the ocean, does not affect the ocean's total enthalpy budget. This is quite in contrast to entropy exchange, $d_e\eta$, of the given sample in the form of reversible heat flux across its boundary. Such irreversible processes affect the ocean's total potential enthalpy much less than its total entropy (McDougall et al. 2021). For this reason the OHC reference state should explicitly be defined in terms of potential enthalpy, $h^{SW}(S, \eta_{ref}, p_0)$, and this way only implicitly in terms of entropy by specifying $\eta_{ref}(S)$.

(ii)    *Provided that the ocean's mass remains the same between any two ocean states (1) and (2), the difference OHC(1) – OHC(2) should depend only on the surface heat flux balance during the time in between*. For this reason, the OHC reference value should be independent of changes occurring in the density distribution, $\rho^{SW}(S, \eta, p)$. This can be achieved by assigning to each ocean parcel the same reference potential enthalpy, $h^{SW}(S, \eta_{ref}, p_0) = \text{const}$, even though such a state may hardly ever be observed in the real ocean.

(iii)   *Quantitatively, OHC values estimated at different times or places should be mutually comparable without estimation bias resulting from possibly changing methods of OHC calculation*. For this reason, resulting OHC values should be independent of the inevitable arbitrary, physically irrelevant reference-state conditions imposed on energy and entropy, such as eqs. (1)-(3). This can be achieved by assigning to each ocean parcel the same standard-ocean enthalpy as its reference potential enthalpy, $h^{SW}(S, \eta_{ref}, p_0) = h_{SO}$. In the special case of TEOS-10 enthalpy, this value is defined by eq. (2), $h_{SO} = 0$. This choice is implicitly made by the definition (4) but needed to be considered explicitly as soon as alternative equations for seawater enthalpy or entropy are employed, such as those of Millero and Leung (1976) and Millero (1982, 1983)."

**3. Bible quotation**

**Review 1:** "I discourage the inclusion of religion in scientific papers. First, the bible is not a scientific book, nor is it scientifically correct since its discussion of the arrival of humans on earth (in its first chapter) contradicts the known science of evolution. Second, scientific papers should be able to be read by authors of all religions without them encountering quotes which somehow endorses the basic textbook of one religion. Hence, I think that biblical quotes, just like quotes from the textbooks of any religion, should not be allowed in Ocean Science. Please delete these lines."

**RF:** I agree that religious arguments should be excluded from scientific papers. However, more than 2000 years ago, there was no science apart from religion; modern science and modern religion have common roots. They diverged when religious branches turned into frozen dogmatic prescriptions to be used as instruments of political power, while science remained open for change, correction and evolution. Regardless of that, both religion and science still are – even if very distinct – mental models for the structure and the causal functioning of the perceived world.

With respect to the hydrological cycle, the Bible quotations in the paper are the oldest documented observations of nature that I could find. Similar other "holy books", such as the *Popol Vuh* of the Maya, or the *Teaching of Buddha*, are almost exclusively focussed on human life and history, rather than observations of natural phenomena (although the scientific term *hurricane* is borrowed from the Maya god *Huracan*). People of the past had noticed that all rivers discharge into the sea whose

level did not rise though, and that clouds may release vast amounts of water while floating virtually weightlessly across the sky. They had no plausible causal explanations for such mysteries and credited those to divine intervention.

The history of understanding the hydrological cycle has ancient roots and is, remarkably, not finished yet. This paper addresses relevant pending problems of modelling that cycle, and in so far it seems scientifically appropriate to refer to the exceptionally few written cases of revealing the poor very beginning of this understanding.

**4. SMOW**

**Review 1:** "Surely the water which IAPWS-95 describes is not Standard Mean Ocean Seawater"

**RF:** IAPWS-95 describes IAEA Standard Mean Ocean Water (SMOW) which is the solvent of IAPSO Standard Seawater.

**5. Absolute Salinity**

**Review 1:** "Line 201 and in hundreds of places throughout the paper, Absolute Salinity and Conservative Temperature are used without their upper-case letters. This goes against what IOC et al. (2010) and Valladares, J., Fennel, W., and Morozov, E.G (2011) and Spall et al (2013) [see below] dictate. I think the field should stick with the upper-case letters, simply because there are many different possible definitions of absolute salinity and of conservative temperature, but there is only one definition (each) of Absolute Salinity and Conservative Temperature."

**RF:** In section 2, the paragraph introducing salinity has been edited to read:

"… The variable $S$, at which a subscript A is omitted here for simplicity, is the specific or *Absolute Salinity*, the mass fraction of dissolved salt in seawater, which differs from *Practical Salinity*, $S_P$, measured by present-day oceanographic instruments, as well as from various other obsolete salinity scales (Millero et al. 2008). Throughout this paper, the term "salinity" is short hand exclusively for TEOS-10 Absolute Salinity. …"

Lower-case conservative temperature appears only once in the text; fixed.

**6. Internal energy *E* or *U**

**Review 1:** "Replace E with U in this and subsequent equations. IOC et al (2010) has used U and u for internal energy (extensive and intensive), and this review paper should do the same."

**RF:** The use of E rather U has now been justified below eq. (B.1):

"Note that IOC et al. (2010) uses the symbol $U$ for the *internal energy* rather than $E$ in eq. (B.1). This replacement is done here for denoting with $u$ the wind speed, eq. (6), rather than specific internal energy, which is defined here by $e = E/m$, eqs. (1) and (B.3). The symbols $E$ and $e$ are frequently used for internal energy in the thermodynamic literature, for example by Gibbs (1873a) or Landau and Lifschitz (1966)."

Appendix B is just about thermodynamics in general without special emphasis on oceanography.

**7. Minus sign of eq. (B.10)**

**Review 1:** "equation (B.10) has a sign error. The last term should be added, not subtracted."

**RF:** Consider the seawater case $n = 2$, $w_0 = 1 - S$, $w_1 = S$. We get from (B.10)

$$\mu_S = \mu_1 = g + \left(\frac{\partial g}{\partial w_1}\right)_{T,p} - \sum_{j=1}^{1} w_j \left(\frac{\partial g}{\partial w_j}\right)_{T,p} = g + (1-S)\left(\frac{\partial g}{\partial S}\right)_{T,p}$$

in agreement with eq. (2.9.5) of the Manual. No sign error.

Despite this, for clarity, the sentence above (B.10) has been changed to

"Because the Gibbs function depends only on the independent intensive variables, $g(w_1, \ldots, w_{n-1}, T, p)$, the solutes' chemical potentials, $i > 0$, are"

**8. Partial derivative of (B.11)**

**Review 1:** "The things that are held constant during the differentiation in the last term in this equation are not correct. They should be the same as the corresponding term in the next equation. Eq. (B.11)."

**RF:** Yes, but (B.11) is to be adjusted to (B.10). Corrected to

$$\mu_0 = \left(\frac{\partial G}{\partial m_0}\right)_{T,p,m_{j>0}} = \left(\frac{\partial (m\,g)}{\partial m_0}\right)_{T,p,m_{j>0}} = g - \sum_{j=1}^{n-1} w_j \left(\frac{\partial g}{\partial w_j}\right)_{T,p,w_{k\neq j}}. \tag{B.11}$$

**Reviewer 2:**

**RF:** Thanks for careful reading and detailed discussing.

    **1. OHC**

**Review 2:** „Section 3. Unlike the other sections, this section uses persuasive writing rather than scientific writing to convince the reader of the legitimate and rigorous character of the TEOS10 approach to defining ocean heat content. In essence, this amounts to providing a solution to a question that has not been properly formulated first; as result, the reader is not given the scientific elements necessary to assess the legitimacy of the author's assertions. Moreover, the topic is not properly reviewed or discussed in the context of past research on the issue. As a result, this section does not conform to accepted scientific standards, and therefore should either be significantly improved, or removed from the paper.

… the problem of how to define heat was originally defined as the problem of how to separate the total energy transport into a dynamical and thermodynamic part …

TEOS-10 or the author's section gives the impression that there is only a unique way to address the problem and that there is nothing left to be solved, when this is clearly not the case."

**RF:** Entropy (here, "$N$") was originally discovered and defined by Clausius describing heat exchange in the form of dN = dQ/T. Using Clausius entropy, a problem of defining heat does not exist: entropy is formally defined for the first time in terms of heat whatever "heat" may actually be. Meanwhile, there are various alternative definitions of entropies in the subsequent literature, more or less related to heat, but here empirical Clausius entropy is used in the same sense as by numerous textbooks from Gibbs to Prigogine.

Section 3 explains OHC in terms of surface entropy flux. This is no balance of the total ocean energy, nor is it a description of the real heat exchange between ocean and atmosphere. Section 3 proposes a fictitious thermodynamic process by which proper heat exchange is formally related to thermodynamic state properties.

Definitions are neither right nor wrong per se; they may be more or less useful for a certain purpose. OHC is a matter of definition. The main intention of Section 3 is placing emphasis on the physical arbitrariness of any OHC definition. It is up to the oceanographic community to adopt one option as a standard to ensure comparability of reported figures. Note that Section 3 had already substantially been modified in response to Reviewer 1.

As a measurable physical quantity, "heat" is defined only as a heat exchange between two bodies rather than any "heat substance" contained in a volume. It is clearly said that the common term "OHC" is thermodynamically sloppy and ambiguous; Section 3 suggests one option of defining a heat flux and a reference state consistent with the TEOS-10 OHC definition given by McDougall et al. (2021). This simple option does not require any details of complex energy transformation processes within the real ocean. However, a sentence hinting on more complex analyses of the OHC problem has been added to Section 3:

==“OHC as a part of the total energy balance of the ocean is analysed by Tailleux (2010, 2018) and Tailleux and Dubos (2024)”.==

    **2. Line 65.**

**Review 2:** "Typically, present numerical climate models suffer from an "ocean heat budget closure problem" (Josey et al. 1999) and describe the m–2 m–2 ocean-atmosphere heat flux only to within uncertainties between 10 W and 30 W (Josey et al. 2013).

I find this statement confusing because my understanding of the Josey et al papers relate to the `observational' closure problem arising from the technical difficulties of measuring the different heat fluxes component reliably enough and with the desired accuracy. The closure problem in numerical ocean models is a completely different thing. Numerical ocean models will in general exhibit drift depending on many different factors, such as model resolution, and various model errors. The author needs to review the literature more carefully to avoid confusing observational and modelling issues."

**RF:** I do not see the need for such a distinction in the context of my paper. Numerical models can hardly describe ocean surface heat fluxes more precisely than observations by which they were tuned. If those models have numerical problems even larger than 10 W and 30 W per m-2, they lack significance in explaining the discussed effects of 1 W per m-2.

**3. Lines 70-72.**

**Review 2:** "… countless climate projections have been published that reproduce ocean warming like that observed. Presumably, air-sea interactions in such simulations have been analysed. It would therefore be useful if the author could summarise the state of knowledge on the matter, including discussions of the nature of uncertainties, rather than just speculate on the matter."

**RF:** I myself do neither run nor assess climate models or similar products; I just report published statements of renowned experts. All analyses I am aware of do conclude that the uncertainty in estimating the global mean air-sea heat flux is hardly any better than 10 W m-2, if at all. Contributions to this uncertainty are various and complicated, and are certainly not the topic of this paper. What does matter here, however, is the plausible conclusion that a model with uncertainty larger than 10 W m-2, see e.g. Fig. 5.10 in Josey et al. (2013), cannot reliably distinguish and explain effects of the magnitude of 1 W m-2. Attempts to blindly do this may well be considered as speculative.

"Despite a certain success it must still be stated that the modellers among my colleagues are not yet aware of the severity of the problem. As a rule, they work with energy-balance equations whose coefficients had been verified by measurements, while their latent-heat fluxes are mostly determined as the remaining left-over term, which therefore received the residual as an additive. This has implications for the surface temperature that is important for most of the models, and in particular for the water-vapour flux" (Daniela Kracher et al. 2009, doi:10.1127/0941-2948/2009/0412). "The global water cycle and the exchange of freshwater between the atmosphere and ocean is poorly understood" (Penny Holliday et al., 2011). "For most products, it is not possible to close the [global ocean] heat budget to within 10 W m-2 and in some cases the bias is of the order of 30 W m-2" (Simon Josey et al. 2013: p. 128). "The drivers of a larger Earth energy imbalance in the 2000s than [before] are still unclear. … Future studies are needed to further explain the drivers" (Karina von Schuckmann 2023). "Climate models struggle to explain why planetary temperatures spiked suddenly. … No year has confounded climate scientists' predictive capabilities more than 2023" (Gavin Schmidt, Nature, 21 March 2024).

**4. Lines 78-80.**

**Review 2:** "it would be useful to the reader if the author could translate these numbers in terms of implied change in net evaporation or precipitation, assumes that the two balance on average."

**RF:** A global mean evaporation / precipitation of about 1000 mm corresponds to an oceanic latent heat flux of roughly 100 W m-2. RH uncertainty of 1 – 5 %rh corresponds to 5 – 25 W m-2 latent heat flux uncertainty, or 50 - 250 mm of annual precipitation. Related text inserted:

"Unfortunately, marine RH is observed only with uncertainties between 1 and 5 %rh (Lovell-Smith et al. 2016), or, accordingly, between 5 and 25 W m–2 of latent heat flux, which is roughly corresponding to unknown variations ranging up to 50 … 250 mm evaporation."

**Review 2:** "May be the author could also discuss the fact that global warming is expected to heat up land area faster than ocean area. As a result, this may decrease relative humidity, with a possible compensating effect over the ocean like the one suggested by the author."

**RF:** Such a compensation effect is speculative. Dominating 85% of global evaporation occur at the ocean. Global warming on land is rather different from that at sea, and does not belong to "Ocean-Atmosphere Interaction" of this paper. For details see e.g.: Blunden, J., Boyer, T., and Bartow-Gillies, E. (eds.): State of the Climate in 2022, Bull. Amer. Meteor. Soc. 104, S60–S61, https://doi.org/10.1175/2023BAMSStateoftheClimate.1, 2023

5. **Line 95.**

**Review 2:** "…It seems to me that while TEOS10 is clearly a success in providing such improved formulations, it is unclear how it can claim to contribute to the understanding of the functioning of the ocean heat engine… "

**RF:** TEOS-10 does not describe the "heat engine" dynamics of the climate system; TEOS-10 only provides the most accurate, comprehensive and mutually consistent thermodynamic tools for use in climate research. This paper explains that and just offers some simple tutorial examples for the use of TEOS-10 in the climate context.

6. **Figures 3 and 4.**

**Review 2:** "Shouldn't credit or copyright for the photo be indicated? Can these be re-used by others?"

**RF:** Ocean Science is publishing under the Creative Commons Attribution 4.0 License.

7. **Lines 129-134.**

**Review 2:** "The question is whether the TEOS-10 definition of heat is as rigorous as the author claims, as the definition seems an ad-hoc one to me. TEOS-10 proposes a solution to a question that they never define in the first place. See my comments in the major points section."

**RF:** Neither TEOS-10 nor this paper have ever attempted to define "heat". Clausius' original definition of entropy is in terms of heat exchange. "We have … two of the fundamental ideas of the science of heat – the idea of temperature … and the idea of heat as a measurable quantity, which may be transferred from hotter bodies to colder one" (James Clerk Maxwell 1888, Theory of Heat, Longmans & Green: p. 9). "The variation in entropy during an infinitesimal reversible transformation is obtained by dividing the amount of heat absorbed by the system by the temperature of the system" (Enrico Fermi 1937, Thermodynamics. Prentice-Hall: p. 52).

8. **Line 200-203**.

**Review 2:** "Can the author provide some explanation about why a Helmholtz potential is preferred in that case rather than a Gibbs function? The use of a Gibbs function as the basis for TEOS10 is generally understood from the fact that S, T, and p are variables that are the most easily

measured/fixed in practice. We are also told that density is a variable that is very hard to measure in practice, which makes the usefulness of a Helmholtz function hard to understand. So, what are the physical arguments in favour of it?"

**RF:** In statistical physics, a canonical ensemble with the partition function $Z$ determines the Helmholtz energy by $F = -kT \ln Z$. The function $Z$ depends on the particle number $N$, the temperature $T$ and the volume $V$, and is a functional of the microscopic particle interaction energy, entirely independently of the macroscopic phase the substance may actually take. For a given substance or mixture, this formula $F(N, T, V)$ is single-valued and universally valid, be it a gas, a liquid, or any solid phase, such as for water around its critical point and for each of the various ice phases. By contrast to $F$, the Gibbs energy $G(N, T, p)$ is multi-valued in the $T$-$p$ vicinity of phase transitions, where each phase is represented by its own separate "leaf" of $G$ that intersects the leaf of the other phase. TEOS-10 includes a single Helmholtz function jointly for liquid water and water vapour, but two different Gibbs functions. References to related textbooks introducing that matter have been added to this paper:

"For theoretical reasons (namely, the statistical so-called *canonical ensemble*, Landau and Lifschitz 1966: §31; Kittel 1969: Ch. 18), ..."

9. **Line 259-260.**

**Review 2:** Conservative Temperature "Preferred by whom?"

**RF:** Preferred by ocean modellers who started using TEOS-10, see e.g. Almeida et al. (2018), as far as I know this. See also Young (2010), https://doi.org/10.1175/2009JPO4294.1; https://en.wikipedia.org/wiki/Conservative_temperature; or Pawlowicz, R. (2013) Key Physical Variables in the Ocean: Temperature, Salinity, and Density. Nature Education Knowledge 4(4):13

10. **Line 358, Equation 6:**

**Review 2:** "Can you be more specific as to the form of the transfer coefficient Df(u) by providing examples from the literature? I am confused by the author's statement that such a coefficient only depends on u, because my understanding is that such a coefficient also depends on many other things, such as a sea surface roughness, nature of the boundary layer, and so on..."

**RF:** In fact there is a wealth of different definitions of the transfer coefficient Dq(u) of latent heat in oceanography, meteorology and hydrology. In this paper reference is only made to the recent definitions given by Josey (2013: eq. 5.1), Stewart (2008: eq. 5.10c) or Pinker et al. (2014, doi:10.1002/2013JC009386: eq. 1) who parameterise this coefficient simply as a linear function of wind speed, independent of the various other surface properties. To my knowledge, there is no suggestion available yet from the literature for the functional form of Df(u).

11. **Lines 396-398.**

**Review 2:** "This sounds like an important result warranting further attention. However, can the author guarantee that Dq(u) does not depend indirectly on q in a way that would compensate the effect discussed? Change in q may modify the nature of the turbulent boundary layer and the transfer coefficient."

**RF:** Available from TEOS-10 for the first time, relative fugacity is the proper irreversible thermodynamic driving force for the water transport across the air-sea interface. Properties of the turbulent boundary layer may certainly depend on additional properties beyond those provided by TEOS-10.

**12. Lines 637.**

**Review 2:** "The author only discusses irreversibility associated with non-zero relative humidity under the assumption that the oceans and atmosphere have the same temperature. In reality, the latter may also have different temperatures. Can the author comment as to the implications that this would have for his theory?"

**RF:** The thermal "skin effect" of the air-sea interface has been studied by several authors such as Peter Saunders (1967) or Kristina Katsaros (1980). The sensible heat flux affected by this effect is generally small compared to the latent heat flux. In the sense of Onsager linear irreversible thermodynamics, the heat flux driven by the temperature difference, and the evaporation flux driven by the different chemical potentials (that is, relative fugacity) will also have a cross effect of evaporation driven by the temperature gradient and, symmetrically, of sensible heat flux driven by the relative fugacity. However, the magnitude of the cross effect is unclear and has so far been assumed to be negligible.

**13. Lines 723-725.**

**Review 2:** "My understanding is that the Zlcl is to be obtained by integrating the hydrostatic relationship, which can only lead to the author's formula (52) if the entropy and specific humidity are perfectly uniform from the surface to the bottom of the cloud. Is that really the case in reality?"

**RF:** In the LCL model presented here, it is assumed that the uplift of air occurs at constant entropy and specific humidity in order to compute the LCL pressure. The same assumption is also applied for computing the LCL height. Of course, this is an idealised model of reality. Figure 20 in Feistel et al. (2010a) shows measured radiosonde profiles of those quantities with an approximately isentropic surface layer over the tropical Atlantic.

**14. Lines 930-934.**

**Review 2:** "I am surprised to see the quantities $-pdV$ and $Td\eta$ equated with the work and heat transfers $\delta W$ and $\delta Q$, because this is only true for reversible and quasi-static transfers. As far as I am aware, the exact relations are $T\,d\eta \geq \delta Q$ and $-p\,dV \leq \delta W$. This can be verified for an adiabatic expansion of a piston in a vacuum. In that case, $\delta Q$=0 yet the entropy increase; moreover, $\delta W$=0, yet V increases so that $-p\,dV$<0. Moreover, note that p, V, T and eta relates to internal properties of the fluid, while the concepts of heat and work transfers relate to external properties describing the interactions of the fluid with its environment, so that it is dangerous and confusing to equate internal and external properties without further discussion."

**RF:** It is true that TEOS-10 describes equilibrium thermodynamics of seawater, ice and humid air. In classical thermodynamics all exchange processes are idealised as reversible and quasi-static. For application of TEOS-10 to geophysical processes, TEOS-10 may be generalised under the assumption of local equilibrium, see e.g. Feistel and Hellmuth (2024a) and the discussion of entropy production in Section 5.3 of this paper.

The discussion on OHC in this paper is explicitly focussed on the fact that heat is an exchange quantity rather than a state quantity. "We have … a right to speak of heat as a *measurable quantity*, … however, … we have no right to treat heat as a *substance*" (J. C. Maxwell, 1888, Theory of Heat, p. 7).

As an aside, the gas expansion into vacuum violates the condition of local equilibrium (namely, the existence of a local Maxwell distribution of particle velocities) so that neither temperature nor entropy may properly be defined in that case.

**15. Lines 962-963.**

**Review 2:** "I thought that this condition was also true in the presence of gravity. Can the author explain how gravity affects these conditions, given that this is obviously relevant to the oceanic case?"

**RF:** Under gravity, the equilibrium condition of equal (molar) chemical potentials $\mu$ is replaced by the condition that for each species the form $(\mu + M \phi)$ must take equal values across a volume, where M is the molar mass and $\phi$ is the gravity potential (Guggenheim 1949: chapter XI).

---

## Referee Report (RR1)

**Review of revised version of:**

**TEOS-10 and the climatic relevance of ocean-atmosphere interaction**
**By Rainer Feistel**

**Summary and recommendation.** I believe that the revised version of this manuscript is much improved and is now closer to be acceptable for publication. However, I believe that the justification for the manuscript, as well as elements of Section 3, still need to be substantially improved before the manuscript can be accepted for publication.

**Specific comments**

**Lines 6-7 Abstract: 'Unpredicted observations [...] are challenging the numerical models'** Not sure what that means or what this is based upon. Numerical climate models are generally able to predict both global atmospheric and oceanic warming, although this might be admittedly for the wrong reasons.

**Lines 71-74. 'Typically present numerical climate models suffer from an "ocean heat budget problem"'** As previously mentioned, I don't understand what that means. Numerical climate models offer an energetically consistent description of the climate system, in the sense that all the energy that comes in either comes out or is absorbed within the climate system. The term 'heat budget problem' suggests that models do not close their heat budget, but it seems to me that they do, so I have no clue what the author is actually talking about. sentence is especially confusing as the references cited relate to observations, not numerical models. The author should make an effort at more clearly articulating the nature of the problem, its origin, and how his proposed approach can help. The author should also carefully distinguish between numerical versus observational issues, errors versus uncertainties, because these distinctions are crucial to identify how to make progress.

More generally, I am quite puzzled by the way the author justifies his work, because climate projections using comprehensive climate models have been able to predict global atmospheric and oceanic warming, as documented by all IPCC reports. Moreover, CMIP have made the outputs of numerous simulations openly available to the community, meaning that it is possible to diagnose from such simulations which processes are responsible for ocean warming in such simulations. A priori, such warming can come from: 1) increased incoming shortwave radiation due to cloud cover redistribution; 2) reduced latent heat release in regions of deep water formation, resulting in less deep water formation; 3) reduced latent heat release; 4) increased incoming long wave radiation due to changes in cloud and cloud cover; 5) increased sensible heat flux. It would be clearer if the author would review the possible different mechanisms by which the ocean can warm, state that this is realistic remains

uncertain, and focus on the aspects that potentially can be clarified using rigorous thermodynamic arguments as a way to identify possible shortcomings in coupled climate models.

**Section 3** I continue to believe that the approach developed by the author to justify the TEOS-10 definition of heat is heuristic rather than deductive and therefore scientifically unsatisfactory. Of course, heuristics are very common in science, so that this can be perfectly acceptable if this is explicitly acknowledged. My main concern is that this is not really the case here, and that the author present potential enthalpy as more rigorous than it is, without acknowledging the limitations of the approach. I think that the author should try to write this section more objectively and in a more balanced way before the paper can be accepted for publication. As far as I know, potential enthalpy does not solve the problem and should not be presented as if it did. The author should try to identify and discuss what remains to be done to achieve a more satisfactory solution. The following lists a number of places where the discussion can be improved.

**Lines 294-296: 'The obsolete hypothesis of heat being a substance is excluded'** I think this phrasing is potentially confusing to readers, because 'obsolete' is often understood (especially in its US meaning) as being superseded by something new, but this is not the case, isn't it? As far as I am aware., classical thermodynamics does not define 'heat' as an internal property of the system, but rather tries to limit the use of `heat' as a mode of heat transfer (Romer, 2001, 'Heat is not a noun', Amer. J. Phys., https://doi.org/10.1119/1.1341254), which I think is what the author is trying to convey in this section. For this reason, I think it would be clearer to say that the view of heat as a substance promoted by the calorimetric theory of heat has been debunked, rather than it is obsolete (even nicer would be to explain the scientific basis for its refutation, as I must admit I never fully understood the arguments). Here, I think it would be helpful to reader if the author could point out that the only accepted sub-forms of energy that appears to be well accepted in classical thermodynamics is the partition of total energy into 'useful' and 'useless' forms of energy, which thermodynamicists refer to as 'exergy' versus 'anergy', or 'free energy' versus 'dead energy', with a review of existing terminology comprehensively reviewed by (Marquet, 1991, on the concept of exergy and available enthalpy: application to atmospheric energetics https://doi.org/10.1002/qj.49711749903 I bring this up, because it seems to me that the concept of 'OHC' used by oceanographers is the counterpart of the concepts of 'anergy' or 'dead energy' of classical thermodynamics. I think that this is relevant because oceanographers often regard 'heat' as the dynamically inactive part of the total energy that is passively transported poleward to remove the excess of energy imparted to the equatorial regions. For instance, Young (2010) and Nycander (2010) both pointed out that defining `heat' in terms of potential enthalpy implies that the useful part of potential energy should be defined in terms of dynamic enthalpy/effective potential energy. I therefore think that discussing this point would potentially greatly enhance the scientific value of Section 3.

**Lines 315-316 `However, this OHC definition has no rigorous thermodynamic justification'** I am not sure that I agree with this statement, because if you re-read Bryan (1962) and the ensuing literature, it is apparent that OHC was introduced as a way to separate the total energy transport into a dynamical and thermodynamic component, the latter being assumed to be represented by the non-mechanical energy part of internal energy. The idea was plausible at the time, because kinetic energy and gravitational potential energy have been traditionally assumed as mechanical forms of energy. On this basis, it seemed logical to assume the thermodynamic component of total energy to be related to the non-mechanical part of internal energy, which $c_{p0}\theta$ is meant to approximate. The phrasing suggests that the thermodynamic justification of potential enthalpy is more rigorous, but this is not really the impression one gets from McDougall (2003). Indeed, McDougall redefines the problem of defining heat as the problem of heuristically manipulating one of the expressions for the first law of thermodynamics into an equation for a thermodynamic variable that is as conservative as feasible and whose surface flux matches the neat surface heat flux. Clearly, this way of approaching the problem admits several solutions, so cannot define the concept uniquely. For this reason, I think that it would be more accurate to say that potential enthalpy has a clearer and more transparent thermodynamic justification rather than rigorous, because scientifically, McDougall (2003) does not qualify as `rigorous' since it is essentially heuristic in its approach. For instance, it does not justify why heat should be defined as a quasi-material function of specific entropy and salinity. Note here that atmosphericists study heat transport in terms of the transport of (moist) static energy, which is also accurately conservative, and whose boundary fluxes coincide exactly with the boundary heat transfer. It also does not discuss the limitations of the approach, or what potential enthalpy is supposed to approximate. For this reason, it would be useful if the author could clarify these or at least comment on these points.

**Lines 332-333 – The work required to lift and lower the parcel is balanced.** I don't understand this because once the temperature (and salinity) of the parcel lifted to the surface has been modified, the work needed to lower it back to its original position will in general be different than that necessary to lift it up. I therefore don't understand what the author means by it is 'balanced'. Please explain.

**Lines 334-336 – The reference state relative to which OHC is measured is arbitrary [...]** I don't think that this is generally true. I believe that this is true only for systems whose mass does not change with time but not for systems for which mass changes with time, see Lang et al. (2018) 'poleward energy transport: is the standard definition physically relevant at all time scales? https://doi.org/10.1007/s00382-017-3722-x The latter study suggests that if the mass changes, the relevant reference state should be related to the global mean value of the system.

---

## Author Response (AR3)

**Point by point response to reviewer comments - round #2**

**Reviewer 1: Submitted on 12 Aug 2024**

**R1:** The revised manuscript is ready for publication, in my opinion. There is just one sentence that I think should be deleted. It is at lines 332-333 of the revised manuscript where it says "The work required to lift and lower the parcel is balanced." This is not correct. The raising happens at one specific volume and the lowering at another, so the two works are different. I suggest deleting the sentence.

**RF:** It is important to understand here that the fictitious excursion of the parcel to the surface and back must not alter the energy balance of the ocean. For this reason the parcel's heat exported reversibly at the surface is imported reversibly again so that the parcel is returned to its original state before it is lowered back to its depth. After the parcels excursion for the purpose of heat exchange and heat measurement, the ocean's state is considered to be exactly the same as before. The ocean should not gain or lose any energy as a result of the excursion. I have tried to clarify this:

"The work required to lift and lower the parcel is balanced because the parcel's thermodynamic state is exactly the same before and after the balanced reversible heat exchange across the surface. The "heat content" defined this way for a single parcel is added up then over all ocean parcels to result in its total OHC value."

**R1:** One of the reviewers was concerned with the arbitrary nature of the reference states of TEOS-10. This has been addressed in many previous publications, and the additional text between lines 243 and 254 reviews this material for the benefit of that reviewer and for the general reader.

**RF:** Thanks for this support. I can only hope that these arguments make those things clear enough.

**Reviewer 2: Submitted on 01 Sep 2024**

Summary and recommendation. I believe that the revised version of this manuscript is much improved and is now closer to be acceptable for publication. However, I believe that the justification for the manuscript, as well as elements of Section 3, still need to be substantially improved before the manuscript can be accepted for publication.

**Specific comments**

**R2: Lines 6-7 Abstract**: 'Unpredicted observations [...] are challenging the numerical models' Not sure what that means or what this is based upon. Numerical climate models are generally able to predict both global atmospheric and oceanic warming, although this might be admittedly for the wrong reasons.

**RF:** I cannot add quotations to the Abstract, but I can add a couple here to explain what I mean and upon which statements it is based. I trust in the correctness of such expert statements (the first quotation has been added to the text):

"Climate models struggle to explain why planetary temperatures spiked suddenly. … No year has confounded climate scientists' predictive capabilities more than 2023. … This sudden heat spike greatly exceeds predictions made by statistical climate models." [Schmidt, G. (2024): Climate models can't explain 2023's huge heat anomaly - we could be in uncharted territory. Nature 627, 467. https://doi.org/10.1038/d41586-024-00816-z]

"The drivers of a larger Earth energy imbalance in the 2000s than [before] are still unclear. ... Future studies are needed to further explain the drivers."

[von Schuckmann, K. et al. (2023): Heat stored in the Earth system 1960–2020: where does the energy go? Earth System Science Data 15, 1675–1709. https://doi.org/10.5194/essd-15-1675-2023]

**R2: Lines 71-74.** 'Typically present numerical climate models suffer from an "ocean heat budget problem" As previously mentioned, I don't understand what that means.

**RF:** The phrase "ocean heat budget problem" is a literal quotation from Josey et al. (1999, 2013), as cited in my text. There, I find statements like "the ocean heat budget closure problem that remains a major unresolved issue in the field" (2013: p. 115). Please find detailed explanations there. I do not run such models and do only rely on such published assessments. It is not the task of my paper to go into those details reported elsewhere.

**R2:** Numerical climate models offer an energetically consistent description of the climate system, in the sense that all the energy that comes in either comes out or is absorbed within the climate system. The term 'heat budget problem' suggests that models do not close their heat budget, but it seems to me that they do, so I have no clue what the author is actually talking about. sentence is especially confusing as the references cited relate to observations, not numerical models. The author should make an effort at more clearly articulating the nature of the problem, its origin, and how his proposed approach can help. The author should also carefully distinguish between numerical versus observational issues, errors versus uncertainties, because these distinctions are crucial to identify how to make progress.

**RF:** If the ocean surface heat budget is uncertain to  $10 - 30 \text{ W/m}^2$ , and unclear flux anomalies amount to about  $1 \text{ W/m}^2$ , this does not seem to me a proper "closure of the heat budget problem".

To my knowledge, ALL numerical circulation models fail to strictly obey energy conservation:

"Unfortunately, it is not always possible to maintain the exact conservation laws and symmetries in the [discretized model] equations."

[Griffies, S.M., Adcroft, A.J. (2008): Formulating the Equations of Ocean Models. In: Hecht, M.W., Hasumi, H. (eds.): Ocean Modeling in an Eddying Regime, Geophysical Monograph Series 177, American Geophysical Union, pp. 281-317. doi:10.1029/177GM18]

Certainly, TEOS-10 can and will not resolve any such problems, but it may improve model details such as parameterizations of the evaporation flux by using chemical potentials of water rather than specific humidities, as current models still do following Dalton. This is the aim of my paper.

**R2:** More generally, I am quite puzzled by the way the author justifies his work, because climate projections using comprehensive climate models have been able to predict global atmospheric and oceanic warming, as documented by all IPCC reports.

RF: Again, I do not run any climate models, I only refer to publications of climate modellers.

"Climate models struggle to explain why planetary temperatures spiked suddenly. … No year has confounded climate scientists' predictive capabilities more than 2023." [Schmidt, G. (2024): Climate models can't explain 2023's huge heat anomaly - we could be in uncharted territory. Nature 627, 467. https://doi.org/10.1038/d41586-024-00816-z]

**R2:** Moreover, CMIP have made the outputs of numerous simulations openly available to the community, meaning that it is possible to diagnose from such simulations which processes are responsible for ocean warming in such simulations. A priori, such warming can come from: 1) increased incoming shortwave radiation due to cloud cover redistribution; 2) reduced latent heat release in regions of deep water formation, resulting in less deep water formation; 3) reduced latent heat release; 4) increased incoming long wave radiation due to changes in cloud and cloud cover; 5)

increased sensible heat flux. It would be clearer if the author would review the possible different mechanisms by which the ocean can warm, state that this is realistic remains uncertain, and focus on the aspects that potentially can be clarified using rigorous thermodynamic arguments as a way to identify possible shortcomings in coupled climate models.

**RF:** Again, analysing deficiencies of climate models and suggesting specific remedies is far beyond the scope of this paper. I only refer to such analyses published by experts:

"Most CMIP6 models fail to provide as much heat into the ocean as observed." [Weller, R.A.; Lukas, R.; Potemra, J.; Plueddemann, A.J.; Fairall, C.; Bigorre, S. Ocean Reference Stations: Long-Term, Open-Ocean Observations of Surface Meteorology and Air–Sea Fluxes Are Essential Benchmarks. Cover. Bull. Am. Meteorol. Soc. 2022, 103, E1968–E1990. https://doi.org/10.1175/BAMS-D-21-0084.1. p. E1968]

"The drivers of a larger EEI in the 2000s than in the long-term period since 1971 are still unclear, and several mechanisms are discussed in literature. For example, Loeb et al. (2021) argue for a decreased reflection of energy back into space by clouds (including aerosol cloud interactions) and sea ice and increases in well-mixed greenhouse gases (GHG) and water vapor to account for this increase in EEI. Kramer et al. (2021) refer to a combination of rising concentrations of well-mixed GHG and recent reductions in aerosol emissions to be accounting for the increase, and Liu et al. (2020) address changes in surface heat flux together with planetary heat redistribution and changes in ocean heat storage."

[Von Schuckmann, K. et al. (2023): Heat stored in the Earth system 1960–2020: Where does the energy go? Earth Syst. Sci. Data 15, 1675–1709, https://doi.org/10.5194/essd-15-1675-2023]

I am just recommending TEOS-10 to be used in climate models and provide some tutorial examples for doing this, but the ultimate net improvement from using TEOS-10 can only be concluded from future practical implementations in global models which are missing yet, unfortunately, even a decade after the TEOS-10 adoption by IUGG.

**R2: Section 3** I continue to believe that the approach developed by the author to justify the TEOS-10 definition of heat is heuristic rather than deductive and therefore scientifically unsatisfactory. Of course, heuristics are very common in science, so that this can be perfectly acceptable if this is explicitly acknowledged. My main concern is that this is not really the case here, and that the author present potential enthalpy as more rigorous than it is, without acknowledging the limitations of the approach. I think that the author should try to write this section more objectively and in a more balanced way before the paper can be accepted for publication. As far as I know, potential enthalpy does not solve the problem and should not be presented as if it did. The author should try to identify and discuss what remains to be done to achieve a more satisfactory solution.

**RF:** There is no "TEOS-10 definition of heat" in this paper. Rather, the reader is reminded of the conventional definition of heat by Clausius, Maxwell and others, and the century-old textbook knowledge that there does not exist any state quantity exactly representing "heat". After a cyclic process, when a system has returned to its previous state, all state quantities take the same values as before, but the heat exchange does not need to be balanced during that cycle.

Heat is a measurable exchange quantity rather than a state quantity (Maxwell 1888). However, there are state quantities whose difference matches the particular heat exchange associated with a specific process. Different heat exchange processes may require different such state quantities which in that case represent the related quantity of transferred heat.

McDougall et al. (2021) proposed the state quantity "potential enthalpy" as a measure of the ocean heat content. This constitutes an exact measure of "heat" only if it is associated with a suitably designed heat exchange process. Here, such a conceptual process is proposed. This attempt does not need to present the only suitable process. Defining such a process along with the OHC formula would give the OHC definition an improved thermodynamic justification. This "remains to be done to achieve a more satisfactory solution".

Advantages and deficiencies of the use of potential enthalpy are in detail discussed in the TEOS-10 Manual and several related articles. There is no need to repeat these aspects in the current paper.

R2: The following lists a number of places where the discussion can be improved.

Lines 294-296: 'The obsolete hypothesis of heat being a substance is excluded' I think this phrasing is potentially confusing to readers, because 'obsolete' is often understood (especially in its US meaning) as being superseded by something new, but this is not the case, isn't it? As far as I am aware., classical thermodynamics does not define 'heat' as an internal property of the system, but rather tries to limit the use of `heat' as a mode of heat transfer (Romer, 2001, 'Heat is not a noun', Amer. J. Phys., https://doi.org/10.1119/1.1341254), which I think is what the author is trying to convein this section. For this reason, I think it would be clearer to say that the view of heat as a substance promoted by the calorimetric theory of heat has been debunked, rather than it is obsolete (even nicer would be to explain the scientific basis for its refutation, as I must admit I never fully understood the arguments). Here, I think it would be helpful to reader if the author could point out that the only accepted sub-forms of energy that appears to be well accepted in classical thermodynamics is the partition of total energy into 'useful' and 'useless' forms of energy, which thermodynamicists refer to as 'exergy' versus 'anergy', or 'free energy' versus 'dead energy', with a review of existing terminology comprehensively reviewed by (Marquet, 1991, on the concept of exergy and available enthalpy: application to atmospheric energetics https://doi.org/10.1002/qj.49711749903 I bring this up, because it seems to me that the concept of

https://doi.org/10.1002/qj.49/11/499031 bring this up, because it seems to me that the concept of 'OHC' used by oceanographers is the counterpart of the concepts of 'anergy' or 'dead energy' of classical thermodynamics. I think that this is relevant because oceanographers often regard 'heat' as the dynamically inactive part of the total energy that is passively transported poleward to remove the excess of energy imparted to the equatorial regions. For instance, Young (2010) and Nycander (2010) both pointed out that defining `heat' in terms of potential enthalpy implies that the useful part of potential energy should be defined in terms of dynamic enthalpy/effective potential energy. I therefore think that discussing this point would potentially greatly enhance the scientific value of Section 3.

**RF:** James Clarke Maxwell and Arnold Sommerfeld are renowned experts in thermodynamics, and I have included their literal quotations for their rigour. To understand why heat "is not a substance" (an historical obsolete idea known as caloricum or phlogiston) it is sufficient to read the explanations of Clausius, Maxwell or many other textbook authors. The main argument is that after an excursion, a system can precisely return to its former state while having consumed (or lost) a non-zero amount of heat during that cycle. Any "substance", however, had to return to its original amount, gaining zero during a cycle. All I need and want to do here is to hint the reader on classical thermodynamic textbooks. There is nothing new to say.

Thanks for the Romer reference; I have added the Romer quotation: "Heat is not a substance! More formally: Heat is not a thermodynamic function of state" (Romer 2001: p. 107). Readers in doubt may look it up.

I do not see the need here to review the various forms of energy introduced in geophysics for special purposes. I only want to draw attention to the relation between OHC and proper thermodynamic "heat", appreciating novel definitions that have become available through TEOS-10.

**R2: Lines 315-316** 'However, this OHC definition has no rigorous thermodynamic justification' I am not sure that I agree with this statement, because if you re-read Bryan (1962) and the ensuing literature, it is apparent that OHC was introduced as a way to separate the total energy transport into a dynamical and thermodynamic component, the latter being assumed to be represented by the non-mechanical energy part of internal energy. The idea was plausible at the time, because kinetic energy and gravitational potential energy have been traditionally assumed as mechanical forms of energy. On this basis, it seemed logical to assume the thermodynamic component of total energy to be related to the non-mechanical part of internal energy, which cp0 theta is meant to approximate.

**RF:** To be clearer, this phrase has been changed to:

"However, in representing a kind of "heat substance", this OHC definition has no rigorous thermodynamic justification"

**R2:** The phrasing suggests that the thermodynamic justification of potential enthalpy is more rigorous, but this is not really the impression one gets from McDougall (2003). Indeed, McDougall redefines the problem of defining heat as the problem of heuristically manipulating one of the expressions for the first law of thermodynamics into an equation for a thermodynamic variable that is as conservative as feasible and whose surface flux matches the neat surface heat flux. Clearly, this way of approaching the problem admits several solutions, so cannot define the conceptuniquely. For this reason, I think that it would be more accurate to say that potential enthalpy has a clearer and more transparent thermodynamic justification rather than rigorous, because scientifically, McDougall (2003) does not qualify as `rigorous' since it is essentially heuristic in its approach. For instance, it does not justify why heat should be defined as a quasi-material function of specific entropy and salinity. Note here that atmosphericists study heat transport in terms of the transport of (moist) static energy, which is also accurately conservative, and whose boundary fluxes coincide exactly with the boundary heat transfer. It also does not discuss the limitations of the approach, or what potential enthalpy is supposed to approximate. For this reason, it would be useful if the author could clarify these or at least comment on these points.

**RF:** From my perspective, McDougall has thoroughly explained his motivation to define potential enthalpy, and has done a lot of mathematics to derive or estimate the properties of this quantity. Definitions are a matter of usefulness, not a matter of right or wrong. My only task in this context is an attempt to relate the (path-independent) state quantity "potential enthalpy" to the conventional thermodynamic (path-dependent) exchange quantity "heat". The OHC part of this paper goes already into the relation between heat and potential enthalpy.

**R2: Lines 332-333** – The work required to lift and lower the parcel is balanced. I don't understand this because once the temperature (and salinity) of the parcel lifted to the surface has been modified, the work needed to lower it back to its original position will in general be different than that necessary to lift it up. I therefore don't understand what the author means by it is 'balanced'. Please explain.

**RF:** The lifting and lowering of the parcel is assumed to perform an exact thermodynamic cycle, so that the ocean's state before that excursion is the same as afterwards. The parcel's salinity remains the same all the time, and its entropy at arrival at the surface is restored by putting back the exchanged heat before the parcel is lowered again. I have added text to emphasise this.

"The work required to lift and lower the parcel is balanced because the parcel's thermodynamic state is exactly the same before and after the balanced reversible heat exchange across the surface. The "heat content" defined this way for a single parcel is added up then over all ocean parcels to result in its total OHC value."

**R2: Lines 334-336** – The reference state relative to which OHC is measured is arbitrary [...] I don't think that this is generally true. I believe that this is true only for systems whose mass does not change with time but not for systems for which mass changes with time, see Lang et al. (2018) 'poleward energy transport: is the standard definition physically relevant at all time scales? https://doi.org/10.1007/s00382-017-3722-x The latter study suggests that if the mass changes, the relevant reference state should be related to the global mean value of the system.

**RF:** As part of an arbitrary definition, reference states may always be defined arbitrarily. Of course, such a state definition may be relevant or not, useful or not, possess certain advantages or disadvantages. Reasonable conditions for the choice of the OHC reference state are discussed in points (ii) and (iii) below eq. (5).

The related sentence has been modified as:

"The **reference state** relative to which OHC is measured may freely be specified at will, but beneficially be chosen with respect to its convenience or usefulness."